# CAN STOCHASTIC GRADIENT LANGEVIN DYNAMICS PROVIDE DIFFERENTIAL PRIVACY FOR DEEP LEARNING?

## ABSTRACT

Bayesian learning via Stochastic Gradient Langevin Dynamics (SGLD) has been suggested for differentially private learning. While previous research provides differential privacy bounds for SGLD when close to convergence or at the initial steps of the algorithm, the question of what differential privacy guarantees can be made in between remains unanswered. This interim region is essential, especially for Bayesian neural networks, as it is hard to guarantee convergence to the posterior. This paper will show that using SGLD might result in unbounded privacy loss for this interim region, even when sampling from the posterior is as differentially private as desired.

## 1 INTRODUCTION

Machine learning and, specifically, deep learning models show state-of-the-art results in various fields such as computer vision, natural language processing, and signal processing (e.g., Carion et al. (2020); Devlin et al. (2019); Balevi & Andrews (2021)). Training these models requires data, which in some problems, e.g., healthcare, finance, can include private information that should not be made public. Unfortunately, it has been shown (Fredrikson et al. (2015); Carlini et al. (2021)) that private information from the training data can sometimes be extracted from the trained model. One common approach to handle this issue is Differential Privacy (DP). Differential Privacy is a framework that ensures that the distribution of training output would be the same, even if we switch one of the training participants, thus ensuring privacy.

As privacy is usually obtained by adding random noise, it is natural to investigate whether Bayesian inference, which uses a distribution over models, can give private predictions. Previous works have shown that sampling from the posterior is differentially private under certain mild conditions (Wang et al. (2015); Foulds et al. (2016); Dimitrakakis et al. (2017)). The main disadvantage of this method is that sampling from the posterior is generally hard. The posterior usually does not have a closed-form solution, and iterative methods such as Markov Chain Monte Carlo (MCMC) are needed. While theoretical bounds on the convergence of MCMC methods for non-convex problems exist (Ma et al., 2019), they usually require an infeasible number of steps to guarantee convergence in practice.

Stochastic Gradient Langevin Dynamics (SGLD) is a popular MCMC algorithm used to approximately sample from an unnormalized distribution (Welling & Teh, 2011). The privacy guarantees of this specific sampling algorithm are interesting as it not only returns a sample from the posterior, which can be private, but the process itself of stochastic gradient descent with Gaussian noise mirrors the common Gaussian mechanism in DP. Previous work Wang et al. (2015) gives two disjoint privacy analyses: The first is for approximate sampling from the Bayesian posterior, which is only relevant when the SGLD almost converges. The second uses the standard DP analysis utilizing the Gaussian mechanism and the Advanced Composition theorem (Dwork & Roth, 2014), which only applies for a limited number of steps and is not connected to Bayesian sampling.

From these two lines of research, differential privacy bounds for SGLD are provided for its initial steps or when close to convergence. Neither of these cases is suitable for deep learning and many other problems, as one would limit the model's accuracy, and the other is unattainable in a reasonable time. Consequently, the privacy properties of SGLD in the interim region, between these two private

sections, are of high importance. One could speculate that since the initial steps of the algorithm are private, and it converges to the posterior that is also private, then sampling at the interim region will be private as well. If so, SGLD could be considered a solution for training differentially private deep neural networks. Unfortunately, as we will show, this is not the case.

**Our Contributions:** This work provides a counter-example, based on a Bayesian linear regression problem, showing that approximate sampling using SGLD might result in an unbounded loss of privacy in the interim regime. Moreover, this loss of privacy can even occur under strong conditions - when sampling from the posterior is as private as desired, and the problem is complex - even stronger conditions than what we can assume for most Deep Neural Network problems. This implies that special care should be given when using SGLD for private predictions, especially for problems where it is infeasible to guarantee convergence.

## 2 RELATED WORK

Several previous works investigate the connection between Bayesian inference and differential privacy (Wang et al. (2015); Foulds et al. (2016); Zhang et al. (2016); Dimitrakakis et al. (2017); Geumlek et al. (2017); Ganesh & Talwar (2020)). None of these papers provide guarantees over SGLD differential privacy in the interim regime. The closest work to ours is Wang et al. (2015) that specifically investigates stochastic MCMC algorithms such as SGLD. As mentioned, its analysis only covers the initial phase and when approximate convergence is achieved.

As many of the privacy bounds require sampling from the posterior, if SGLD is to be used, it requires non-asymptotic convergence bounds. Dalalyan (2014) provided non-asymptotic bounds on the error of approximating a target smooth and log-concave distribution by Langevin Monte Carlo. Cheng & Bartlett (2018) studied the non-asymptotic bounds on the error of approximating a target density $p^*$ where $\log p^*$ is smooth and strongly convex.

For the non-convex setting, Raginsky et al. (2017) showed non-asymptotic bounds on the 2-Wasserstein distance between SGLD and the invariant distribution solving Itô stochastic differential equation. However, to provide $(\epsilon, \delta)$ differential privacy, an algorithm should produce a distribution that is $O(\delta)$ close to neighbouring databases. Total Variation (for details about Total Variation see Tsybakov (2008)) is a more suitable distance for working with differential privacy. Ma et al. (2019) examined a target distribution $p^*$, which is strongly log-concave outside of a region of radius R, and where $-\ln p^*$ is $L$-Lipschitz. They provided a bound on the number of steps needed for the Total Variation distance between the distribution at the last step and $p^*$ to be smaller than $\epsilon$. This bound is proportional to $O(e^{32LR^2}\frac{d}{\epsilon^2})$, where $d$ is the model dimension. This result suggests that even little non-convexity will render running until close to convergence impractical. A conclusion from this work is that basing the differential privacy of SGLD on the proximity to the posterior is impractical for non-convex settings.

## 3 BACKGROUND

### 3.1 DIFFERENTIAL PRIVACY

Differential Privacy (Dwork et al. (2006b;a); Dwork (2011); Dwork & Roth (2014)) is a definition and a framework that enables performing data analysis on a database while reducing one's risk of exposing its personal data to the database. An algorithm is differentially private if it does not change its output distribution by much due to a single record change in its database.

**Definition 1.** *Approximate Differential Privacy: A randomized algorithm $M : \mathcal{D} \rightarrow Range(M)$ is $(\epsilon, \delta)$-differentially private if $\forall S \subseteq Range(M)$ and $\{\forall D, \hat{D} \in \mathcal{D} : \|D - \hat{D}\| \leq 1\}$ eq. 1 holds. $D, \hat{D}$ are called neighboring databases, and while the metric can change per application, Hamming distance is typically used.*

$$Pr[M(D) \in S] \leq \exp(\epsilon)Pr[M(\hat{D}) \in S] + \delta \tag{1}$$

Mironov (2017) suggested Rényi Differential Privacy (Definition 3), a relaxation to differential privacy, and a way to translate RDP guarantees into approximate differential privacy guarantees.

**Definition 2.** *Rényi Divergence (Rényi, 1961): For two probability distributions $Z$ and $Q$ over $\mathcal{R}$, the Réyni divergence of order $\nu > 1$ is*

$$\mathrm{D}_\nu(Z||Q) \triangleq \frac{1}{\nu - 1} \log \mathbb{E}_{x \sim Q}\left[\left(\frac{Z(x)}{Q(x)}\right)^\nu\right].$$

**Definition 3.** $(\nu, \epsilon)$*-RDP: A randomized mechanism $f : \mathcal{D} \rightarrow \mathcal{R}$ is said to have $\epsilon$-Rényi differential privacy of order $\nu$, or $(\nu, \epsilon)$-RDP in short, if for any adjacent databases $D, \hat{D} \in \mathcal{D}$ eq. 2 holds, where $\mathrm{D}_\nu$ is Rényi divergence of order $\nu$.*

$$\mathrm{D}_\nu(f(D)||f(\hat{D})) \leq \epsilon \tag{2}$$

**Lemma 3.1.** *(Mironov (2017) Proposition 3). If $f$ is $(\nu, \epsilon)$-RDP, it also satisfies $(\epsilon + \frac{\log \frac{1}{\delta}}{\nu - 1}, \delta)$ Differential Privacy for any $0 < \delta < 1$.*

### 3.2 STOCHASTIC GRADIENT LANGEVIN DYNAMICS

Stochastic Gradient Langevin Dynamics (SGLD) is an MCMC method that is commonly used for Bayesian Inference (Welling & Teh, 2011). The update step of SGLD is shown in eq. 3, where $\theta_j$ is the parameter vector at step $j$, $\eta_j$ is the step size at step $j$, $p(\theta_j)$ is the prior distribution, $p(y_i|\theta_j)$ is the likelihood of sample $y_i$ given model parameterized by $\theta_j$, $b$ is the batch size, and $n$ is the database size. SGLD can be seen as a Stochastic Gradient Descent with Gaussian noise, where the variance of the noise is calibrated to the step size.

$$
\begin{aligned}
\theta_{j+1} &= \theta_j + \frac{\eta_j}{2}\left[\nabla_{\theta_j} \ln p(\theta_j) + \frac{n}{b}\sum_{i=1}^{b} \nabla_{\theta_j} \ln p(y_{i_j}|\theta_j)\right] + \sqrt{\eta_j}\xi_j \\
i_j &\sim uniform\{1, ..., n\} \\
\xi_j &\sim \mathcal{N}(0, 1)
\end{aligned}
\tag{3}
$$

A common practice in deep learning is to use *cyclic* Stochastic Gradient Descent. This flavour of SGD first randomly shuffles the database samples and then cyclically uses the samples in this order. For optimization, there is empirical evidence that it works as well or better than SGD with reshuffling, and it was conjectured that it converges at a faster rate (Yun et al. (2021)). Cyclic-SGLD is the analog of cyclic-SGD for SGLD, where the difference is the use of the SGLD step instead of the SGD step. For simplicity, we will consider cyclic-SGLD in this work.

## 4 METHOD

Our goal is to prove that even when the posterior is as private as desired, sampling using SGLD for $T$ steps can be as non-private as desired. This requires analysing the distribution of SGLD after $T$ steps, which is hard in the general case. However, we show that we can get the desired behaviour when looking at a simple Bayesian linear regression problem where everything is a Gaussian with closed-form expressions. Our result is summarized in theorem 1.

**Theorem 1.** $\forall \delta < 0.5, \epsilon, \epsilon'$ *there exists a domain and a Bayesian inference problem where a single sample from the posterior distribution is $(\epsilon, \delta)$ differentially private, but, there is a number, $T$, for which performing approximate sampling by running SGLD for $T$ steps is not $(\epsilon', \delta)$ differentially private.*

As $\epsilon'$ can be as big as desired, and $\epsilon$ can be as small as desired, a corollary of Theorem 1 is that we could always find a problem for which the posterior is $(\epsilon, \delta)$ differentially private, but there will be a step in which SGLD will result in unbounded loss of privacy. Therefore, SGLD alone can not provide any privacy guarantees in the interim regime, even if the posterior is private.

To prove our theorem, we consider a Bayesian regression problem for a linear model with Gaussian noise, as defined in eq. 4, on domain $\mathcal{D}$ defined in eq. 5.

$$y = \theta x + \xi$$
$$\xi \sim \mathcal{N}(0, \beta^{-1})$$
$$\theta \sim \mathcal{N}(0, \alpha^{-1}) \tag{4}$$
$$\log p(y|x, \theta) = -\beta(y - \theta x)^2/2 - \frac{1}{2}\log(2\pi/\beta)$$

$$\mathcal{D}(n, \gamma_1, x_h, x_l, c) =$$
$$\{x_i, y_i \mid |\frac{y_i}{x_i} - c| \leq n^{\gamma_1}; \ x_i, y_i, c, \gamma_1 \in \mathbb{R}_{>0}; \ x_l \leq x_i \leq x_h\}_{i=1}^{n} \tag{5}$$
We assume that $x_h^2\beta > 3$ and that $\gamma_1 < \frac{1}{2}$

$n, c, x_l, x_h, \gamma_1$ are parameters of the problem ($c, x_l, x_h,$ and $\gamma_1$ are used, together with the database size - $n$, to bound the database samples to a chosen region). For every $\epsilon, \epsilon'$ and $\delta$, we will show there exist parameters $n, c, x_l, x_h, \gamma_1$ that have the privacy properties required to prove Theorem 1. The restrictions on the dataset simplify the proof but are a bit unnatural as it assumes we approximately know $c$, the parameter we are trying to estimate. Later we show in subsection 4.3 that they can be replaced with a Propose-Test-Release phase. We will address the problem of Bayesian Linear Regression for the model described in eq. 4 on domain $\mathcal{D}$ as *Bayesian linear regression problem on domain $\mathcal{D}$*. This problem has a closed-form solution for both the posterior distribution and the distribution at each SGLD step, thus enabling us to get tight bounds on the differential privacy in each case.

The heart of our proof is showing that for $n$ big enough sampling from the posterior is $(\epsilon, \delta)$ differentially private, with $\epsilon \sim \mathcal{O}(\frac{c^2}{n^3})$, while for SGLD there exists a step in which releasing a sample will not be $(\epsilon', \delta)$ differentially private for $\epsilon' = \Omega(\frac{c^2}{n^2})$. Therefore, by considering instances of the problem where $c \sim \mathcal{O}(n^{\frac{3}{2}}\sqrt{\epsilon})$ and $n$ is big enough, sampling from the posterior will be $(\epsilon, \delta)$ differentially private, while there will be an SGLD step in which releasing a sample will not be $(\epsilon', \delta)$ differentially private for $\epsilon' = \Omega(n\epsilon)$. We note that the bounds contain dependency over $\delta$, but since we are using a fixed and equal $\delta$ for both the posterior and SGLD privacy analysis, we omit it from the bounds for simplicity.

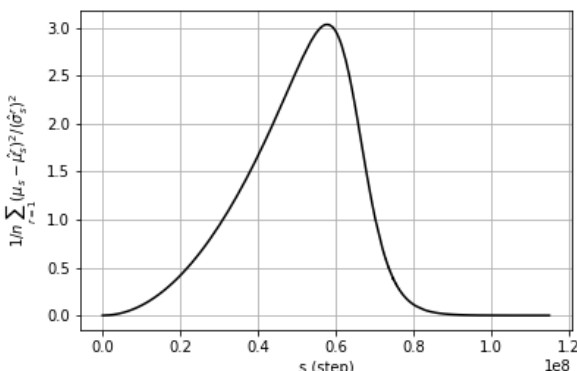

Figure 1: A value indicative of the distance between the distributions of samples from two SGLD processes running on neighbouring databases for the linear Gaussian model. For details on $\mu_s, \mu_s^r, \sigma_s^r$ see subsection 4.2. The parameters used for this experiment ($n = 1149019, \alpha = 0.1, \beta = 1, x_h = 1.8, x_l = 0.9, \gamma_1 = 0.1, \gamma_2 = 1.001, c = 1165165$) ensure $(1.6, 0.01)$ differential privacy when sampling from the posterior.

Figure 1 depicts an indicative value of the distance between the distributions of samples from two SGLD processes running on adjacent databases for the *Bayesian linear regression problem*. As we

will later show, SGLD on one of these examples is a Gaussian while the other is a mixture of $n$ Gaussians. We plot $\frac{1}{n}\sum_i \frac{(\mu_t - \mu_t^i)^2}{(\sigma_t^i)^2}$, where $\mu_t$ is the mean of the single Gaussian at timestep $t$, $\mu_t^i$ is the mean of the $i$'th Gaussian component at timestep $t$ and $(\sigma_t^i)^2$ its variance. We can see that even though the distributions are close at the initial iterations and at convergence (which implies differential privacy in those areas), in the interim region, they are significantly apart, which implies a lack of differential privacy.

## 4.1 POSTERIOR SAMPLING PRIVACY

To prove Theorem 1, we first need to show that $\forall \delta < 0.5, \epsilon$, there exists a domain and a Bayesian inference problem where a single sample from the posterior distribution is $(\epsilon, \delta)$ differentially private. In order to do so, this section will consider the differential privacy guarantees provided by one sample from the posterior for the *Bayesian linear regression problem* on domain $\mathcal{D}$.

We begin by using a well-known result for the closed-form-solution of the posterior distribution for a Bayesian linear regression problem (see Bishop (2006) for further details). By using the parameters of our problem, we get Lemma 4.1.

**Lemma 4.1.** *The posterior distribution for Bayesian linear regression problem on domain $\mathcal{D}$ is*

$$p(\theta|D) = \mathcal{N}(\theta; \mu, \sigma^2); \ \mu = \frac{\sum_{i=1}^n x_i y_i \beta}{\alpha + \sum_{i=1}^n x_i^2 \beta}; \ \sigma^2 = \frac{1}{\alpha + \sum_{i=1}^n x_i^2 \beta}. \tag{6}$$

Using the posterior distribution, one can calculate the Renyi divergence between every two neighbouring databases, thus getting an expression for the Rényi differential privacy, as shown in Lemma 4.2.

**Lemma 4.2.** *For a Bayesian linear regression problem on domain $\mathcal{D}$, such that $n > \max\{1 + 10\frac{x_h^2}{x_l^2}\frac{\nu}{\beta}, 1 + \nu \frac{x_h^2}{x_l^2}\}$, one sample from the posterior is $(\nu, \epsilon_1)$-Rényi differentially private. $\epsilon_1 \sim \mathcal{O}(\frac{c^2}{n^3})$ for $c >> n^{1+\gamma_1}$.*

$$\epsilon_1 = \frac{x_h^2}{2(n-1)x_l^2} + \frac{1}{2}(\nu - 1)\frac{\nu x_h^2}{(n-1)x_l^2 - \nu x_h^2} + 2\nu\beta\frac{x_h^4}{\frac{9}{10}n^{1-2\gamma_1}x_l^2} +$$
$$2\nu\beta \cdot \frac{(x_h^2\beta)(x_h^2\alpha + x_h^4\beta)}{\frac{9}{10}(x_l^2\beta)^2} \cdot \frac{(c+n^{\gamma_1})}{n^{2-\gamma_1}} + \frac{\nu}{2} \cdot \frac{(x_h^2\alpha + x_h^4\beta)^2}{\frac{9}{10}x_l^6\beta} \cdot \frac{(c+n^{\gamma_1})^2}{n^3}.$$

We can show that for $c >> n^{1+\gamma_2}$, each of the terms is bounded by $\mathcal{O}(\frac{c^2}{n^3})$. The first and second terms are bounded by $\mathcal{O}(\frac{1}{n})$. The third term is bounded by $\mathcal{O}(n^{2\gamma_1-1})$. Noticing that $n^{2\gamma_1-1} = \frac{n^{2(1+\gamma_1)}}{n^3} < \frac{c^2}{n^3}$, we get that the third term is bounded by $\mathcal{O}(\frac{c^2}{n^3})$. As $c >> n^{\gamma_1}$, the fourth term is bounded by $\mathcal{O}(\frac{cn^{\gamma_1}}{n^2})$, and since $\frac{cn^{\gamma_1}}{n^2} = \frac{cn^{1+\gamma_1}}{n^3} < \frac{c^2}{n^3}$, the term is bounded by $\mathcal{O}(\frac{c^2}{n^3})$. Lastly, since $c >> n^{1+\gamma_1}$, the last term is bounded by $\mathcal{O}(\frac{c^2}{n^3})$. For the full proof, see subsection A.1 in the appendix.

Translating the Rényi differential privacy guarantees into approximate differential privacy terms can be done according to Lemma 3.1, which gives Lemma 4.3.

**Lemma 4.3.** *With the conditions of Lemma 4.2, one sample from the posterior is $(\epsilon_1 + \frac{\ln(\frac{1}{\delta})}{\nu - 1}, \delta)$ differentially private.*

By choosing $\nu$ such that $\frac{\ln(\frac{1}{\delta})}{\nu - 1} < \frac{\epsilon}{2}$ and then choosing $n$ big enough such that $\epsilon_1 < \frac{\epsilon}{2}$, we get that the posterior is $(\epsilon, \delta)$ differentially private.

## 4.2 STOCHASTIC GRADIENT LANGEVIN DYNAMICS PRIVACY

To complete the proof of Theorem 1, we need to show that even if one sample from the posterior is $(\epsilon, \delta)$ differentially private for a *Bayesian linear regression problem on domain $\mathcal{D}$*, it does not

provide any guarantees on the privacy of SGLD for that problem. In order to do so, this section will first consider the loss in privacy when using SGLD for the *Bayesian linear regression problem on domain $\mathcal{D}$*, and then, together with the results of section 4.1, will prove Theorem 1.

In order to show that SGLD is not differentially private after initial steps and before convergence, it is enough to find two neighbouring databases for which the loss in privacy is as big as desired in those steps. We define neighbouring databases $D_1$ and $D_2$ in eq. 7 and consider the *Bayesian linear regression problem* on $D_1$ and $D_2$. We set the learning rate to be $\eta = \frac{2}{(\alpha + n x_h^2 \beta)^2}$.

$$D_1 = \{x_i, y_i : x_i = x_h, y_i = c \cdot x_h\}_{i=1}^n$$
$$D_2 = \{x_i, y_i : x_i = x_h, y_i = c \cdot x_h\}_{i=1}^{n-1} \cup \{\frac{x_h}{2}, c \cdot \frac{x_h}{2}\} \tag{7}$$

To tightly analyze the differential privacy loss when approximately sampling via SGLD at each step, we need to get a closed-form solution for the distribution for each step. For database $D_1$, the solution is Normal distribution. For database $D_2$, different shuffling of samples produces different Gaussian distributions, therefore giving a mixture of Gaussians.

We look at cyclic-SGLD with a batch size of 1 and mark by $\theta_j, \hat{\theta}_j$ the samples on the $j$'th SGLD step when using databases $D_1$ and $D_2$ accordingly. Since $D_1$ samples are all equal, the update step of the cyclic-SGLD is the same for every step (with different noise generated for each step). This update-step contains only multiplication by a scalar, addition of a scalar, and addition of Gaussian noise, therefore, together with a conjugate prior results in Normal distribution for $\theta_j$: $\mathcal{N}(\theta_j; \mu_j, \sigma_j^2)$.

For $D_2$, there is only one sample different from the rest. We mark by $r$ the index in which this sample is used in the cyclic-SGLD and call this order $r$-order. Note that there are only $n$ different values for $r$ and, as such, effectively only $n$ different samples orders. Since every order of samples is chosen with the same probability, $r$ is distributed uniformly in $\{1, .., n\}$. We mark by $\hat{\theta}_j^r$ the sample on the $j$'th SGLD step when using $r$-order. Since, for a given order, $\hat{\theta}_j^r$ is formed by a series of multiplications by a scalar, addition of scalar, and addition of Gaussian noise, and since the prior is also Gaussian, then $\hat{\theta}_j^r$ is distributed Normally, $\mathcal{N}(\hat{\theta}_j^r; \hat{\mu}_j^r, (\hat{\sigma}_j^r)^2)$. As $r$ is distributed uniformly, $\hat{\theta}_j$ distribution mass is distributed evenly between all $\hat{\theta}_j^r$, resulting in a mixture of Gaussians.

Intuitively what will happen is that each Gaussian components ,$\hat{\theta}_j$ as well as $\theta_j$, will move towards the similar posterior Gaussian. However, at each epoch, $\hat{\theta}_j$ will drag a bit behind because in one batch one gradient is smaller. While this gap can be quite small, for large $n$, the Gaussians are very peaked with very small standard deviations; thus, they separate enough that we can easily distinguish between the two distributions.

According to approximate differential privacy definition (Definition 1), it is enough to find one set $S$ such that $p(\theta_j \in S) > e^\epsilon p(\hat{\theta}_j \in S) + \delta$ to show that releasing $\theta_j$ is not $(\epsilon, \delta)$ private. We choose $S = \{s | s > \mu_j\}$ at some step $j$ we will define later on. It is clear from symmetry that $p(\theta_j > \mu_j) = 1/2$, and by using Chernoff bound we can bound $p(\hat{\theta}_j > \mu_j)$.

**Lemma 4.4.** $p(\hat{\theta}_j > \mu_j) \leq \frac{1}{n} \sum_{r=1}^n \exp(-\frac{(\mu_j - \hat{\mu}_j^r)^2}{2(\hat{\sigma}_j^r)^2})$.

Using Lemma 4.4, we can upper bound the mass of $\hat{\theta}_j$ in $S$, and thus lower bound the difference between $\theta_j$ and $\hat{\theta}_j$ distribution masses in $S$ for some step - $j$. To use Lemma 4.4, we first need to lower bound $\frac{(\mu_j - \hat{\mu}_j^r)^2}{(\sigma_j^r)^2}$ for a certain step. This is done in Lemma 4.5.

**Lemma 4.5.** $\exists k \in \mathbb{Z}_{>0}$ such that $\frac{(\mu_{(k+1)n} - \hat{\mu}_{(k+1)n}^r)^2}{(\hat{\sigma}_{(k+1)n}^r)^2} = \Omega(\frac{c^2}{n^2})$, for $n$ big enough.

To prove Lemma 4.5, we first find closed-form solutions for $\hat{\theta}_{(k+1)n}^r, \theta_{(k+1)n}$ distributions (Lemma A.1). Using the closed-form solutions, we find a lower bound over $(\mu_{(k+1)n} - \hat{\mu}_{(k+1)n}^r)^2$ as a function of $k$, which applies for all $k$ (Lemma A.2). To upper bound $(\hat{\sigma}_{(k+1)n}^r)^2$, we find an approximation to the epoch in which the data and prior effects on the variance are approximately equal,

marked $\dot{k}$. We choose the step in which we will consider the privacy loss as $(\lceil \dot{k} \rceil + 1)n$ and show that $(\hat{\sigma}^r_{(\lceil \dot{k} \rceil + 1)n})^2$ is upper bounded at this step (Lemma A.4). Using the upper bound on the difference in means and the lower bound on the variance, Lemma 4.5 is proved. By using the lower bound from Lemma 4.5 in Lemma 4.4, we get Lemma 4.6.

**Lemma 4.6.** *For the Bayesian linear regression problem over database $D_1$, such that $n$ is big enough, $\exists T \in \mathbb{Z}_{>0}$ such that approximate sampling by running SGLD for $T$ steps will not be $(\epsilon, \delta)$ private for $\epsilon < \Omega(\frac{c^2}{n^2}), \delta < 0.5$.*

From Lemma 4.3, we see that sampling from the posterior is $(\epsilon, \delta)$ differentially private for $\epsilon = \mathcal{O}(\frac{c^2}{n^3})$. From Lemma 4.6, we see that for SGLD, there exists a step in which releasing a sample will not be $(\epsilon', \delta)$ differentially private for $\epsilon' = \Omega(\frac{c^2}{n^2})$. Therefore, considering instances of the problem where $c = \mathcal{O}(n^{\frac{3}{2}}\sqrt{\epsilon})$, sampling from the posterior will be $(\epsilon, \delta)$ differentially private. However, there will be an SGLD step in which releasing a sample will not be $(\epsilon', \delta)$ differentially private for $\epsilon' = \Omega(n\epsilon)$. Since we can choose $n$ to be big as desired, we can make the lower bound over $\epsilon'$ as big as we desire. This completes the proof of Theorem 1.

## 4.3 Propose Test Sample

Our analysis of the posterior and SGLD is done on a restricted domain $\mathcal{D}$ as defined in eq. 5. These restrictions over the dataset simplify the proof but are a bit unnatural as they assume we approximately know $c$, the parameter we are trying to estimate. This section shows that these restrictions could be replaced with a Propose-Test-Release phase (Dwork & Lei, 2009) and common practices in deep learning.

When training a statistical model, it is common to first preprocess the data by enforcing it in a bounded region and removing outliers. After the data is cleaned, the training process is performed. This is especially important in DP, as outliers can significantly increase the algorithm's sensitivity to a single data point and thus hamper privacy.

Informally, algorithm 1 starts by clipping the input to the accepted range. It then estimates a weighted average of the ratio $\frac{y_i}{x_i}$ (line 12) and throws away outliers that deviate too much from it. The actual implementation of this notion is a bit more complicated because of the requirement to do so privately. Once the database is cleaned, algorithm 1 privately verifies that the number of samples is big enough, so the sensitivity of $p(\theta|W)$ to a single change in the database will be small, therefore making sampling from $p(\theta|W)$ $(\epsilon, \delta)$ differentially private. This method is regarded as Propose-Test-Release, where we first propose a bound over the sensitivity, then test if the database holds this bound, and finally release the result if so.

We define $n_{min}$ in eq. 26 in the appendix to be the minimum size of $W$ for which the algorithm will sample from $p(\theta|W)$ with high probability. We will show later on that this limit ensures that sampling from $p(\theta|W)$ is $(\epsilon, \delta)$ differentially private.

We define $p(\theta|W)$ to be the posterior for the *Bayesian linear regression problem* over database W. From Lemma 4.1, it follows that $p(\theta|W)$ has the form of

$$p(\theta|W) = \mathcal{N}(\theta; \mu, \sigma^2); \ \mu = \frac{\sum_{(x_i, y_i) \in W} x_i y_i \beta}{\alpha + \sum_{(x_i, y_i) \in W} x_i^2 \beta}; \ \sigma^2 = \frac{1}{\alpha + \sum_{(x_i, y_i) \in W} x_i^2 \beta}.$$

**Claim 4.1.** *Algorithm 1 is $(5\epsilon, 2\delta)$ differentially private.*

By claim C.9, steps 6-13 are $(3\epsilon, \delta)$ differentially private. By corollary C.3, steps 14-19 are $(2\epsilon, \delta)$ differentially private for given $\breve{m}$ and $n_2$. Therefore by the sequential composition theorem, the composition is $(5\epsilon, 2\delta)$ differentially private. The claim proved by noticing that if steps 6-19 are private with respect to the updated database (after step 5), then they are also private for the original database.

**Claim 4.2.** *When replacing line 19 with sampling via SGLD with step size $\eta = \frac{1}{(\alpha + n_1 x_h^2 \beta)^2}$, then $\exists T(n_1) : \mathbb{Z}_{>0} \to \mathbb{Z}_{>0}$ such that the updated algorithm is not $(\epsilon, \delta)$ differentially private $\forall \epsilon \in \mathbb{R}_{>0}, \delta < \frac{1}{6}$ if ran for $T(n_1)$ steps.*

**Algorithm 1** Propose Test Sample

---

**Input:** $D = \{x_i, y_i\}_{i=1}^{n_1}$

**Parameters:** $\epsilon, \delta < 0.5, x_l > 0, x_h > x_l, \alpha > 0, \beta \geq \frac{3}{x_h^2}, \rho_1 \in (1, \frac{3}{2}), \rho_2 \in (0, \frac{1}{2}), \gamma_1 \in (\rho_2, \frac{1}{2})$

1: **for** $i = 1, 2, \ldots, N$ **do**
2:      $x_i \leftarrow \max\{x_i, x_l\}$
3:      $x_i \leftarrow \min\{x_i, x_h\}$
4:      $y_i \leftarrow \max\{y_i, 0\}$
5: **end for**
6: $\breve{n}_1 \leftarrow n_1 - \frac{1}{\epsilon} \log \frac{1}{2\delta} + \mathrm{Lap}(\frac{1}{\epsilon})$
7: $V = \{x_i, y_i | \frac{y_i}{x_i} \leq \breve{n}_1^{\rho_1}\}$
8: $n_2 \leftarrow |V| - \frac{1}{\epsilon} \log \frac{1}{2\delta} + \mathrm{Lap}(\frac{1}{\epsilon})$
9: **if** $n_2 \leq 1$ **then**
10:      return null
11: **end if**
12: $m \leftarrow \frac{\sum_{(x_i, y_i) \in V} x_i y_i}{\sum_{(x_i, y_i) \in V} x_i^2}$
13: $\breve{m} \leftarrow m + \mathrm{Lap}(\frac{1}{\epsilon} \breve{n}_1^{\rho_1} \frac{2(n_2-1)x_h^2 x_l^2 + x_h^4}{n_2(n_2-1)x_l^4})$
14: $W \leftarrow \{(x_i, y_i) : |\frac{y_i}{x_i} - \breve{m}| \leq n_2^{\rho_2}\}$
15: $n_W \leftarrow |W| - \frac{1}{\epsilon} \log(\frac{1}{2\delta}) + \mathrm{Lap}(\frac{1}{\epsilon})$
16: **if** $n_W < n_{min}$ **then**
17:      return null
18: **end if**
19: return sample from $p(\theta|W)$

---

Proof sketch (See appendix for full proof). We first note that by choosing $1 + \rho_2 > \rho_1$, the sensitivity of $\breve{m}$ grows slower than the bound over the distance $|\frac{y_i}{x_i} - \breve{m}|$. Therefore for $n_1$ big enough, samples for which $\frac{y_i}{x_i} = m$ will be included in $W$ with high probability. Consequently, databases $D_3, D_4 \in \mathcal{D}$ will reach, with high probability, to step 19, which from our previous analysis over SGLD (see subsection 4.2) will cause an unbounded loss in privacy.

$$
\begin{aligned}
&\rho_1 > \rho_3 > 1 \\
&D_3 = \{x_i, y_i : x_i = x_h, y_i = n_1^{\rho_3} \cdot x_h\}_{i=1}^{n_1} \\
&D_4 = \{x_i, y_i : x_i = x_h, y_i = n_1^{\rho_3} \cdot x_h\}_{i=1}^{n-1} \cup \{\frac{x_h}{2}, n_1^{\rho_3} \cdot \frac{x_h}{2}\}
\end{aligned}
\tag{8}
$$

## 5   WASSERSTEIN DISTANCE AND DIFFERENTIAL PRIVACY

As we have shown in Theorem 1, one cannot give any DP guarantees for SGLD in the interim region. That means that to get private samples using SGLD, one must limit the number of iterations, thus utilizing the Gaussian mechanism, or run until approximate convergence. Therefore, it is of interest to get non-asymptotic convergence bounds for SGLD so that we guarantee privacy after a known number of steps. Previously, several works have given non-asymptotic bounds; however, some of those do so for the 2-Wasserstein metric (Raginsky et al. (2017); Cheng et al. (2018)). This is unfortunate as the 2-Wasserstein metric is unsuitable for differential privacy - it is easy to create two distributions with 2-Wasserstein distance as small as desired but with disjoint support.

It is, however, interesting to ask whether combining bounds on the 2-Wasserstein metric with Lipschitz continuous probability densities will allow us to get privacy guarantees. The intuition why this should be enough is simple: If $p, q$ are two distributions with small 2-Wasserstein distance, then there is (under mild conditions) a mapping, $f : \mathcal{X} \to \mathcal{X}$, such that the pushforward maintains $f_\sharp p = q$ (i.e. for each measurable set $S$ $q(S) = p(f^{-1}(S))$) and that $\mathbb{E}_p[||x - f(x)||^2] < \epsilon$. One can assume that $p(x) \approx q(f(x))$ and $q(x) \approx q(f(x))$ as $x \approx f(x)$ with high probability. Unfortunately, this intuition does not hold exactly, as the map $f$ can change the density considerably but still be a

pushforward by changing the volume. For example, if we assume $f$ is smooth and bijective, we get the standard change of variable formula such that $p(x) = q(f(x)) |\det(J_f)|$, so $p(x) \approx q(f(x))$ only if $|\det(J_f)| \approx 1$. This issue becomes more severe as the dimensionality increases.

For completeness, we will share our results connecting $p(x)$ to $q(x)$ when $\mathcal{W}_2(p, q)$ is small, and both distributions are $L$-Lipschitz continuous. This bound scale poorly with dimension, and as such ill-suited for SGLD on deep networks, but can still be useful for Bayesian sampling in low-dimensional problems.

For distribution $p$, we define the density $p_\lambda(x)$ as the average of $p(x)$ on a ball of radius $\lambda$ centered around $x$ - $p_\lambda(x) = \frac{1}{vol_d(\lambda)} \int_{B_\lambda^d(0)} p(x + z)dz$, where $B_\lambda^d(x)$ is the ball in $\mathbb{R}^d$ of radius $\lambda$ centered around $x$, and $vol_d(\lambda)$ is its volume.

**Claim 5.1.** *For $L$-Lipschitz continuous distribution $p$ we have $|p(x) - p_\lambda(x)| \leq \lambda L$.*

**Theorem 2.** *Let $P, Q$ be absolutely continuous w.r.t the Lebesgue measure in $\mathbb{R}^d$, with finite second-moment and $L$-Lipschitz continuous densities $p, q$. If $\mathcal{W}_2(p, q) < \epsilon^2$ then we have*

$$p_\lambda(x) \leq \frac{vol_d(\lambda)}{vol_d(\lambda - \epsilon)} q_\lambda(x) + \left( \frac{vol_d(\lambda)}{vol_d(\lambda - \epsilon)} - 1 \right) 2\lambda L + \frac{\epsilon}{vol_d(\lambda - \epsilon)}. \tag{9}$$

The proof is an extension of the proof of theorem 2.1 in Walker (2004) to dimensions larger than 1. The detailed proof is in the supplementary material.

It is easy to see that as $\frac{vol_d(\lambda)}{vol_d(\lambda - \epsilon)} = \left( 1 + \frac{\epsilon}{\lambda - \epsilon} \right)^d$, the bounds usefulness quickly diminishes with dimensionality as it requires extremely small $\epsilon$ to give non-vacuous results.

This, however, can still give useful results in low-dimensional problems.

## 6 CONCLUSION

As shown in this work, while SGLD has interesting connections to privacy and some guarantees, caution is required if one wishes to use it to get private predictions. This is especially important for models such as deep neural networks, where it is infeasible to guarantee convergence.

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

## A  SGLD AND POSTERIOR PRIVACY

*Proof Theorem 1.*  Define

$$\frac{1}{2} > \gamma_1 > 0; \ \frac{3}{2} > \gamma_2 > 1 + \gamma_1; \ x_l = \frac{x_h}{2}$$

$$\nu_1 = \frac{2\ln(\frac{1}{\delta})}{\epsilon} + 1$$

$$n_1 = \max\{\frac{1}{2\alpha x_h^2 \beta} - \frac{1}{x_h^2 \beta}, \frac{\alpha}{x_h^2 \beta}, \frac{\alpha}{x_h^2 \beta}(e^{\frac{2}{x_h^2 \beta}} - 2) + \frac{1}{2x^2 \beta}\}$$

$$n_2 = \max\{1 + \frac{x_h^2}{x_l^2}\frac{8}{\epsilon}, 1 + \nu\frac{x_h^2}{x_l^2}(1 + 8\frac{(\nu-1)}{\epsilon}), (\frac{16\nu\beta x_h^4}{\frac{9}{10}\epsilon x_l^2})^{\frac{1}{1-2\gamma_1}},$$

$$(\frac{16\nu\beta}{\epsilon}(\frac{(x_h^2\beta)(x_h^2\alpha + x_h^4\beta)}{\frac{9}{10}(x_l^2\beta)^2})(1 + \frac{1}{(1 + 10\frac{x_h^2}{x_l^2}\frac{\nu}{\beta})^{\gamma_2 - \gamma_1}}))^{\frac{1}{2-\gamma_1-\gamma_2}},$$

$$(\frac{4\nu}{\epsilon}(\frac{(x_h^2\alpha + x_h^4\beta)^2}{\frac{9}{10}x_l^6\beta})(1 + \frac{1}{(1 + 10\frac{x_h^2}{x_l^2}\frac{\nu}{\beta})^{\gamma_2 - \gamma_1}}))^{\frac{2}{3-2\gamma_2}}\}$$

$$n_3 = \max\{1 + 10\frac{x_h^2}{x_l^2}\frac{\nu_1}{\beta}, 1 + \nu\frac{x_h^2}{x_l^2}\}$$

$$n_p = \max\{n_1, n_2, n_3, ((\epsilon' - \ln(0.5 - \delta))e^{\frac{2}{x^2\beta}}(\frac{32x_h^2\beta}{3})^2\frac{2\nu_1}{\alpha})^{\frac{1}{2(\gamma_2 - 1)}}\}$$

$$v_1 = \max\{6, 1 + 2e^{\frac{1}{x_h^2\beta}}\}$$

$$c_p = n_p^{\gamma_2}$$

We consider the *Bayesian linear regression problem* over database $D_1$ (defined in eq. 7) with $n = n_p$ and $c = c_p$. Since $n_p > n_1$, the problem holds the constraints of lemma $A.6$. Consequently, there exists a step for which one approximate sample from the posterior using SGLD is not $(\epsilon'', \delta)$ private for all $\epsilon''$ such that $\epsilon'' \le e^{-\frac{2}{x^2\beta}}\frac{\alpha}{2v_1}(\frac{3}{32x_h^2\beta})^2(\frac{c_p}{n_p})^2 + \ln(0.5 - \delta)$. From eq. 10, the choice of $n_p$ promises that $\epsilon' \le e^{-\frac{2}{x^2\beta}}\frac{\alpha}{2v_1}(\frac{3}{32x_h^2\beta})^2(\frac{c_p}{n_p})^2 + \ln(0.5 - \delta)$; Therefore approximate sampling from the posterior using SGLD is not $(\epsilon', \delta)$ differentially private.

Since $n_p > n_2$ and $n_p > n_3$, the problem holds the constraints of Claim D.30. Therefore one sample from the posterior is $(\epsilon, \delta)$ differentially private.

$$e^{-\frac{2}{x_h^2\beta}}\frac{\alpha}{2v_1}(\frac{3}{32x_h^2\beta})^2(\frac{c_p}{n_p})^2 + \ln(0.5 - \delta) \ge \epsilon'$$

$$(\frac{c_p}{n_p})^2 \ge (\epsilon' - \ln(0.5 - \delta))e^{\frac{2}{x_h^2\beta}}(\frac{32x_h^2\beta}{3})^2\frac{2v_1}{\alpha}$$

$$n_p^{2(\gamma_2 - 1)} \ge (\epsilon' - \ln(0.5 - \delta))e^{\frac{2}{x_h^2\beta}}(\frac{32x_h^2\beta}{3})^2\frac{2v_1}{\alpha} \tag{10}$$

$$n_p \ge ((\epsilon' - \ln(0.5 - \delta))e^{\frac{2}{x_h^2\beta}}(\frac{32x_h^2\beta}{3})^2\frac{2v_1}{\alpha})^{\frac{1}{2(\gamma_2 - 1)}}$$

$\square$

### A.1  POSTERIOR SAMPLING PRIVACY

*Proof Lemma 4.1.*  eq. 11 is a known result for the Bayesian inference problem for a linear model with Gaussian noise with known precision parameter ($\beta$) and a conjugate prior (Bishop (2006) - 3.49-4.51. for details). By choosing the basis function to be $\phi(x) = x$, working in one dimension,

and choosing $\boldsymbol{m}_0 = 0$, $\boldsymbol{S}_0 = \alpha^{-1}$, we get the linear model defined in eq. 4 and matching posterior described in Lemma 4.1.

$$p(\boldsymbol{w}|\boldsymbol{t}) = \mathcal{N}(\boldsymbol{w}; \boldsymbol{m}_N, \boldsymbol{S}_N); \; \boldsymbol{m}_N = \boldsymbol{S}_N(\boldsymbol{S}_0^{-1}\boldsymbol{m}_0 + \beta\Phi^T\boldsymbol{t}); \; \boldsymbol{S}_N^{-1} = \boldsymbol{S}_0^{-1} + \beta\Phi^T\Phi \qquad (11)$$

$\square$

*Proof Lemma 4.2.* By definition 3, for a single sample from the posterior to be $(\nu, \epsilon')$ RDP, the Rényi divergence of order $\nu$ between any adjacent databases needs to be bounded. We consider two adjacent databases $D, \hat{D} \in \mathcal{D}$, and w.l.o.g, define that they differ in the last sample (where it is also allowed to be $(0, 0)$ for one of them, which saves us the need to consider also a neighbouring database with a size smaller by 1). To ease the already complex and detailed calculations, we use definitions in eq .12.

$$D = \{x_i, y_i\}_{i=1}^{n-1} \cup \{x_n, y_n\}, \; \hat{D} = \{x_i, y_i\}_{i=1}^{n-1} \cup \{\hat{x}_n, \hat{y}_n\}$$
$$z = \sum_{i=1}^{n-1} x_i^2, \; q = \sum_{i=1}^{n-1} y_i x_i \qquad (12)$$

According to Lemma 4.1 and with definitions in eq. 12, the posterior distributions are

$$p(\theta|D) = \mathcal{N}(\theta; \mu, \sigma^2); \; \mu = \frac{\beta(q + x_n y_n)}{\alpha + (z + x_n^2)\beta}; \; \sigma^2 = \frac{1}{\alpha + (z + x_n)\beta}$$
$$p(\theta|\hat{D}) = \mathcal{N}(\theta; \hat{\mu}, \hat{\sigma}^2); \; \hat{\mu} = \frac{\beta(q + \hat{x}_n \hat{y}_n)}{\alpha + (z + \hat{x}_n^2)\beta}; \; \hat{\sigma}^2 = \frac{1}{\alpha + (z + \hat{x}_n)\beta} \qquad (13)$$

By Gil et al. (2013), the Réyni divergence of order $\nu$, - $D_\nu(f_1||f_2)$ - for $f_1, f_2$, uni-variate normal distributions with means $\mu_1, \mu_2$ and variances $\sigma_1, \sigma_2$ accordingly, is

$$D_\nu(f_1||f_2) = \ln \frac{\sigma_1}{\sigma_2} + \frac{1}{2}(\nu - 1)\ln \frac{\sigma_2^2}{(\sigma_{f_1,f_2}^2)_\nu^*} + \frac{1}{2}\frac{\nu(\mu_1 - \mu_2)^2}{(\sigma^2)_\nu^*} .$$
$$(\sigma_{f_1,f_2}^2)_\nu^* = \nu\sigma_2^2 + (1 - \nu)\sigma_1^2 > 0$$

Therefore, for $p(\theta|D)$ and $p(\theta|\hat{D})$, the Rényi divergence of order $\nu$ is as shown in eq. 14, where we omit the subscript for $(\sigma^2)_\nu^*$ since it is clear from context to which distributions it applies.

$$D_\nu(p(\theta|D)||p(\theta|\hat{D})) = \ln \frac{\sigma}{\hat{\sigma}} + \frac{1}{2}(\nu - 1)\ln \frac{\hat{\sigma}^2}{(\sigma^2)_\nu^*} + \frac{1}{2}\frac{\nu(\mu - \hat{\mu})^2}{(\sigma^2)_\nu^*}$$
$$(\sigma^2)_\nu^* = \nu\hat{\sigma}^2 + (1 - \nu)\sigma^2 \qquad (14)$$

According to claim D.25, $(\sigma^2)_\nu^* > 0$. Therefore the value $D_\nu(p(\theta|D), p(\theta|\hat{D}))$ exists. In order to prove Rényi differential privacy, each of the terms of $D_\nu(p(\theta|D), p(\theta|\hat{D}))$ is bounded separately. The bounds on each of the terms are proved at claims D.26,D.27, and D.28. $\square$

*Proof Lemma 4.3.* By Lemma 4.2, sampling from the posterior is $(\nu, \epsilon_1)$-RDP, therefore by Lemma 3.1, sampling from the posterior is also $(\epsilon_1 + \frac{\ln(\frac{1}{\delta})}{\nu - 1}, \delta)$ differentially private. $\square$

## A.2 STOCHASTIC GRADIENT LANGEVIN DYNAMICS PRIVACY

*Proof Lemma 4.4.*

$$p(\hat{\theta}_j > \mu_j | D_2) = \sum_{r=1}^{n} p(\hat{\theta}_j^r > \mu_j | D_2) p(\hat{\theta}_j = \hat{\theta}_j^r | D_2) =$$

$$\sum_{r=1}^{n} p(\hat{\theta}_j - \hat{\mu}_j^r > \mu_j - \hat{\mu}_j^r | D_2) p(\hat{\theta}_j = \hat{\theta}_j^r | D_2) =$$

$$\frac{1}{n} \sum_{r=1}^{n} p(\hat{\theta}_j - \hat{\mu}_j^r > \mu_j - \hat{\mu}_j^r | D_2) \leq$$

$$\frac{1}{n} \sum_{r=1}^{n} \exp(-\frac{(\mu_j - \hat{\mu}_j^r)^2}{2(\sigma_j^r)^2})$$

Where the inequality holds due to Chernoff bound (For further details, see Wainwright (2019)). $\square$

*Proof Lemma 4.5.* By lemma A.5, for $n > \max\{\frac{\alpha}{x^2\beta}, \frac{\alpha}{x_h^2\beta}(e^{\frac{2}{x^2\beta}} - 2) + \frac{1}{2x^2\beta}, \frac{1}{2\alpha x_h^2\beta} - \frac{1}{x_h^2\beta}\}$ eq. 15 holds for some $\dot{k} \in \mathbb{R}_{>0}$. We can see that this lower bound is dominated by $\frac{c^2}{n^2}$, therefore proving Lemma 4.5.

$$\frac{(\mu_{(\lceil\dot{k}\rceil+1)n} - \hat{\mu}_{(\lceil\dot{k}\rceil+1)n}^r)^2}{(\sigma_{(\lceil\dot{k}\rceil+1)n}^r)^2} \geq e^{-\frac{2}{x_h^2\beta}} \frac{\alpha}{v_1}(\frac{3}{32x_h^2\beta})^2(\frac{c}{n})^2 \tag{15}$$

$$v_1 = \max\{6, 1 + 2e^{\frac{1}{x_h^2\beta}}\}$$

$\square$

*Proof Lemma 4.6.* By Lemma A.6, for $n > \max\{\frac{\alpha}{x_h^2\beta}, \frac{\alpha}{x_h^2\beta}(e^{\frac{2}{x_h^2\beta}} - 2) + \frac{1}{2x_h^2\beta}, \frac{1}{2\alpha x_h^2\beta} - \frac{1}{x_h^2\beta}\}$, there exists $T \in \mathbb{Z}_{>0}$ (Marked in Lemma A.6 as $\lceil\dot{k}\rceil$) such that running SGLD for the *Bayesian linear regression problem* over $D_1$ for $T$ steps will not be $(\epsilon, \delta)$ differentially for $\epsilon < \epsilon'$, as defined in eq. 16, and $\delta < 0.5$. Since $\epsilon'$ is dominated by $\frac{c^2}{n^2}$, this proves the lemma.

$$\epsilon' = e^{-\frac{2}{x^2\beta}} \frac{\alpha}{2v_1}(\frac{3}{32x_h^2\beta})^2(\frac{c}{n})^2 + \ln(0.5 - \delta) \tag{16}$$

$$v_1 = \max\{6, 1 + 2e^{\frac{1}{x_h^2\beta}}\}$$

$\square$

## A.3 STOCHASTIC GRADIENT LANGEVIN DYNAMICS DETAILED ANALYSIS

In order to ease the analysis of the SGLD process, markings in eq. 17 are used. $x_h, \alpha, \beta, c, n$ are as defined for the *Bayesian linear regression problem*, and $\eta$ is defined in subsection 4.2.

$$\lambda = [1 - \frac{\eta}{2}(\alpha + nx_h^2\beta)], \hat{\lambda} = [1 - \frac{\eta}{2}(\alpha + n(\frac{x_h}{2})^2\beta)], \rho = \frac{\eta}{2}ncx_h^2\beta, \hat{\rho} = \frac{\eta}{2}nc(\frac{x_h}{2})^2\beta \tag{17}$$

**Lemma A.1.** *The forms of $\hat{\theta}_{(k+1)n}^r$ are*

$$\hat{\theta}_{(k+1)n}^1 = \theta_0\hat{\lambda}^{k+1}\lambda^{(n-1)(k+1)} + \sum_{j=0}^{k}(\hat{\lambda}\lambda^{n-1})^j[\hat{\rho}\lambda^{n-1} + \rho\sum_{i=0}^{n-2}\lambda^i + \sqrt{\eta}\sum_{i=0}^{n-1}\lambda^i\xi_i]$$

$$\hat{\theta}_{(k+1)n}^{r>1} = \theta_0(\hat{\lambda}\lambda^{n-1})^{k+1} +$$

$$\Big(\sum_{i=1}^{r-1}(\rho + \sqrt{\eta}\xi)\hat{\lambda}\lambda^{n-i-1} + (\hat{\rho} + \sqrt{\eta}\xi)\lambda^{n-r} + \sum_{j=r+1}^{n}(\rho + \sqrt{\xi}\eta)\lambda^{n-j}\Big)\sum_{l=0}^{k}(\hat{\lambda}\lambda^{n-1})^l.$$

*Proof Lemma A.1.* Welling & Teh (2011) define the SGLD update rule as in eq. 3. This rule can be applied to the *Bayesian linear regression problem* over databases $D_1, D_2$ as following

$$p(\theta_j) = \mathbb{N}(\theta_j; 0, \alpha^{-1}) \Rightarrow \ln p(\theta_j) = \ln(\frac{1}{\sqrt{2\pi\alpha^{-1}}}) - \frac{1}{2}\theta_j^2\alpha \Rightarrow \nabla_{\theta_j} p(\theta_j) = -\theta_j\alpha$$

$$p(y_i|\theta_j) = \mathbb{N}(y_j; \theta_j x_i, \beta^{-1}) \Rightarrow \ln p(y_i|\theta_j) = \ln(\frac{1}{\sqrt{2\pi\beta^{-1}}}) - \frac{1}{2}(y_i - \theta_j x_i)^2\beta \Rightarrow$$

$$\nabla_{\theta_j} p(y_i|\theta_j) = (y_i - \theta_j x_i)x_i\beta \Rightarrow$$

$$\theta_{j+1} = \theta_j + \frac{\eta}{2}[-\theta_j\alpha + n(y_i - \theta_j x_j)x_i\beta] + \sqrt{\eta_j}\xi_i =$$

$$\theta_j[1 - \frac{\eta}{2}(\alpha + nx_j^2\beta)] + \frac{\eta}{2}ny_ix_i\beta + \sqrt{\eta}\xi_j =$$

$$\theta_j[1 - \frac{\eta}{2}(\alpha + nx_j^2\beta)] + \frac{\eta}{2}ncx_i^2\beta + \sqrt{\eta}\xi_j$$

By using standard tools for solving first-order non-homogeneous recurrence relations with variable coefficients, the value of $\hat{\theta}_n^1$ can be found.

$$\hat{\theta}_n^1 = \hat{\lambda}\lambda^{n-1}(\frac{\theta_0\hat{\lambda} + \hat{\rho} + \sqrt{\eta}\xi}{\hat{\lambda}} + \sum_{i=2}^{n} \frac{\rho + \sqrt{\eta}\xi}{\hat{\lambda}\lambda^{i-1}}) =$$

$$\theta_0\hat{\lambda}\lambda^{n-1} + (\hat{\rho} + \sqrt{\eta}\xi)\lambda^{n-1} + (\rho + \sqrt{\eta}\xi)\sum_{i=2}^{n} \lambda^{n-1-(i-1)} =$$

$$\theta_0\hat{\lambda}\lambda^{n-1} + (\hat{\rho} + \sqrt{\eta}\xi)\lambda^{n-1} + (\rho + \sqrt{\eta}\xi)\sum_{i=2}^{n} \lambda^{n-1-(i-1)} =$$

$$\theta_0\hat{\lambda}\lambda^{n-1} + (\hat{\rho} + \sqrt{\eta}\xi)\lambda^{n-1} + (\rho + \sqrt{\eta}\xi)\sum_{i=0}^{n-2} \lambda^{i} =$$

$$\theta_0\hat{\lambda}\lambda^{n-1} + \hat{\rho}\lambda^{n-1} + \rho\sum_{i=0}^{n-2} \lambda^{i} + \sqrt{\eta}\xi\sum_{i=0}^{n-1} \lambda^{i}.$$

Now by defining a new series - $\hat{\theta}_{(k+1)n}^1 = c_1\hat{\theta}_{kn}^1 + c_2$, and using the tools for solving first order non-homogeneous recurrence relations with constant coefficients, the value of $\hat{\theta}_{kn}^1$ can be found

$$\hat{\theta}_{kn}^1 = c_1^k(\frac{\hat{\theta}_n^1}{c_1} + \sum_{i=2}^{k} \frac{c_2}{c_1^i}) = \theta_n^1 c_1^{k-1} + \sum_{i=2}^{k} c_2 c_1^{k-i} =$$

$$\theta_n^1 c_1^{k-1} + c_2\sum_{i=0}^{k-2} c_1^i = (\theta_0 c_1 + c_2)c_1^{k-1} + c_2\sum_{i=0}^{k-2} c_1^i =$$

$$\theta_0(\hat{\lambda}\lambda^{n-1})^k + (\hat{\rho}\lambda^{n-1} + \rho\sum_{i=0}^{n-2} \lambda^{i} + \sqrt{\eta}\xi\sum_{i=0}^{n-1} \lambda^{i})\sum_{j=0}^{k-1}(\hat{\lambda}\lambda^{n-1})^j.$$

The proof for $\hat{\theta}_{kn}^r$ is done in similar manner. $\square$

**Corollary A.1.** $\hat{\theta}_{(k+1)n}^r \sim \mathbb{N}(\hat{\theta}_{(k+1)n}^r; \hat{\mu}_{(k+1)n}^r, (\hat{\sigma}_{(k+1)n}^r)^2)$.

**Lemma A.2.** $\mu_{kn+n} - \hat{\mu}_{kn+n}^r \geq \lambda^{n-1}\frac{ncx_h^2\beta}{\alpha+nx_h^2\beta}\lambda^{k(n-1)}(\hat{\lambda}^{k+1} - \lambda^{k+1})$.

*Proof Lemma A.2.* The proof of this lemma is separated into two cases, for $r = 1$ and for $r > 1$. For $r = 1$, it is easy to derive eq. 18 from lemma A.1, using $\mathbb{E}[\theta_0] = 0$ and $\mathbb{E}[\xi] = 0$.

$$\hat{\mu}_{(k+1)n}^1 = \rho\sum_{i=0}^{n-2} \lambda^{i}\sum_{j=0}^{k}(\hat{\lambda}\lambda^{(n-1)})^j + \hat{\rho}\lambda^{n-1}\sum_{j=0}^{k}(\hat{\lambda}\lambda^{(n-1)})^j$$

$$\mu_{kn+n} = \rho\sum_{i=0}^{n-2} \lambda^{i}\sum_{j=0}^{k}\lambda^{jn} + \rho\lambda^{n-1}\sum_{r=0}^{k}\lambda^{rn}$$

(18)

We use the sum of a geometric sequence to get

$$\hat{\mu}^1_{(k+1)n} =$$

$$\rho \sum_{i=0}^{n-2} \lambda^i \sum_{j=0}^{k} (\hat{\lambda}\lambda^{(n-1)})^j + \hat{\rho}\lambda^{n-1} \sum_{j=0}^{k} (\hat{\lambda}\lambda^{(n-1)})^j = (\rho(\frac{1-\lambda^{n-1}}{1-\lambda}) + \hat{\rho}\lambda^{n-1}) \frac{1 - (\hat{\lambda}\lambda^{n-1})^{k+1}}{1 - \hat{\lambda}\lambda^{n-1}} \cdot$$

Therefore the difference between the means can be lower bounded:

$$\mu_{kn+n} - \hat{\mu}^1_{kn+n} =$$

$$\frac{1-\lambda^{(k+1)n}}{1-\lambda^n}[\rho(\frac{1-\lambda^{n-1}}{1-\lambda}) + \rho\lambda^{n-1}] - \frac{1-(\hat{\lambda}\lambda^{(n-1)})^{k+1}}{1-\hat{\lambda}\lambda^{n-1}}[\rho(\frac{1-\lambda^{n-1}}{1-\lambda}) + \hat{\rho}\lambda^{n-1}] =^*$$

$$\frac{1-\lambda^{(k+1)n}}{1-\lambda^n} \frac{ncx^2\beta}{\alpha + nx_h^2\beta}(1-\lambda^n) - \frac{1-(\hat{\lambda}\lambda^{(n-1)})^{k+1}}{1-\hat{\lambda}\lambda^{n-1}} \frac{ncx^2\beta}{\alpha + nx_h^2\beta}(1 - \lambda^{n-1}(\frac{1}{4}\lambda + \frac{3}{4})) =$$

$$(1-\lambda^{(k+1)n}) \frac{ncx^2\beta}{\alpha + nx_h^2\beta} - \frac{1-(\hat{\lambda}\lambda^{(n-1)})^{k+1}}{1-\hat{\lambda}\lambda^{n-1}} \frac{ncx^2\beta}{\alpha + nx_h^2\beta}(1 - \lambda^{n-1}(\frac{1}{4}\lambda + \frac{3}{4})) =$$

$$\frac{ncx^2\beta}{\alpha + nx_h^2\beta}[(1-\lambda^{(k+1)n}) - \frac{1-(\hat{\lambda}\lambda^{(n-1)})^{k+1}}{1-\hat{\lambda}\lambda^{n-1}}(1 - \lambda^{n-1}(\frac{1}{4}\lambda + \frac{3}{4}))] =^{**}$$

$$\frac{ncx^2\beta}{\alpha + nx_h^2\beta}\Big(\frac{\lambda^{n-1}\frac{3}{4}\frac{\eta}{2}\alpha(1-\hat{\lambda}^{k+1}\lambda^{(k+1)(n-1)}) + \lambda^{(k+1)(n-1)}(\hat{\lambda}^{k+1} - \lambda^{k+1})(1-\lambda^{n-1}\hat{\lambda})}{1-\lambda^{n-1}\hat{\lambda}}\Big) \geq$$

$$\frac{ncx^2\beta}{\alpha + nx_h^2\beta}\Big(\frac{\lambda^{(k+1)(n-1)}(\hat{\lambda}^{k+1} - \lambda^{k+1})(1-\lambda^{n-1}\hat{\lambda})}{1-\lambda^{n-1}\hat{\lambda}}\Big) =$$

$$\frac{ncx^2\beta}{\alpha + nx_h^2\beta}\lambda^{(k+1)(n-1)}(\hat{\lambda}^{k+1} - \lambda^{k+1}) =$$

$$\lambda^{n-1} \frac{ncx^2\beta}{\alpha + nx_h^2\beta}\lambda^{k(n-1)}(\hat{\lambda}^{k+1} - \lambda^{k+1})$$

where equality * holds from claims D.1, D.2, D.3, equality ** holds from claim D.5, and the inequality holds because $\lambda < \hat{\lambda} < 1$. This proves Lemma A.2 for $r = 1$.

For the case of $r > 1$, from Lemma A.1 it is easy to see

$$\hat{\theta}^{r>1}_{(k+1)n} = [[\theta_0\lambda^{r-1} + \rho\sum_{i=0}^{r-2}\lambda^i + \sqrt{\eta}\sum_{i=0}^{r-2}\lambda^i\xi_i]\hat{\lambda}^k\lambda^{k(n-1)} +$$

$$\sum_{j=0}^{k-1}(\hat{\lambda}\lambda^{n-1})^j[\hat{\rho}\lambda^{n-1} + \rho\sum_{i=0}^{n-2}\lambda^i + \sqrt{\eta}\sum_{i=0}^{n-1}\lambda^i\xi_i]]\hat{\lambda}\lambda^{n-r} + \ .$$

$$\hat{\rho}\lambda^{n-r} + \rho\sum_{j=0}^{n-r-1}\lambda^j + \sqrt{\eta}\sum_{j=0}^{n-r}\xi\lambda^j$$

Therefore $\hat{\mu}^{r>1}_{kn+n}$ follows

$$\hat{\mu}^{r>1}_{(k+1)n} =$$

$$[[\rho\sum_{i=0}^{r-2}\lambda^i]\hat{\lambda}^k\lambda^{k(n-1)} + \sum_{j=0}^{k-1}(\hat{\lambda}\lambda^{n-1})^j[\hat{\rho}\lambda^{n-1} + \rho\sum_{i=0}^{n-2}\lambda^i]]\hat{\lambda}\lambda^{n-r} + \hat{\rho}\lambda^{n-r} + \rho\sum_{j=0}^{n-r-1}\lambda^j \cdot$$

Consequently the difference in means for $r > 1$ can be lower bounded:

$$\mu_{kn+n} - \hat{\mu}_{kn+n}^r =$$

$$\lambda^{n-r}[\lambda\rho\lambda^k\lambda^{k(n-1)}\sum_{i=0}^{r-2}\lambda^i + \lambda\sum_{j=0}^{k-1}(\lambda\lambda^{n-1})^j(\rho\lambda^{n-1} + \rho\sum_{i=0}^{n-2}\lambda^i) -$$

$$\hat{\lambda}\rho\hat{\lambda}^k\lambda^{k(n-1)}\sum_{i=0}^{r-2}\lambda^i - \hat{\lambda}\sum_{j=0}^{k-1}(\hat{\lambda}\lambda^{n-1})^j(\hat{\rho}\lambda^{n-1} + \rho\sum_{i=0}^{n-2}\lambda^i)] + \lambda^{n-r}(\rho - \hat{\rho}) =$$

$$\lambda^{n-r}\lambda^{k(n-1)}\rho(\lambda^{k+1} - \hat{\lambda}^{k+1})\sum_{i=0}^{r-2}\lambda^i + \lambda^{n-r}\sum_{j=0}^{k-1}\lambda^{(n-1)j}[\lambda^{n-1}(\rho\lambda^{j+1} -$$

$$\hat{\rho}\hat{\lambda}^{j+1}) + (\lambda^{j+1} - \hat{\lambda}^{j+1})\rho\sum_{i=0}^{n-2}\lambda^i] + \lambda^{n-r}(\rho - \hat{\rho}) =^*$$

$$\lambda^{n-r}\lambda^{k(n-1)}\rho(\lambda^{k+1} - \hat{\lambda}^{k+1})\frac{1 - \lambda^{r-1}}{1 - \lambda} + \lambda^{n-r}\frac{ncx_h^2\beta}{\alpha + nx_h^2\beta}[\lambda(1 - \lambda^{kn}) -$$

$$\hat{\lambda}\frac{1 - (\lambda^{n-1}\hat{\lambda})^k}{1 - \lambda^{n-1}\hat{\lambda}}(1 - \lambda^n(\frac{3}{4}\lambda^{-1} + \frac{1}{4}))] + \lambda^{n-r}(\rho - \hat{\rho}) =^{**}$$

$$\lambda^{n-r}\lambda^{k(n-1)}\rho(\lambda^{k+1} - \hat{\lambda}^{k+1})\frac{1 - \lambda^{r-1}}{1 - \lambda} + \lambda^{n-r}\frac{ncx_h^2\beta}{\alpha + nx_h^2\beta}[(\lambda - \hat{\lambda}) +$$

$$\frac{\lambda^{n-1}[\frac{3}{4}\frac{\eta}{2}\alpha(1 - \hat{\lambda}^k\lambda^{k(n-1)})]}{1 - \hat{\lambda}\lambda^{n-1}} + \lambda^{k(n-1)}(\hat{\lambda}^{k+1} - \lambda^{k+1})] + \lambda^{n-r}(\rho - \hat{\rho}) =^{***}$$

$$\lambda^{n-r}\frac{ncx_h^2\beta}{\alpha + nx_h^2\beta}\lambda^{k(n-1)}(\lambda^{k+1} - \hat{\lambda}^{k+1})(1 - \lambda^{r-1}) + \lambda^{n-r}\frac{ncx_h^2\beta}{\alpha + nx_h^2\beta}[(\lambda - \hat{\lambda})$$

$$+ \frac{\lambda^{n-1}[\frac{3}{4}\frac{\eta}{2}\alpha(1 - \hat{\lambda}^k\lambda^{k(n-1)})]}{1 - \hat{\lambda}\lambda^{n-1}} + \lambda^{k(n-1)}(\hat{\lambda}^{k+1} - \lambda^{k+1})] + \lambda^{n-r}(\rho - \hat{\rho}) =$$

$$\lambda^{n-r}\frac{ncx_h^2\beta}{\alpha + nx_h^2\beta}\lambda^{k(n-1)}(\lambda^{k+1} - \hat{\lambda}^{k+1})[1 - \lambda^{r-1} - 1] +$$

$$\lambda^{n-r}\frac{ncx_h^2\beta}{\alpha + nx_h^2\beta}(\lambda - \hat{\lambda} + \frac{\lambda^{n-1}[\frac{3}{4}\frac{\eta}{2}\alpha(1 - \hat{\lambda}^k\lambda^{k(n-1)})]}{1 - \hat{\lambda}\lambda^{n-1}}) + \lambda^{n-r}(\rho - \hat{\rho}) =$$

$$\lambda^{n-r}\frac{ncx_h^2\beta}{\alpha + nx_h^2\beta}\lambda^{k(n-1)}(\hat{\lambda}^{k+1} - \lambda^{k+1})\lambda^{r-1} +$$

$$\lambda^{n-r}\frac{ncx_h^2\beta}{\alpha + nx_h^2\beta}(\lambda - \hat{\lambda} + \frac{\lambda^{n-1}[\frac{3}{4}\frac{\eta}{2}\alpha(1 - \hat{\lambda}^k\lambda^{k(n-1)})]}{1 - \hat{\lambda}\lambda^{n-1}}) + \lambda^{n-r}(\rho - \hat{\rho}) =$$

$$\lambda^{n-1}\frac{ncx_h^2\beta}{\alpha + nx_h^2\beta}\lambda^{k(n-1)}(\hat{\lambda}^{k+1} - \lambda^{k+1}) +$$

$$\lambda^{n-r}\frac{ncx_h^2\beta}{\alpha + nx_h^2\beta}(\lambda - \hat{\lambda} + \frac{\lambda^{n-1}[\frac{3}{4}\frac{\eta}{2}\alpha(1 - \hat{\lambda}^k\lambda^{k(n-1)})]}{1 - \hat{\lambda}\lambda^{n-1}}) + \lambda^{n-r}(\rho - \hat{\rho}) >^{****}$$

$$\lambda^{n-1}\frac{ncx_h^2\beta}{\alpha + nx_h^2\beta}\lambda^{k(n-1)}(\hat{\lambda}^{k+1} - \lambda^{k+1})$$

where equality * holds from claims D.6 and D.7, equality ** holds from claim D.10, equality ***
holds from claim D.1, and equality **** holds from claim D.11 and $\hat{\lambda} > \lambda$. $\qquad\square$

**Lemma A.3.** *For $x_h^2\beta > 3, n > \frac{1}{2\alpha x_h^2\beta} - \frac{1}{x_h^2\beta}, \exists \dot{k} \in \mathbb{R}^+$ such that upper bounds defined in eq. 19 hold for all $0 < k \le \dot{k}$:*

$$
\begin{aligned}
(\sigma_{(k+1)n}^1)^2 &\le 2(\hat{\lambda}\lambda^{n-1})^2 \frac{1}{\alpha}(\hat{\lambda}\lambda^{n-1})^{2k} \\
(\sigma_{(k+1)n}^{r>1})^2 &\le 6(\hat{\lambda}\lambda^{n-r})^2 \frac{1}{\alpha}(\hat{\lambda}\lambda^{n-1})^{2k}.
\end{aligned}
\tag{19}
$$

*Proof Lemma A.3.* The proof will be separated into two cases, $r = 1$ and $r > 1$. $(\hat{\sigma}_{kn+n}^1)^2$ can be easily computed from lemma A.1 using the fact that both the noise and prior are distributed normally. A first general upper bound on $(\hat{\sigma}_{kn+n}^1)^2$ is found at eq. 20.

$$
\begin{aligned}
(\sigma_{kn+n}^1)^2 &= \frac{1}{\alpha}(\hat{\lambda}\lambda^{(n-1)})^{2(k+1)} + \eta\sum_{i=0}^{n-1}\lambda^{2i}\sum_{j=0}^{k}(\hat{\lambda}^2\lambda^{2(n-1)})^j = \\
&(\hat{\lambda}\lambda^{(n-1)})^{2(k+1)}\Big[\frac{1}{\alpha} + \eta\sum_{i=0}^{n-1}\lambda^{2i}\sum_{j=0}^{k}\frac{(\hat{\lambda}\lambda^{(n-1)})^{2j}}{(\hat{\lambda}\lambda^{(n-1)})^{2(k+1)}}\Big] = \\
&(\hat{\lambda}\lambda^{(n-1)})^{2(k+1)}\Big[\frac{1}{\alpha} + \eta\sum_{i=0}^{n-1}\lambda^{2i}\sum_{j=0}^{k}(\hat{\lambda}\lambda^{(n-1)})^{2(j-(k+1))}\Big] \le \\
&(\hat{\lambda}\lambda^{(n-1)})^{2(k+1)}\Big[\frac{1}{\alpha} + \eta\sum_{i=0}^{n-1}\lambda^{2i}\sum_{j=0}^{k}\lambda^{2n(j-(k+1))}\Big] = \\
&(\hat{\lambda}\lambda^{(n-1)})^{2(k+1)}\Big[\frac{1}{\alpha} + \eta\sum_{i=1}^{(k+1)n}\lambda^{-2i}\Big] = \\
&(\hat{\lambda}\lambda^{(n-1)})^{2(k+1)}\Big[\frac{1}{\alpha} + \eta\lambda^{-2}\frac{1-\lambda^{-2(k+1)n}}{1-\lambda^{-2}}\Big]
\end{aligned}
\tag{20}
$$

where the inequality holds because $\lambda < \hat{\lambda}$

By claim D.12, this upper bound can be further bounded for $\dot{k} \le \frac{1}{2n}\log_\lambda\big(\frac{1}{1+\frac{1}{\alpha\eta}(1-\lambda^2)}\big) - 1$ such that eq. 21 will hold, therefore proving the bound for $r = 1$.

$$
(\hat{\lambda}\lambda^{(n-1)})^{2(\dot{k}+1)}\Big[\frac{1}{\alpha} + \eta\lambda^{-2}\frac{1-\lambda^{-2(\dot{k}+1)n}}{1-\lambda^{-2}}\Big] \le 2(\hat{\lambda}\lambda^{(n-1)})^{2(\dot{k}+1)}\frac{1}{\alpha}
\tag{21}
$$

For $r > 1$, $(\hat{\sigma}_{kn+n}^{r>1})^2$ can be bounded as following

$$(\sigma_{kn+n}^{r>1})^2 =$$

$$(\hat{\lambda}\lambda^{n-r})^2\eta[(\hat{\lambda}^k\lambda^{k(n-1)})^2\sum_{i=0}^{r-2}\lambda^{2i} + \sum_{j=0}^{k-1}(\hat{\lambda}\lambda^{n-1})^{2j}\sum_{i=0}^{n-1}\lambda^{2i}]+$$

$$\eta\sum_{i=0}^{n-r}\lambda^{2i} + \frac{1}{\alpha}(\hat{\lambda}\lambda^{n-1})^{2k}(\hat{\lambda}\lambda^{n-1})^2 \leq^*$$

$$(\hat{\lambda}\lambda^{n-r})^2\eta[(\hat{\lambda}^k\lambda^{k(n-1)})^2\sum_{i=0}^{r-2}\lambda^{2i} + \sum_{j=0}^{k-1}(\hat{\lambda}\lambda^{n-1})^{2j}\sum_{i=0}^{n-1}\lambda^{2i}]+$$

$$\eta\sum_{i=0}^{n-r}\lambda^{2i} + \frac{1}{\alpha}(\hat{\lambda}\lambda^{n-1})^{2k}(\hat{\lambda}\lambda^{n-r})^2 =$$

$$(\hat{\lambda}\lambda^{n-r})^2[\frac{1}{\alpha}(\hat{\lambda}\lambda^{n-1})^{2k} + \eta(\hat{\lambda}\lambda^{n-1})^{2k}\sum_{i=0}^{r-2}\lambda^{2i} + \eta\sum_{j=0}^{k-1}(\hat{\lambda}\lambda^{n-1})^{2j}\sum_{i=0}^{n-1}\lambda^{2i}] + \eta\sum_{i=0}^{n-r}\lambda^{2i} \leq^{**}$$

$$(\hat{\lambda}\lambda^{n-r})^2[\frac{1}{\alpha}(\hat{\lambda}\lambda^{n-1})^{2k} + \eta(\hat{\lambda}\lambda^{n-1})^{2k}\sum_{i=0}^{n-1}\lambda^{2i} + \eta\sum_{j=0}^{k-1}(\hat{\lambda}\lambda^{n-1})^{2j}\sum_{i=0}^{n-1}\lambda^{2i}] + \eta\sum_{i=0}^{n-r}\lambda^{2i} \leq^{***}$$

$$2(\hat{\lambda}\lambda^{n-r})^2[\frac{1}{\alpha}(\hat{\lambda}\lambda^{n-1})^{2\dot{k}} + \eta(\hat{\lambda}\lambda^{n-1})^{2k}\sum_{i=0}^{n-1}\lambda^{2i} + \eta\sum_{j=0}^{k-1}(\hat{\lambda}\lambda^{n-1})^{2j}\sum_{i=0}^{n-1}\lambda^{2i}]$$

$$(22)$$

where inequality * follows from $\lambda < 1$ and $r > 1$, inequality ** follows from $r \leq n$, and inequality *** holds from claim D.15.

For $k \leq \frac{1}{2n}\log_\lambda(\frac{1}{1+\frac{1}{\alpha\eta}(1-\lambda^2)}) - 1$, this bound can be further developed

$$2(\hat{\lambda}\lambda^{n-r})^2[\frac{1}{\alpha}(\hat{\lambda}\lambda^{n-1})^{2\dot{k}} + \eta(\hat{\lambda}\lambda^{n-1})^{2k}\sum_{i=0}^{n-1}\lambda^{2i} + \eta\sum_{j=0}^{k-1}(\hat{\lambda}\lambda^{n-1})^{2j}\sum_{i=0}^{n-1}\lambda^{2i}] \leq$$
$$6(\hat{\lambda}\lambda^{n-r})^2\frac{1}{\alpha}(\hat{\lambda}\lambda^{n-1})^{2\dot{k}} \quad . \quad (23)$$

The inequality holds from claims D.12, D.14, which provide the bound for $r > 1$. All that is left is to prove that $\frac{1}{2n}\log_\lambda(\frac{1}{1+\frac{1}{\alpha\eta}(1-\lambda^2)}) - 1 > 0$, which is done in Claim D.22. □

**Lemma A.4.** *Mark* $\dot{k} = \frac{1}{2n}\log_\lambda(\frac{1}{1+\frac{1}{\alpha\eta}(1-\lambda^2)}) - 1$*, for the conditions of Lemma A.3*

$$(\sigma_{\lceil\dot{k}\rceil n+n}^1)^2 \leq (1 + 2e^{\frac{1}{x^2\beta}})(\hat{\lambda}\lambda^{n-1})^2\frac{1}{\alpha}(\hat{\lambda}\lambda^{(n-1)})^{2\lceil\dot{k}\rceil}$$

$$(\sigma_{\lceil\dot{k}\rceil n+n}^{r>1})^2 \leq 6(\hat{\lambda}\lambda^{n-r})^2\frac{1}{\alpha}(\hat{\lambda}\lambda^{n-1})^{2\lceil\dot{k}\rceil}.$$

*Proof Lemma A.4.* This proof will be separated into two cases, for $r > 1$ and for $r = 1$. For $r > 1$, the bound found in eq. 22, has no dependence on the choice of $k$, therefore holds also for $\lceil\dot{k}\rceil$. This bound was, in turn, developed for $\dot{k}$ at eq. 23 using three claims. If these claims also hold for $\lceil\dot{k}\rceil$, then the bound in eq. 23 also holds for $\lceil\dot{k}\rceil$, and the lemma is proved for $r > 1$.

Claims D.14, D.12 hold for all $k \leq \frac{1}{2n}\log_\lambda(\frac{1}{1+\frac{1}{\alpha\eta}(1-\lambda^2)})$, and since $\lceil\dot{k}\rceil \leq \dot{k} + 1 = \frac{1}{2n}\log_\lambda(\frac{1}{1+\frac{1}{\alpha\eta}(1-\lambda^2)})$, they holds for $\lceil\dot{k}\rceil$. Claim D.15 was proved for all k, hence also holds for $\lceil\dot{k}\rceil$.

For $r = 1$, the bound found at eq. 20 is applicable for all k, hence

$$(\sigma^1_{(\lceil \dot{k} \rceil+1)n})^2 \leq (\hat{\lambda}\lambda^{n-1})^{2(\lceil \dot{k} \rceil+1)}[\frac{1}{\alpha} + \eta\lambda^{-2}\frac{1-\lambda^{-2(\lceil \dot{k} \rceil+1)n}}{1-\lambda^{-2}}] \leq (\hat{\lambda}\lambda^{n-1})^{2(\lceil \dot{k} \rceil+1)}\frac{1}{\alpha}(1+2e^{\frac{1}{x^2\beta}})$$

where the last inequality holds from claim D.17. $\qquad \square$

**Lemma A.5.** *For $\dot{k}$ defined in Lemma A.4, the conditions of Lemma A.3, and $n > \max\{\frac{\alpha}{x_h^2\beta}, \frac{\alpha}{x_h^2\beta}(e^{\frac{2}{x_h^2\beta}} - 2) + \frac{1}{2x_h^2\beta}\}$*

$$\frac{(\mu_{\lceil \dot{k} \rceil n+n} - \hat{\mu}^r_{\lceil \dot{k} \rceil n+n})^2}{(\sigma^r_{\lceil \dot{k} \rceil n+n})^2} \geq e^{-\frac{2}{x_h^2\beta}}\frac{\alpha}{v_1}(\frac{3}{32x_h^2\beta})^2(\frac{c}{n})^2$$

$$v_1 = \max\{6, 1 + 2e^{\frac{1}{x_h^2\beta}}\}.$$

*Proof Lemma A.5.*

$$\frac{(\mu_{\lceil \dot{k} \rceil n+n} - \hat{\mu}^r_{\lceil \dot{k} \rceil n+n})^2}{\sigma^r_{\lceil \dot{k} \rceil n+n}} \geq \frac{\left(\lambda^{n-1}\frac{ncx_h^2\beta}{\alpha+nx_h^2\beta}\lambda^{\lceil \dot{k} \rceil(n-1)}(\hat{\lambda}^{\lceil \dot{k} \rceil+1} - \lambda^{\lceil \dot{k} \rceil+1})\right)^2}{v_1(\hat{\lambda}\lambda^{n-r})^2\frac{1}{\alpha}(\hat{\lambda}\lambda^{n-1})^{2\lceil \dot{k} \rceil}} =$$

$$\frac{\lambda^{2\lceil \dot{k} \rceil(n-1)}\left(\lambda^{n-1}\frac{ncx_h^2\beta}{\alpha+nx_h^2\beta}(\hat{\lambda}^{\lceil \dot{k} \rceil+1} - \lambda^{\lceil \dot{k} \rceil+1})\right)^2}{v_1(\hat{\lambda}\lambda^{n-r})^2\frac{1}{\alpha}(\hat{\lambda}\lambda^{n-1})^{2\lceil \dot{k} \rceil}} = \frac{\left(\lambda^{n-1}\frac{ncx_h^2\beta}{\alpha+nx_h^2\beta}(\hat{\lambda}^{\lceil \dot{k} \rceil+1} - \lambda^{\lceil \dot{k} \rceil+1})\right)^2}{v_1(\hat{\lambda}\lambda^{n-r})^2\frac{1}{\alpha}\hat{\lambda}^{2\lceil \dot{k} \rceil}} =$$

$$\frac{\alpha\lambda^{2(r-1)}}{v_1}(\frac{ncx_h^2\beta}{\alpha+nx_h^2\beta})^2\frac{(\hat{\lambda}^{\lceil \dot{k} \rceil+1} - \lambda^{\lceil \dot{k} \rceil+1})^2}{\hat{\lambda}^{2\lceil \dot{k} \rceil+1}} = \frac{\alpha\lambda^{2(r-1)}}{v_1}(\frac{ncx_h^2\beta}{\alpha+nx_h^2\beta})^2(1 - \frac{\lambda^{\lceil \dot{k} \rceil+1}}{\hat{\lambda}^{\lceil \dot{k} \rceil+1}})^2 \geq$$

$$\frac{\alpha\lambda^{2(r-1)}}{v_1}(\frac{ncx_h^2\beta}{\alpha+nx_h^2\beta})^2(1 - (1 - \frac{\frac{3}{4}nx^2\beta}{(\alpha+nx_h^2\beta)^2 - (\alpha+\frac{1}{4}nx^2\beta)}))^2 =$$

$$\frac{\alpha\lambda^{2(r-1)}}{v_1}(\frac{ncx_h^2\beta}{\alpha+nx_h^2\beta})^2(\frac{\frac{3}{4}nx_h^2\beta}{(\alpha+nx_h^2\beta)^2 - (\alpha+\frac{1}{4}nx^2\beta)})^2 \geq$$

$$\frac{\alpha\lambda^{2(r-1)}}{v_1}(\frac{ncx_h^2\beta}{\alpha+nx_h^2\beta})^2(\frac{\frac{3}{4}nx^2\beta}{(\alpha+nx_h^2\beta)^2})^2 \geq \frac{\alpha\lambda^{2(r-1)}}{v_1}(\frac{ncx^2\beta}{2nx_h^2\beta})^2(\frac{\frac{3}{4}nx^2\beta}{(2nx^2\beta)^2})^2 =$$

$$\frac{\alpha\lambda^{2(r-1)}}{v_1}(\frac{c}{2})^2(\frac{\frac{3}{4}}{4nx_h^2\beta})^2 = \frac{\alpha\lambda^{2(r-1)}}{v_1}(\frac{3}{32x_h^2\beta})^2(\frac{c}{n})^2 \geq$$

$$\frac{\alpha\lambda^{2(n-1)}}{v_1}(\frac{3}{32x_h^2\beta})^2(\frac{c}{n})^2 \geq e^{-\frac{2}{x_h^2\beta}}\frac{\alpha}{v_1}(\frac{3}{32x_h^2\beta})^2(\frac{c}{n})^2$$

where first inequality holds from A.3, A.4 and the definition of $v_1$, the second inequality follows claim D.17 and claim D.22, fourth inequality holds under the assumption of $nx_h^2\beta > \alpha \iff n > \frac{\alpha}{x_h^2\beta}$, and last inequality holds from claim D.19. $\qquad \square$

**Lemma A.6.** *For the Bayesian linear regression problem over database $D_1$, the conditions of Lemma A.5, and $\dot{k}$, as defined in Lemma A.4, approximate sampling, by running SGLD for $(\lceil \dot{k} \rceil+1)n$ steps, will not be $(\epsilon, \delta)$ differentially private for*

$$\delta < 0.5, \epsilon < e^{-\frac{2}{x^2\beta}}\frac{\alpha}{2v_1}(\frac{3}{32x_h^2\beta})^2(\frac{c}{n})^2 + \ln(0.5 - \delta)$$

$$v_1 = \max\{6, 1 + 2e^{\frac{1}{x_h^2\beta}}\}.$$

*Proof Lemma A.6.* According to definition 1, it is enough that there is one group, $S$, such that $p(\theta_{(\lceil \dot{k} \rceil+1)n} \in S|D_1) > e^\epsilon p(\hat{\theta}_{(\lceil \dot{k} \rceil+1)n} \in S|D_2) + \delta$, to show that releasing $\theta_{(\lceil \dot{k} \rceil+1)n}$ is not $(\epsilon, \delta)$ private. Consider $S = \{s|s > \mu_{(\lceil \dot{k} \rceil+1)n}\}$. From claim D.23 and since $\theta_{(\lceil \dot{k} \rceil+1)n} \sim \mathbb{N}(\theta_{(\lceil \dot{k} \rceil+1)n}; \mu_{(\lceil \dot{k} \rceil+1)n}, \sigma^2_{(\lceil \dot{k} \rceil+1)n})$, eq. 24 holds. The conditions for the right term to be smaller

than 0 (thus making the approximate sampling not $(\epsilon, \delta)$ private) are found in eq. 25, therefore proving the lemma.

$$e^\epsilon p(\hat{\theta}_{(\lceil k \rceil + 1)n} \in S | D_2) + \delta - p(\hat{\theta}_{(\lceil k \rceil + 1)n} \in S | D_1) \leq e^\epsilon e^{-e^{-\frac{2}{x^2\beta}} \frac{\alpha}{2v_1} (\frac{3}{32x_h^2\beta})^2 (\frac{c}{n})^2} + \delta - 0.5 \quad (24)$$

$$
\begin{aligned}
e^{\epsilon - e^{-\frac{2}{x^2\beta}} \frac{\alpha}{2v_1} (\frac{3}{32x_h^2\beta})^2 (\frac{c}{n})^2} + \delta - 0.5 &< 0 \\
e^{\epsilon - e^{-\frac{2}{x^2\beta}} \frac{\alpha}{2v_1} (\frac{3}{32x_h^2\beta})^2 (\frac{c}{n})^2} &< 0.5 - \delta \\
\epsilon - e^{-\frac{2}{x^2\beta}} \frac{\alpha}{2v_1} (\frac{3}{32x_h^2\beta})^2 (\frac{c}{n})^2 &< \ln(0.5 - \delta) \\
\epsilon &< e^{-\frac{2}{x^2\beta}} \frac{\alpha}{2v_1} (\frac{3}{32x_h^2\beta})^2 (\frac{c}{n})^2 + \ln(0.5 - \delta)
\end{aligned}
\quad (25)
$$

$\square$

## B    PROPOSE TEST SAMPLE SUPPLEMENTARY

$n_{min}$ which is used in algorithm 4.3 is defined as following

$$
\begin{aligned}
\nu &= \frac{2\ln(\frac{1}{\delta})}{\epsilon} + 1 \\
n_{b1} &= \max\{1 + \frac{x_h^2}{x_l^2} \frac{8}{\epsilon}, 1 + \nu \frac{x_h^2}{x_l^2}(1 + 8\frac{(\nu - 1)}{\epsilon}), (\frac{16\nu\beta x_h^4}{\frac{9}{10}\epsilon x_l^2})^{\frac{1}{1-2\gamma_1}}, \\
&\quad (\frac{32\nu\beta}{\epsilon}(\frac{(x_h^2\beta)(x_h^2\alpha + x_h^4\beta)}{\frac{9}{10}(x_l^2\beta)^2})\check{m})^{\frac{1}{2-\gamma_1}}, \\
&\quad (\frac{32\nu\beta}{\epsilon}(\frac{(x_h^2\beta)(x_h^2\alpha + x_h^4\beta)}{\frac{9}{10}(x_l^2\beta)^2}))^{\frac{1}{2-2\gamma_1}}, \\
&\quad (\frac{8\nu}{\epsilon}(\frac{(x_h^2\alpha + x_h^4\beta)^2}{\frac{9}{10}x_l^6\beta})\check{m})^{\frac{2}{3}}, \\
&\quad (\frac{8\nu}{\epsilon}(\frac{(x_h^2\alpha + x_h^4\beta)^2}{\frac{9}{10}x_l^6\beta}))^{\frac{2}{3-2\gamma_1}}\} \\
n_{b2} &= \max\{1 + \frac{x_h^2}{x_l^2} \frac{10\nu}{\beta}, 1 + \nu \frac{x_h^2}{x_l^2}\} \\
n_{min} &= \max\{n_{b1}, n_{b2}, n_1^{\frac{\rho_2}{\gamma_1}}\}
\end{aligned}
\quad (26)
$$

## C    PROPOSE TEST SAMPLE PRIVACY

*Proof Claim 4.2.* We set the algorithm parameters in eq. 27, and matching databases $D_3, D_4$ defined in eq. 8. We note that we only define a lower bound over $n_1$, which will be updated later on.

$$
\begin{aligned}
&\rho_3 = 1.15; \rho_2 = 0.45; \rho_1 = 1.25, \gamma_1 = 0.49; \\
&x_l = x_h/2 \\
&n_1 > \max\{2\frac{1}{\epsilon}\log\frac{1}{2\delta}, 4\frac{1}{\epsilon}\log\frac{1}{2\delta}, 2^{10\rho_1}, 2^{9+10\rho_2}\} \\
&\beta = 3; x_h = 1 \\
&\alpha = 1
\end{aligned}
\quad (27)
$$

Mark the return value of the algorithm as $r$, the event of the algorithm running on database $D_3$ and $W = D_3$ as $A_{D_3}$, the event of the algorithm running on database $D_4$ and $W = D_4$ as $A_{D_4}$, and

$S = \{s | s > \mu_i\}$, where $\mu_i$ is the mean of the sample distribution at the SGLD $i$'th step given database $D_3$ (Similarly to as defined in subsection 4.2). We will show that $\forall \epsilon \in \mathbb{R}_{>0}, \delta < \frac{1}{6}, \exists n_1$ such that eq. 28 holds.

$$P(r \in S | D_3) > e^\epsilon P(r \in S | D_4) + \delta \tag{28}$$

We first show that

$$
\begin{aligned}
P(r \in S \wedge A^c_{D_3} | D_3) = 0 \\
P(r \in S \wedge A^c_{D_4} | D_4) = 0
\end{aligned}
\tag{29}
$$

Notice that the algorithm can return result in $S$ only if it reached step 19. Consider an event where the algorithm reached step 19 and $A^c_{D_3}$. From $A^c_{D_3}$, $\exists (x_i, y_i) \in D_3$ such that $|\frac{y_i}{x_i} - \check{m}| \geq n_2^{\rho_2}$. However, since $\forall (x_i, y_i) \in D_3 : \frac{y_i}{x_i} = n_1^{\rho_3}$ then $\forall (x_i, y_i) \in D_3 : |\frac{y_i}{x_i} - \check{m}| > n_2^{\rho_2}$ and therefore $|W| = 0$. Under the assumption that sample from $p(\theta | \{\})$ returns $null$ then in this case the algorithm also returns $null$ and therefore $P(r \in S \wedge A^c_{D_3} | D_3) = 0$. Same arguments hold for $D_4$.

Following eq. 29, to prove eq. 28 it is enough to prove eq. 30.

$$
\begin{aligned}
P(r \in S | D_3, A_3) P(A_3 | D_3) \geq^* \\
P(r \in S | D_3, A_3) - 5\delta >^{**} e^\epsilon P(r \in S | D_4, A_4) + \delta \geq \\
e^\epsilon P(r \in S | D_4, A_4) P(A_4 | D_4) + \delta = e^\epsilon P(r \in S \wedge A_4 | D_4) + \delta
\end{aligned}
\tag{30}
$$

From claim C.1 $\exists n_{bound_1}$ such that $\forall n_1 > n_{bound_1}$ inequality * holds. From Lemma 4.6, for $n_1$ big enough $\exists T \in \mathbb{Z}_{>0}$ such that eq. 31 hold (Where $6\delta < 0.5$ according to the claim conditions). Therefore, $\exists k, n_{bound2} \in \mathbb{R}_{>0}$ such that $\forall n_1 > n_{bound2} : \epsilon' > k n_1^{2(1-\rho_3)}$ and eq. 31 hold. As $\rho_3 > 1$, by choosing $n_1 > \max\{n_{bound2}, (\frac{\epsilon}{k})^{\frac{1}{2(\rho_3-1)}}\}$ get that $\epsilon' > \epsilon$. Consequently, by choosing $n_1 > \max\{n_{bound_1}, n_{bound_2}, (\frac{\epsilon}{k})^{\frac{1}{2(\rho_3-1)}}\}$, inequalities * and ** hold, and the claim is proved.

$$
\begin{aligned}
\epsilon' = \Omega(n_1^{2(\rho_3 - 1)}) \\
P(r \in S | D_3, A_3) > e^{\epsilon'} P(r \in S | D_4, A_4) + 6\delta
\end{aligned}
\tag{31}
$$

$\square$

**Claim C.1.** $\exists n_{bound_1} \in \mathbb{Z}_{>0}$ *such that the probability for algorithm 4.3 to reach step 19 with* $W = D_3$ *(marked event A) is greater or equal to* $1 - 5\delta$ *for all* $n_1 > n_{bound_1}$.

*Proof.* Mark the event of $n_W > n_{min} \wedge \check{m} \in [m - n_2^{\rho_2}, m + n_2^{\rho_2}] \wedge n_2^{1+\rho_2-0.1} > \check{n}_1^{\rho_1} \wedge \check{n}_1 \leq n_1 \wedge V = D$ as $B$. Since $P(A | D_3, B) = 1$ it follows that $P(A | D_3) \geq P(A \wedge B | D_3) = P(A | B, D_3) P(B | D_3) = P(B | D_3)$. Therefore if $\exists n_{lb}$ such that $\forall n_1 > n_{lb} : P(B | D_3) \geq 1 - 5\delta$ the claim is proved.

$$
\begin{aligned}
&P(B | D_3) = \\
&P(\check{m} \in [m - n_2^{\rho_2}, m + n_2^{\rho_2}] \wedge n_W > n_{min} | D_3, V = D, n_2^{1+\rho_2-0.1} > \check{n}_1^{\rho_1}, \check{n}_1 \leq n_1) \cdot \\
&P(n_2^{1+\rho_2-0.1} > \check{n}_1^{\rho_1} | V = D, \check{n}_1 \leq n_1, D_3) P(V = D, \check{n}_1 \leq n_1 | D_3) \geq \\
&P(\check{m} \in [m - n_2^{\rho_2}, m + n_2^{\rho_2}] \wedge n_W > n_{min} | D_3, V = D, n_2^{1+\rho_2-0.1} > \check{n}_1^{\rho_1}, \check{n}_1 \leq n_1) - 3\delta = \\
&P(n_W > n_{min} | D_3, V = D, n_2^{1+\rho_2-0.1} > \check{n}_1^{\rho_1}, \check{n}_1 \leq n_1, \check{m} \in [m - n_2^{\rho_2}, m + n_2^{\rho_2}]) \\
&P(\check{m} \in [m - n_2^{\rho_2}, m + n_2^{\rho_2}] | D_3, V = D, n_2^{1+\rho_2-0.1} > \check{n}_1^{\rho_1}, \check{n}_1 \leq n_1) - 3\delta \geq \\
&P(n_W > n_{min} | D_3, V = D, n_2^{1+\rho_2-0.1} > \check{n}_1^{\rho_1}, \check{n}_1 \leq n_1, \check{m} \in [m - n_2^{\rho_2}, m + n_2^{\rho_2}]) - 4\delta \geq \\
&1 - 5\delta
\end{aligned}
\tag{32}
$$

By corollary C.1 and claim C.3 for $n_1$ big enough first inequality holds. By claim C.4 for $n_1$ big enough second inequality holds, and by claim C.5 for $n_1$ big enough third inequality holds. Therefore for $n_1$ big enough eq. 32 holds and the claim is proved. $\qquad\square$

**Claim C.2.** *For* $n_1 > \max\{\frac{1}{2}^{-10\rho_1}, 4\frac{1}{\epsilon}\log\frac{1}{2\delta}\}$

$$P(\check{n}_1^{\rho_1} \geq n_1^{\rho_3} \wedge \check{n}_1 \leq n_1 | D_3) \geq 1 - 2\delta.$$

*Proof.* Mark the noise added at step 6 as $l_1$

$$P(\check{n}_1^{\rho_1} \geq n_1^{\rho_3} \wedge \check{n}_1 \leq n_1 | D_3) =$$

$$P((n_1 + l_1 - \frac{1}{\epsilon}\log\frac{1}{2\delta})^{\rho_1} \geq n_1^{\rho_3} \wedge n_1 + l_1 - \frac{1}{\epsilon}\log\frac{1}{2\delta} \leq n_1 | D_3) =$$

$$P((n_1 + l_1 - \frac{1}{\epsilon}\log\frac{1}{2\delta})^{\rho_1} \geq n_1^{\rho_3} \wedge l_1 \leq \frac{1}{\epsilon}\log\frac{1}{2\delta} | D_3) =$$

$$P((n_1 + l_1 - \frac{1}{\epsilon}\log\frac{1}{2\delta})^{\rho_1} \geq n_1^{\rho_3} \wedge |l_1| \leq \frac{1}{\epsilon}\log\frac{1}{2\delta} | D_3) +$$

$$P((n_1 + l_1 - \frac{1}{\epsilon}\log\frac{1}{2\delta})^{\rho_1} \geq n_1^{\rho_3} \wedge l_1 \leq -\frac{1}{\epsilon}\log\frac{1}{2\delta} | D_3) \geq$$

$$P((n_1 + l_1 - \frac{1}{\epsilon}\log\frac{1}{2\delta})^{\rho_1} \geq n_1^{\rho_3} \wedge |l_1| \leq \frac{1}{\epsilon}\log\frac{1}{2\delta} | D_3) =$$

$$P((n_1 + l_1 - \frac{1}{\epsilon}\log\frac{1}{2\delta})^{\rho_1} \geq n_1^{\rho_3} | |l_1| \leq \frac{1}{\epsilon}\log\frac{1}{2\delta}, D_3) P(|l_1| \leq \frac{1}{\epsilon}\log\frac{1}{2\delta} | D_3) \geq$$

$$P((n_1 - 2\frac{1}{\epsilon}\log\frac{1}{2\delta})^{\rho_1} \geq n_1^{\rho_3} | D_3) - 2\delta = 1 - 2\delta$$

Where last inequality holds from following equation

$$P(|l_1| \leq \frac{1}{\epsilon}\log\frac{1}{2\delta}) = 1 - 2P(l_1 \leq -\frac{1}{\epsilon}\log\frac{1}{2\delta}) = 1 - \exp(-\frac{\frac{1}{\epsilon}\log\frac{1}{2\delta}}{\frac{1}{\epsilon}}) = 1 - 2\delta$$

Last equality holds since $n_1 > \frac{1}{2}^{-10\rho_1} \Rightarrow n_1^{-1} < (\frac{1}{2})^{10\rho_1} \Rightarrow n_1^{-0.1} < (\frac{1}{2})^{\rho_1}$ and therefore $(n_1 - 2\frac{1}{\epsilon}\log\frac{1}{2\delta})^{\rho_1} > (\frac{1}{2}n_1)^{\rho_1} > n_1^{\rho_1 - 0.1} = n_1^{\rho_3}$ $\qquad\square$

**Corollary C.1.**

$$\forall n_1 > \max\{\frac{1}{2}^{-10\rho_1}, 4\frac{1}{\epsilon}\log\frac{1}{2\delta}\} : P(V = D_3 \wedge \check{n}_1 \leq n_1) \geq 1 - 2\delta.$$

**Claim C.3.** *For* $n_1 > \max\{\frac{1}{2}^{-(9+10\rho_2)}, 4\frac{1}{\epsilon}\log\frac{1}{2\delta}\}$

$$P(n_2^{1+\rho_2-0.1} > \check{n}_1^{\rho_1} | D_3, V = D_3, \check{n}_1 \leq n_1) \geq 1 - \delta.$$

*Proof.*

$$P(n_2^{0.9+\rho_2} > \check{n}_1^{\rho_1} | D_3, V = D_3, \check{n}_1 \leq n_1) \geq P(n_2^{0.9+\rho_2} > n_1^{\rho_1} | D_3, V = D_3) \geq$$

$$P((n_1 - 2\frac{1}{\epsilon}\log\frac{1}{2\delta})^{0.9+\rho_2} > n_1^{\rho_1} | D_3) p(n_2 \geq |V| - 2\frac{1}{\epsilon}\log\frac{1}{2\delta}) \geq$$

$$P((n_1 - 2\frac{1}{\epsilon}\log\frac{1}{2\delta})^{0.9+\rho_2} > n_1^{\rho_1} | D_3) - \delta = 1 - \delta$$

where the second inequality holds since $P(\text{Lap}(\frac{1}{\epsilon}) < -\frac{1}{\epsilon}\log\frac{1}{2\delta}) < \delta$, and last equality holds since $(n_1 - 2\frac{1}{\epsilon}\log\frac{1}{2\delta})^{0.9+\rho_2} > (\frac{1}{2}n_1)^{0.9+\rho_2} > n_1^{1+\rho_2-0.2} = n_1^{\rho_1}$ $\qquad\square$

**Claim C.4.** $\exists n_{lb_1} \in \mathbb{Z}_{>0}$ *such that*

$$\forall n_1 > n_{lb_1} : P(\check{m} \in [m - n_2^{\rho_2}, m + n_2^{\rho_2}] | D_3, n_2^{0.9+\rho_2} > \check{n}_1^{\rho_1}) \geq 1 - \delta.$$

*Proof.* Mark the noise added at step 13 as $l_1$

$$P(\breve{m} \in [m - n_2^{\rho_2}, m + n_2^{\rho_2}] | D_3, n_2^{0.9+\rho_2} > \breve{n}_1^{\rho_1}) =$$
$$P(l_1 \in [-n_2^{\rho_2}, n_2^{\rho_2}] | D_3, n_2^{0.9+\rho_2} > \breve{n}_1^{\rho_1}) \geq$$
$$1 - 2(\frac{1}{2}\exp(-n_2^{\rho_2} \frac{1}{\frac{1}{\epsilon}\breve{n}_1^{\rho_1} \frac{2(n_2-1)x_h^2 x_l^2 + x_h^4}{n_2(n_2-1)x_l^4}})) =$$
$$1 - \exp(-\frac{n_2^{1+\rho_2}\epsilon(n_2-1)x_l^4}{\breve{n}_1^{\rho_1}(2(n_2-1)x_h^2 x_l^2 + x_h^4)})$$

Since $n_2^{0.9+\rho_2} > \breve{n}_1^{\rho_1}$ then $\breve{n}_1^{\rho_1} = o(n_2^{1+\rho_2})$ and therefore for $n_1$ big enough the exponent is smaller than $\delta$. $\qquad \square$

**Claim C.5.** $\exists n_{lb_2} \in \mathbb{Z}_{>0}$ *such that*

$$\forall n_1 > n_{lb_2} : p(n_W > n_{min} | |V| = D_3 \wedge n_2^{0.9+\rho_2} > \breve{n}_1^{\rho_1} \wedge \breve{m} \in [m - n_2^{\rho_2}, m + n_2^{\rho_2}], D_3) \geq 1 - \delta.$$

*Proof.* For abbreviation mark event $B$ as $B = |V| = D_3 \wedge n_2^{0.9+\rho_2} > \breve{n}_1^{\rho_1} \wedge \breve{m} \in [m - n_2^{\rho_2}, m + n_2^{\rho_2}]$. Mark the Laplace noise used in step 8 as $l_1$ and the Laplace noise used in step 15 as $l_2$.

$$P(n_W > n_{min} | B, D_3) =$$
$$P(n_1 - \frac{1}{\epsilon}\log\frac{1}{2\delta} + l_2 > n_{min} | B, D_3) >$$
$$P(n_1 - \frac{1}{\epsilon}\log\frac{1}{2\delta} - \frac{1}{\epsilon}\log\frac{1}{\delta} > n_{min} | B, D_3)P(l_2 > -\frac{1}{\epsilon}\log\frac{1}{\delta}) +$$
$$P(n_1 - \frac{1}{\epsilon}\log\frac{1}{2\delta} + l_2 > n_{min} \wedge l_2 < -\frac{1}{\epsilon}\log\frac{1}{\delta} | B, D_3) >$$
$$P(n_1 - \frac{2}{\epsilon}\log\frac{1}{2\delta} - \frac{1}{\epsilon}\log\frac{1}{\delta} > n_{min} | B, D_3)(1 - \frac{\delta}{2}) >$$
$$P(n_1 - \frac{2}{\epsilon}\log\frac{1}{2\delta} - \frac{1}{\epsilon}\log\frac{1}{\delta} > n_{min} | B, D_3) - \frac{\delta}{2} =$$
$$P(n_1 - \frac{2}{\epsilon}\log\frac{1}{2\delta} - \frac{1}{\epsilon}\log\frac{1}{\delta} > n_{min} | l_1 < \frac{1}{\epsilon}\log\frac{1}{\delta}, B, D_3)P(l_1 < \frac{1}{\epsilon}\log\frac{1}{\delta} | B, D_3) +$$
$$P(n_1 - \frac{2}{\epsilon}\log\frac{1}{2\delta} - \frac{1}{\epsilon}\log\frac{1}{\delta} > n_{min} \wedge l_1 \geq \frac{1}{\epsilon}\log\frac{1}{\delta} | B, D_3) - \frac{\delta}{2} \geq$$
$$P(n_1 - \frac{2}{\epsilon}\log\frac{1}{2\delta} - \frac{1}{\epsilon}\log\frac{1}{\delta} > n_{min} | l_1 < \frac{1}{\epsilon}\log\frac{1}{\delta}, B, D_3)P(l_1 < \frac{1}{\epsilon}\log\frac{1}{\delta} | B, D_3) - \frac{\delta}{2} \geq$$
$$P(n_1 - \frac{2}{\epsilon}\log\frac{1}{2\delta} - \frac{1}{\epsilon}\log\frac{1}{\delta} > n_{min} | l_1 < \frac{1}{\epsilon}\log\frac{1}{\delta}, B, D_3) - \delta$$

From $B$ it holds that $|m - \breve{m}| < n_2^{\rho_2}$ and therefore $\breve{m} < m + n_2^{\rho_2}$, and for the case of $l_1 < \frac{1}{\epsilon}\log\frac{1}{\delta}$ it holds that $n_2 < n_1 - \frac{1}{\epsilon}\log\frac{1}{2\delta} + \frac{1}{\epsilon}\log\frac{1}{\delta} < n_1 + \frac{1}{\epsilon}\log\frac{1}{\delta}$. Therefore $\breve{m} \leq m + (n_1 + \frac{1}{\epsilon}\log\frac{1}{\delta})^{\rho_2}$. As $n_{min} = O(\max\{\breve{m}^{\frac{2}{3}}, n_1^{\frac{\rho_2}{\gamma_1}}\})$ then for the case of $l_1 < \frac{1}{\epsilon}\log\frac{1}{\delta}$ and $B$, it holds that $n_{min} = O(\max\{(m + n_1^{\rho_2})^{\frac{2}{3}}, n_1^{\frac{\rho_2}{\gamma_1}}\}) = O(\max\{n_1^{\frac{2\rho_3}{3}}, n_1^{\frac{\rho_2}{\gamma_1}}\}) < o(n_1)$, therefore $\exists n_{lb_2}$ such that $\forall n_1 > n_{lb_2} : n_1 - \frac{2}{\epsilon}\log\frac{1}{2\delta} - \frac{1}{\epsilon}\log\frac{1}{\delta} > n_{min}$. Consequently, $\forall n_1 > n_{lb_2} : P(n_1 - \frac{2}{\epsilon}\log\frac{1}{2\delta} - \frac{1}{\epsilon}\log\frac{1}{\delta} > n_{min} | l_1 < \frac{1}{\epsilon}\log\frac{1}{\delta}, B, D_3) = 1$.

$\qquad \square$

**Definition 4.** *A randomized function $f(X, y) : \chi^{n_1} \times \mathbb{R}^{n_2} \to \mathbb{R}$, is $(\epsilon, \delta)$-differentially private with respect to $X$ if $\forall S \subseteq \mathbb{R}$, and $\forall X, \hat{X} \in \chi^n : \|X - \hat{X}\| \leq 1$, eq. 33 holds.*

$$P(f(X, y) \in S) \leq \exp(\epsilon)P(f(\hat{X}, y) \in S) + \delta \qquad (33)$$

**Claim C.6.** *Calculating $\check{n}_1, n_2$ is $(2\epsilon, 0)$ differentially private.*

*Proof.* Since $n_1$ can differ by up to 1 for neighbouring databases, calculating $\check{n}_1$ is protected via the Laplace mechanism. Since for a given $\check{n}_1$ the value $|V|$ can change by up to 1 for two neighbouring databases then calculating $n_2$ is $(\epsilon, 0)$ by the Laplace mechanism. Consequently from sequential composition theorem the sequential composition is $(2\epsilon, 0)$ differentially private. $\qquad\square$

**Claim C.7.** $P(n_2 \leq |V| \| D, \check{n}_1) = 1 - \delta.$

*Proof.* Mark $l \sim \mathrm{Lap}(\frac{1}{\epsilon})$,

$$P(n_2 \leq |V| \| D, \check{n}_1) = P(|V| - \frac{1}{\epsilon} \log \frac{1}{2\delta} + l \leq |V| \| D, \check{n}_1) =$$

$$P(l \leq \frac{1}{\epsilon} \log \frac{1}{2\delta} | D, \check{n}_1) = 1 - \frac{1}{2} \exp(-\frac{\frac{1}{\epsilon} \log \frac{1}{2\delta}}{\frac{1}{\epsilon}}) = 1 - \delta$$

$\qquad\square$

**Claim C.8.** *Calculating $\check{m}$ is $(\epsilon, 0)$ differentially private with respect to $D$ for given $\check{n}_1, n_2$ and $n_2 < |V|$.*

*Proof.* Mark by $\hat{D}$ a neighbouring database to $D$, and $\hat{V}$ as $V$ induced by this database. If $V = \hat{V}$ then the claim follows trivially. In case the $V$'s differ, assume w.l.o.g that $|V| \geq |\hat{V}|$, and that if $|V| = |\hat{V}|$ then they differ in their last sample. Define $q = \sum_{(x_i,y_i)\in V/\{x_{|V|},y_{|V|}\}} x_i y_i$, $z = \sum_{(x_i,y_i)\in V/\{x_{|V|},y_{|V|}\}} x_i^2$.

$$\left| \frac{q + x_{|V|} y_{|V|}}{z + x_{|V|}^2} - \frac{q + \hat{x}_{|V|} \hat{y}_{|V|}}{z + \hat{x}_{|V|}^2} \right| =$$

$$\left| \frac{q\hat{x}_{|V|}^2 + x_{|V|} y_{|V|} \hat{x}_{|V|}^2 + x_{|V|} y_{|V|} z - q x_{|V|}^2 - \hat{x}_{|V|} \hat{y}_{|V|} x_{|V|}^2 - \hat{x}_{|V|} \hat{y}_{|V|} z}{(z + x_{|V|}^2)(z + \hat{x}_{|V|}^2)} \right| \leq$$

$$\frac{q x_h^2 + \check{n}_1^{\rho_1} x_h^2 z + \check{n}_1^{\rho_1} x_h^4}{(z + x_l^2)z} \leq \check{n}_1^{\rho_1} \frac{2z x_h^2 + x_h^4}{(z + x_l^2)z} = \check{n}_1^{\rho_1} \left( \frac{2 x_h^2}{z + x_l} + \frac{x_h^4}{(z + x_l^2)z} \right) \leq$$

$$\check{n}_1^{\rho_1} \left( \frac{2 x_h^2}{|V| x_l^2} + \frac{x_h^4}{|V|(|V| - 1) x_l^4} \right) \leq \check{n}_1^{\rho_1} \left( \frac{2 x_h^2}{n_2 x_l^2} + \frac{x_h^4}{n_2(n_2 - 1) x_l^4} \right) = \check{n}_1^{\rho_1} \frac{2(n_2 - 1) x_h^2 x_l^2 + x_h^4}{n_2(n_2 - 1) x_l^4}$$

therefore by the Laplace mechanism calculating $\check{m}$ is $(\epsilon, 0)$ differentially private. $\qquad\square$

**Claim C.9.** *Steps 6-13 are $(3\epsilon, \delta)$ differentially private.*

*Proof.* Mark $\hat{D}$ as a neighbouring database,

$$P(\breve{m} \in S | D) =$$

$$\int_{r_1, r_2 \in \mathbb{R}_{>0} \times \mathbb{R}_{>0}} P(\breve{m} \in S | D, \breve{n}_1 = r_1, n_2 = r_2) p(\breve{n}_1 = r_1, n_2 = r_2 | D) dr_1 dr_2 =$$

$$\int_{r_1, r_2 \in \mathbb{R}_{>0} \times [1, |V|]} P(\breve{m} \in S | D, \breve{n}_1 = r_1, n_2 = r_2) p(\breve{n}_1 = r_1, n_2 = r_2 | D) dr_1 dr_2 +$$

$$\int_{r_1, r_2 \in \mathbb{R}_{>0} \times (|V|, \infty]} P(\breve{m} \in S | D, \breve{n}_1 = r_1, n_2 = r_2) p(\breve{n}_1 = r_1, n_2 = r_2 | D) dr_1 dr_2 \leq^{*}$$

$$\int_{r_1, r_2 \in \mathbb{R}_{>0} \times [1, |V|]} P(\breve{m} \in S | D, \breve{n}_1 = r_1, n_2 = r_2) p(\breve{n}_1 = r_1, n_2 = r_2 | D) dr_1 dr_2 + \delta \leq^{**}$$

$$\int_{r_1, r_2 \in \mathbb{R}_{>0} \times [1, |V|]} e^{2\epsilon} P(\breve{m} \in S | \hat{D}, \breve{n}_1 = r_1, n_2 = r_2) p(\breve{n}_1 = r_1, n_2 = r_2 | \hat{D}) dr_1 dr_2 + \delta \leq$$

$$\int_{r_1, r_2 \in \mathbb{R}_{>0} \times \mathbb{R}_{>0}} e^{2\epsilon} P(\breve{m} \in S | \hat{D}, \breve{n}_1 = r_1, n_2 = r_2) p(\breve{n}_1 = r_1, n_2 = r_2 | \hat{D}) dr_1 dr_2 + \delta =$$

$$e^{2\epsilon} P(\breve{m} \in S | \hat{D}) + \delta$$

where inequality * follows claim C.7 and inequality ** follows claims C.8 and C.6. $\qquad \square$

**Claim C.10.** *Steps 14-19 are $(\epsilon, \delta)$ differentially private with respect to $D$ for $|W| < n_{min}$ and given $n_2, \breve{m}$.*

*Proof.* Mark $l \sim \text{Lap}(\frac{1}{\epsilon})$, and $\hat{D}$ as a neighbouring database. Eq. 34 proves the claim.

$$
\begin{aligned}
&P(S | D, |W| < n_{min}, \breve{m}, n_2) = \\
&P(S \cap \{null\} | D, |W| < n_{min}, \breve{m}, n_2) + \\
&P(S \cap \{null\}^c | D, |W| < n_{min}, \breve{m}, n_2) \leq \\
&e^{\epsilon} P(S \cap \{null\} | \hat{D}, |W| < n_{min}, \breve{m}, n_2) + \delta \leq \\
&e^{\epsilon} P(S | \hat{D}, |W| < d, \breve{m}, n_2) + \delta
\end{aligned}
\tag{34}
$$

where first inequality is true from eq. 35 and the Laplace mechanism for $n_W$.

$$
\begin{aligned}
&P(null | D, |W| < n_{min}, \breve{m}, n_2) = \\
&P(n_W < n_{min} + \frac{1}{\epsilon} \log(\frac{1}{2\delta}) | D, |W| < n_{min}, \breve{m}, n_2) \geq \\
&P(l < \frac{1}{\epsilon} \log(\frac{1}{2\delta})) \geq 1 - \delta
\end{aligned}
\tag{35}
$$

$\qquad \square$

**Claim C.11.** *Step 19 is $(\epsilon, \delta)$ differentially private with respect to $D$ for $|W| \geq n_{min}$ and given $n_2, \breve{m}$.*

*Proof.* For a given $n_2, \breve{m}$ and a neighbouring database, the group $W$ can change by up to one sample. Mark $n = |W|$ and $c = \breve{m}$. From eq. 36, it follows that $W \in \mathcal{D}$, as defined in eq. 5.

$$
\begin{aligned}
&n \geq n_2^{\frac{\rho_2}{\gamma_1}} \Rightarrow \\
&n^{\frac{1}{2}} > n^{\gamma_1} \geq n_2^{\rho_2}
\end{aligned}
\tag{36}
$$

As $W \in \mathcal{D}$, $n \geq n_{b1}$, and $n \geq n_{b2}$, the problem of sampling from $p(\theta|W)$ for $|W| \geq n_{min}$ holds the constraints of claim D.29. Therefore one sample from $p(\theta|W)$ is $(\epsilon, \delta)$ differentially private. $\quad \square$

**Claim C.12.** *Steps 14-18 are $(\epsilon, 0)$ differentially private with respect to $D$ for $|W| > n_{min}$ and given $\breve{m}, n_2$.*

*Proof.* Only data released is $n_W$, and since the sensitivity of $|W|$ given $\breve{m}, n_2$ is 1, then the Laplace mechanism ensures $(\epsilon, 0)$ differential privacy. □

**Corollary C.2.** *Steps 14-19 are $(2\epsilon, \delta)$ differentially private with respect to $D$ for $|W| > n_{min}$ and given $\breve{m}, n_2$.*

**Corollary C.3.** *Steps 14-19 are $(2\epsilon, \delta)$ differentially private with respect to $D$ given $\breve{m}, n_2$.*

## D   Auxiliary Claims

This subsection contains simple claims used to simplify the reading of the proofs. Claims described in this subsection uses the marking defined in eq. 17.

**Claim D.1.** $\rho \frac{1}{1-\lambda} = \frac{ncx_h^2\beta}{\alpha + nx_h^2\beta}$.

*Proof Claim D.1.*

$$\rho \frac{1}{1-\lambda} = \frac{\eta}{2} ncx_h^2\beta \frac{1}{1 - (1 - \frac{\eta}{2}(\alpha + n(\frac{x_h}{2})^2\beta))} = ncx_h^2\beta \frac{1}{\alpha + nx_h^2\beta} = \frac{ncx_h^2\beta}{\alpha + nx_h^2\beta}$$

□

**Claim D.2.** $\rho \frac{1-\lambda^{n-1}}{1-\lambda} + \rho\lambda^{n-1} = \frac{ncx_h^2\beta}{\alpha + nx_h^2\beta}(1 - \lambda^n)$.

*Proof Claim D.2.*

$$\rho \frac{1 - \lambda^{n-1}}{1 - \lambda} + \rho\lambda^{n-1} = \rho(\frac{1 - \lambda^{n-1} + \lambda^{n-1} - \lambda^n}{1 - \lambda}) = \rho(\frac{1 - \lambda^n}{1 - \lambda}) = \frac{ncx_h^2\beta}{\alpha + nx_h^2\beta}(1 - \lambda^n)$$

where the last equality holds from Claim D.1 □

**Claim D.3.** $\rho(\frac{1-\lambda^{n-1}}{1-\lambda}) + \hat{\rho}\lambda^{n-1} = \frac{ncx_h^2\beta}{\alpha + nx_h^2\beta}(1 - \lambda^n(\frac{3}{4}\lambda^{-1} + \frac{1}{4}))$.

*Proof Claim D.3.*

$$\rho(\frac{1 - \lambda^{n-1}}{1 - \lambda}) + \hat{\rho}\lambda^{n-1} = \rho(\frac{1 - \lambda^{n-1}}{1 - \lambda}) + \rho\frac{1}{4}\lambda^{n-1} = \rho(\frac{1 - \frac{3}{4}\lambda^{n-1} - \frac{1}{4}\lambda^n}{1 - \lambda}) =$$

$$\rho(\frac{1 - \lambda^n(\frac{3}{4}\lambda^{-1} + \frac{1}{4})}{1 - \lambda}) = \frac{ncx_h^2\beta}{\alpha + nx_h^2\beta}(1 - \lambda^n(\frac{3}{4}\lambda^{-1} + \frac{1}{4}))$$

where the last equality holds from Claim D.1. □

**Claim D.4.** $\frac{1}{4}\lambda + \frac{3}{4} - \hat{\lambda} = \frac{3}{4}\frac{\eta}{2}\alpha$.

*Proof Claim D.4.*

$$\frac{1}{4}\lambda + \frac{3}{4} - \hat{\lambda} = \frac{1}{4}(1 - \frac{\eta}{2}(\alpha + nx^2\beta)) + \frac{3}{4} - (1 - \frac{\eta}{2}(\alpha + \frac{1}{4}nx^2\beta)) =$$

$$\frac{\eta}{2}[\alpha + \frac{1}{4}nx^2\beta - \frac{1}{4}(\alpha + nx^2\beta)] = \frac{3}{4}\frac{\eta}{2}\alpha$$

□

**Claim D.5.**

$$(1 - \lambda^{kn})(1 - \hat{\lambda}\lambda^{n-1}) - (1 - (\hat{\lambda}\lambda^{(n-1)}))^k(1 - \lambda^{n-1}(\frac{1}{4}\lambda + \frac{3}{4})) =$$

$$\lambda^{n-1}\frac{3}{4}\frac{\eta}{2}\alpha(1 - \hat{\lambda}^k\lambda^{k(n-1)}) + \lambda^{k(n-1)}(\hat{\lambda}^k - \lambda^k)(1 - \lambda^{n-1}\hat{\lambda}).$$

*Proof Claim D.5.*

$$(1 - \lambda^{kn})(1 - \hat{\lambda}\lambda^{n-1}) - (1 - (\hat{\lambda}\lambda^{(n-1)}))^k(1 - \lambda^{n-1}(\tfrac{1}{4}\lambda + \tfrac{3}{4})) =$$

$$\lambda^{n-1}(\tfrac{1}{4}\lambda + \tfrac{3}{4} - \hat{\lambda}) + \lambda^{kn}(\hat{\lambda}\lambda^{n-1} - 1) + (\hat{\lambda}\lambda^{n-1})^k(1 - \lambda^{n-1}(\tfrac{1}{4}\lambda + \tfrac{3}{4})) =$$

$$\lambda^{n-1}(\tfrac{1}{4}\lambda + \tfrac{3}{4} - \hat{\lambda}) + \lambda^{k(n-1)}(\lambda^k(\hat{\lambda}\lambda^{n-1} - 1) + \hat{\lambda}^k(1 - \lambda^{n-1}(\tfrac{1}{4}\lambda + \tfrac{3}{4}))) =$$

$$\lambda^{n-1}(\tfrac{1}{4}\lambda + \tfrac{3}{4} - \hat{\lambda}) + \lambda^{k(n-1)}(\hat{\lambda}^k(1 - \lambda^{n-1}(\tfrac{1}{4}\lambda + \tfrac{3}{4})) - \lambda^k(1 - \hat{\lambda}\lambda^{n-1})) =$$

$$\lambda^{n-1}(\tfrac{1}{4}\lambda + \tfrac{3}{4} - \hat{\lambda}) + \lambda^{k(n-1)}(\hat{\lambda}^k(1 - \lambda^{n-1}(\tfrac{1}{4}\lambda + \tfrac{3}{4})) - \lambda^k(1 - \lambda^{n-1}\hat{\lambda})) =^*$$

$$\lambda^{n-1}\tfrac{\eta}{2}\tfrac{3}{4}\alpha + \lambda^{k(n-1)}(\hat{\lambda}^k(1 - \lambda^{n-1}(\tfrac{1}{4}\lambda + \tfrac{3}{4})) - \lambda^k(1 - \lambda^{n-1}\hat{\lambda})) =^*$$

$$\lambda^{n-1}\tfrac{\eta}{2}\tfrac{3}{4}\alpha + \lambda^{k(n-1)}(\hat{\lambda}^k(1 - \lambda^{n-1}(\hat{\lambda} + \tfrac{3}{4}\tfrac{\eta}{2}\alpha) - \lambda^k(1 - \lambda^{n-1}\hat{\lambda})) =$$

$$\lambda^{n-1}\tfrac{\eta}{2}\tfrac{3}{4}\alpha - \hat{\lambda}^k\lambda^{n-1}\tfrac{3}{4}\tfrac{\eta}{2}\alpha + \lambda^{k(n-1)}(\hat{\lambda}^k(1 - \lambda^{n-1}\hat{\lambda}) - \lambda^k(1 - \lambda^{n-1}\hat{\lambda})) =$$

$$\lambda^{n-1}\tfrac{3}{4}\tfrac{\eta}{2}\alpha(1 - \hat{\lambda}^k\lambda^{k(n-1)}) + \lambda^{k(n-1)}(\hat{\lambda}^k - \lambda^k)(1 - \lambda^{n-1}\hat{\lambda})$$

where equality * holds from claim D.4 $\hfill\square$

**Claim D.6.** $\lambda\sum_{j=0}^{k-1}\lambda^{(n-1)j}\lambda^j[\lambda^{n-1}\rho + \rho\sum_{i=0}^{n-2}\lambda^i] = \lambda(1 - \lambda^{kn})\frac{ncx_h^2\beta}{\alpha + nx_h^2\beta}.$

*Proof Claim D.6.*

$$\lambda\sum_{j=0}^{k-1}\lambda^{(n-1)j}\lambda^j[\lambda^{n-1}\rho + \rho\sum_{i=0}^{n-2}\lambda^i] = \rho\lambda\sum_{i=0}^{kn-1}\lambda^i = \rho\lambda\frac{1 - \lambda^{kn}}{1 - \lambda} =^* \lambda\frac{ncx_h^2\beta}{\alpha + nx_h^2\beta}(1 - \lambda^{kn})$$

Where equality * follows from claim D.1. $\hfill\square$

**Claim D.7.** $\hat{\lambda}\sum_{j=0}^{k-1}\lambda^{(n-1)j}\hat{\lambda}^j[\lambda^{n-1}\hat{\rho} + \rho\sum_{i=0}^{n-2}\lambda^i] = \hat{\lambda}\frac{1 - (\lambda^{n-1}\hat{\lambda})^k}{1 - \lambda^{n-1}\hat{\lambda}}\frac{ncx_h^2\beta}{\alpha + nx_h^2\beta}(1 - \lambda^n(\tfrac{3}{4}\lambda^{-1} + \tfrac{1}{4})).$

*Proof Claim D.7.*

$$\hat{\lambda}\sum_{j=0}^{k-1}\lambda^{(n-1)j}\hat{\lambda}^j[\lambda^{n-1}\hat{\rho} + \rho\sum_{i=0}^{n-2}\lambda^i] =$$

$$\hat{\lambda}\frac{1 - (\lambda^{n-1}\hat{\lambda})^k}{1 - \lambda^{n-1}\hat{\lambda}}[\lambda^{n-1}\hat{\rho} + \rho\frac{1 - \lambda^{n-1}}{1 - \lambda}] =^* \hat{\lambda}\frac{1 - (\lambda^{n-1}\hat{\lambda})^k}{1 - \lambda^{n-1}\hat{\lambda}}\frac{ncx_h^2\beta}{\alpha + nx_h^2\beta}(1 - \lambda^n(\tfrac{3}{4}\lambda^{-1} + \tfrac{1}{4}))$$

Where equality * follows from claims D.1, D.3. $\hfill\square$

**Claim D.8.**
$\lambda\lambda^k - \lambda^k\lambda^n\hat{\lambda} - \hat{\lambda}\hat{\lambda}^k + \hat{\lambda}\hat{\lambda}^k\lambda^n(\tfrac{3}{4}\lambda^{-1} + \tfrac{1}{4}) = (1 - \hat{\lambda}\lambda^{n-1})(\lambda^{k+1} - \hat{\lambda}^{k+1}) + \hat{\lambda}^{k+1}\lambda^{n-1}(\tfrac{3}{4}\tfrac{\eta}{2}\alpha).$

*Proof Claim D.8.*

$$\lambda\lambda^k - \lambda^k\lambda^n\hat{\lambda} - \hat{\lambda}\hat{\lambda}^k + \hat{\lambda}\hat{\lambda}^k\lambda^n(\tfrac{3}{4}\lambda^{-1} + \tfrac{1}{4}) =$$

$$\lambda^{k+1}(1 - \hat{\lambda}\lambda^{n-1}) - \hat{\lambda}^{k+1}(1 - \lambda^{n-1}(\tfrac{1}{4}\lambda + \tfrac{3}{4})) =^*$$

$$\lambda^{k+1}(1 - \hat{\lambda}\lambda^{n-1}) - \hat{\lambda}^{k+1}(1 - \lambda^{n-1}(\hat{\lambda} + \tfrac{3}{4}\tfrac{\eta}{2}\alpha)) =$$

$$(1 - \hat{\lambda}\lambda^{n-1})(\lambda^{k+1} - \hat{\lambda}^{k+1}) + \hat{\lambda}^{k+1}\lambda^{n-1}(\tfrac{3}{4}\tfrac{\eta}{2}\alpha)$$

where equality * holds from claim D.4. $\hfill\square$

**Claim D.9.**
$\lambda(1-\lambda^{kn})(1-\lambda^{n-1}\hat{\lambda}) - \hat{\lambda}(1-(\lambda^{n-1}\hat{\lambda})^k)(1-\lambda^n(\frac{3}{4}\lambda^{-1}+\frac{1}{4})) =$
$(\lambda-\hat{\lambda})(1-\hat{\lambda}\lambda^{n-1}) + \lambda^{n-1}\hat{\lambda}[\frac{3}{4}\frac{\eta}{2}\alpha(1-\hat{\lambda}^k\lambda^{k(n-1)})] + \lambda^{k(n-1)}[(1-\hat{\lambda}\lambda^{n-1})(\hat{\lambda}^{k+1}-\lambda^{k+1})].$

*Proof Claim D.9.*

$\lambda(1-\lambda^{kn})(1-\lambda^{n-1}\hat{\lambda}) - \hat{\lambda}(1-(\lambda^{n-1}\hat{\lambda})^k)(1-\lambda^n(\frac{3}{4}\lambda^{-1}+\frac{1}{4})) =$

$\lambda - \hat{\lambda} - \lambda^n\hat{\lambda}(1-(\frac{3}{4}\lambda^{-1}+\frac{1}{4})) - \lambda^{k(n-1)}[\lambda\lambda^k - \lambda^k\lambda^n\hat{\lambda} - \hat{\lambda}\hat{\lambda}^k + \hat{\lambda}\hat{\lambda}^k\lambda^n(\frac{3}{4}\lambda^{-1}+\frac{1}{4})] =^*$

$\lambda - \hat{\lambda} - \lambda^n\hat{\lambda}(1-(\frac{3}{4}\lambda^{-1}+\frac{1}{4})) - \lambda^{k(n-1)}[(1-\hat{\lambda}\lambda^{n-1})(\lambda^{k+1}-\hat{\lambda}^{k+1}) + \hat{\lambda}^{k+1}\lambda^{n-1}(\frac{3}{4}\frac{\eta}{2}\alpha)] =$

$\lambda - \hat{\lambda} - \lambda^{n-1}\hat{\lambda}(\lambda-(\frac{3}{4}+\frac{1}{4}\lambda)) - \lambda^{k(n-1)}[(1-\hat{\lambda}\lambda^{n-1})(\lambda^{k+1}-\hat{\lambda}^{k+1}) + \hat{\lambda}^{k+1}\lambda^{n-1}(\frac{3}{4}\frac{\eta}{2}\alpha)] =^{**}$

$\lambda - \hat{\lambda} - \lambda^{n-1}\hat{\lambda}(\lambda-(\hat{\lambda}+\frac{3}{4}\frac{\eta}{2}\alpha)) - \lambda^{k(n-1)}[(1-\hat{\lambda}\lambda^{n-1})(\lambda^{k+1}-\hat{\lambda}^{k+1}) + \hat{\lambda}^{k+1}\lambda^{n-1}(\frac{3}{4}\frac{\eta}{2}\alpha)] =$

$(\lambda-\hat{\lambda})(1-\hat{\lambda}\lambda^{n-1}) + \lambda^{n-1}\hat{\lambda}[\frac{3}{4}\frac{\eta}{2}\alpha(1-\hat{\lambda}^k\lambda^{k(n-1)})] + \lambda^{k(n-1)}[(1-\hat{\lambda}\lambda^{n-1})(\hat{\lambda}^{k+1}-\lambda^{k+1})]$

Where equality * follows from claim D.8 and equality ** follows from claim D.4. $\qquad\square$

**Claim D.10.**

$$\lambda(1-\lambda^{kn}) - \hat{\lambda}\frac{1-(\lambda^{n-1}\hat{\lambda})^k}{1-\lambda^{n-1}\hat{\lambda}}(1-\lambda^n(\frac{3}{4}\lambda^{-1}+\frac{1}{4})) =$$
$$(\lambda-\hat{\lambda}) + \frac{\lambda^{n-1}[\frac{3}{4}\frac{\eta}{2}\alpha(1-\hat{\lambda}^k\lambda^{k(n-1)})]}{1-\hat{\lambda}\lambda^{n-1}} + \lambda^{k(n-1)}(\hat{\lambda}^{k+1}-\lambda^{k+1}).$$

*Proof Claim D.10.*

$$\lambda(1-\lambda^{kn}) - \hat{\lambda}\frac{1-(\lambda^{n-1}\hat{\lambda})^k}{1-\lambda^{n-1}\hat{\lambda}}(1-\lambda^n(\frac{3}{4}\lambda^{-1}+\frac{1}{4})) =_{[M5.d]}$$
$$\frac{(\lambda-\hat{\lambda})(1-\hat{\lambda}\lambda^{n-1}) + \lambda^{n-1}\hat{\lambda}[\frac{3}{4}\frac{\eta}{2}\alpha(1-\hat{\lambda}^k\lambda^{k(n-1)})] + \lambda^{k(n-1)}[(1-\hat{\lambda}\lambda^{n-1})(\hat{\lambda}^{k+1}-\lambda^{k+1})]}{(1-\hat{\lambda}\lambda^{n-1})} =$$
$$(\lambda-\hat{\lambda}) + \frac{\lambda^{n-1}[\frac{3}{4}\frac{\eta}{2}\alpha(1-\hat{\lambda}^k\lambda^{k(n-1)})]}{1-\hat{\lambda}\lambda^{n-1}} + \lambda^{k(n-1)}(\hat{\lambda}^{k+1}-\lambda^{k+1})$$

$\qquad\square$

**Claim D.11.** $\frac{ncx_h^2\beta}{\alpha+nx_h^2\beta}(\lambda-\hat{\lambda}+\frac{\lambda^{n-1}[\frac{3}{4}\frac{\eta}{2}\alpha(1-\hat{\lambda}^k\lambda^{k(n-1)})]}{1-\hat{\lambda}\lambda^{n-1}}) + (\rho-\hat{\rho}) > 0.$

*Proof Claim D.11.*

$$\frac{ncx_h^2\beta}{\alpha + nx_h^2\beta}(\lambda - \hat{\lambda} + \frac{\lambda^{n-1}[\frac{3}{4}\frac{\eta}{2}\alpha(1-\hat{\lambda}^k\lambda^{k(n-1)})]}{1-\hat{\lambda}\lambda^{n-1}}) + (\rho - \hat{\rho}) =$$

$$\frac{ncx_h^2\beta}{\alpha + nx_h^2\beta}(\lambda - \hat{\lambda} + \frac{\lambda^{n-1}[\frac{3}{4}\frac{\eta}{2}\alpha(1-\hat{\lambda}^k\lambda^{k(n-1)})]}{1-\hat{\lambda}\lambda^{n-1}}) + \frac{\eta}{2}ncx_h^2\beta(1-\frac{1}{4}) =$$

$$\frac{ncx_h^2\beta}{\alpha + nx_h^2\beta}(1 - \frac{\eta}{2}(\alpha + nx^2\beta) - (1 - \frac{\eta}{2}(\alpha + \frac{1}{4}nx^2\beta)) +$$

$$\frac{\lambda^{n-1}[\frac{3}{4}\frac{\eta}{2}\alpha(1-\hat{\lambda}^k\lambda^{k(n-1)})]}{1-\hat{\lambda}\lambda^{n-1}}) + \frac{3}{4}\frac{\eta}{2}ncx_h^2\beta =$$

$$\frac{ncx_h^2\beta}{\alpha + nx_h^2\beta}(-\frac{3}{4}\frac{\eta}{2}(nx^2\beta) + \frac{\lambda^{n-1}[\frac{3}{4}\frac{\eta}{2}\alpha(1-\hat{\lambda}^k\lambda^{k(n-1)})]}{1-\hat{\lambda}\lambda^{n-1}}) + \frac{3}{4}\frac{\eta}{2}ncx_h^2\beta =$$

$$\frac{ncx_h^2\beta}{\alpha + nx_h^2\beta}\frac{\lambda^{n-1}[\frac{3}{4}\frac{\eta}{2}\alpha(1-\hat{\lambda}^k\lambda^{k(n-1)})]}{1-\hat{\lambda}\lambda^{n-1}} + ncx_h^2\beta[\frac{3}{4}\frac{\eta}{2} - \frac{3}{4}\frac{\eta}{2}\frac{nx^2\beta}{\alpha + nx^2\beta}] =$$

$$\frac{ncx_h^2\beta}{\alpha + nx_h^2\beta}\frac{\lambda^{n-1}[\frac{3}{4}\frac{\eta}{2}\alpha(1-\hat{\lambda}^k\lambda^{k(n-1)})]}{1-\hat{\lambda}\lambda^{n-1}} + ncx_h^2\beta\frac{3}{4}\frac{\eta}{2}[1 - \frac{nx^2\beta}{\alpha + nx^2\beta}] > 0$$

where the last inequality holds because $\lambda, \hat{\lambda} < 1$ and $\alpha > 0$ ∎

**Claim D.12.** $\frac{1}{\alpha} > \lambda^{-2}\eta\frac{1-\lambda^{-2(k+1)n}}{1-\lambda^{-2}}$ *is true for* $k \le \frac{1}{2n}\log_\lambda(\frac{1}{1+\frac{1}{\alpha\eta}(1-\lambda^2)}) - 1$.

*Proof Claim D.12.*

$$\frac{1}{\alpha} \ge \lambda^{-2}\eta\frac{1-\lambda^{-2\dot{k}n}}{1-\lambda^{-2}} \iff \lambda^2\frac{1}{\alpha}\frac{1}{\eta}(1-\lambda^{-2}) \le 1 - \lambda^{-2\dot{k}n} \iff$$

$$\lambda^2\frac{1}{\alpha}\frac{1}{\eta}(\lambda^{-2}-1) \ge \lambda^{-2\dot{k}n} - 1 \iff 1 + \lambda^2\frac{1}{\alpha}\frac{1}{\eta}(\lambda^{-2}-1) \ge \lambda^{-2\dot{k}n} \iff$$

$$-\dot{k} \ge \frac{1}{2n}\log_\lambda(1 + \frac{1}{\alpha\eta}(1-\lambda^2)) \iff \dot{k} \le \frac{1}{2n}\log_\lambda(\frac{1}{1+\frac{1}{\alpha\eta}(1-\lambda^2)})$$

∎

**Claim D.13.**
$\frac{1}{\alpha}(\hat{\lambda}\lambda^{(n-1)})^{2\dot{k}} > \eta\sum_{i=0}^{n-1}\lambda^{2i}\sum_{j=0}^{\dot{k}-1}(\hat{\lambda}^2\lambda^{2(n-1)})^j$ *is true for* $\dot{k} \le \frac{1}{2n}\log_\lambda(\frac{1}{1+\frac{1}{\alpha\eta}(1-\lambda^2)})$.

*Proof Claim D.13.* First note that the inequality can also be written as
$\frac{1}{\alpha} > \eta\sum_{i=0}^{n-1}\lambda^{2i}\sum_{j=0}^{k-1}(\hat{\lambda}\lambda^{(n-1)})^{2(j-k)}$.

Secondly, the right hand term of the inequality could be upper bound as in eq. 37. Therefore for the claim's inequality to holds it is enough that $\frac{1}{\alpha} \ge \eta\lambda^{-2}\frac{1-\lambda^{-2nk}}{1-\lambda^{-2}}$, which proved by claim D.12 to be true for $\dot{k} \le \frac{1}{2n}\log_\lambda(\frac{1}{1+\frac{1}{\alpha\eta}(1-\lambda^2)})$

$$\eta \sum_{i=0}^{n-1} \lambda^{2i} \sum_{j=0}^{k-1} (\hat{\lambda}\lambda^{(n-1)})^{2(j-k)} = \eta \sum_{i=0}^{n-1} \lambda^{2i} \sum_{j=0}^{k-1} \frac{1}{(\hat{\lambda}\lambda^{(n-1)})^{2(k-j)}} <_{k>j}$$

$$\eta \sum_{i=0}^{n-1} \lambda^{2i} \sum_{j=0}^{k-1} \frac{1}{(\lambda\lambda^{(n-1)})^{2(k-j)}} = \eta \sum_{i=0}^{n-1} \lambda^{2i} \sum_{j=0}^{k-1} \frac{1}{\lambda^{2n(k-j)}} =$$

$$\eta \sum_{i=0}^{n-1} \sum_{j=0}^{k-1} \frac{1}{\lambda^{2[nk-nj-i]}} =_{r=nj+i} \eta \sum_{r=0}^{nk-1} \frac{1}{\lambda^{2[nk-r]}} =_{r'=nk-r, 1<r'<nk} \eta \sum_{r'=1}^{nk} \frac{1}{\lambda^{2[r']}} =$$

$$\eta \sum_{i=1}^{nk} \lambda^{-2i} = \eta \frac{\lambda^{-2} - \lambda^{-2(nk+1)}}{1 - \lambda^{-2}} = \eta \lambda^{-2} \frac{1 - \lambda^{-2nk}}{1 - \lambda^{-2}}$$

(37)

$\square$

**Claim D.14.** $\frac{1}{\alpha}(\hat{\lambda}\lambda^{n-1})^{2\dot{k}} \geq \eta(\hat{\lambda}\lambda^{n-1})^{2\dot{k}} \sum_{i=0}^{n-1} \lambda^{2i}$ *is true for* $\dot{k} \leq \frac{1}{2n} \log_\lambda(\frac{1}{1+\frac{1}{\alpha\eta}(1-\lambda^2)})$.

*Proof Claim D.14.* eq. 38 holds because $\lambda, \hat{\lambda} < 1$. By multiplying both sides with $\sum_{i=0}^{n-1} \lambda^{2i}$ get eq. 39. Then noticing that the right term equals to the right term of claim D.13, and hence smaller than the left term of the claim, the claim is proved.

$$(\hat{\lambda}\lambda^{n-1})^{2k} < 1 < \sum_{i=0}^{k-1} (\hat{\lambda}\lambda^{n-1})^{2j}$$

(38)

$$\eta(\hat{\lambda}\lambda^{n-1})^{2\dot{k}} \sum_{i=0}^{n-1} \lambda^{2i} < \eta \sum_{j=0}^{\dot{k}-1} (\hat{\lambda}\lambda^{n-1})^{2j} \sum_{i=0}^{n-1} \lambda^{2i}$$

(39)

$\square$

**Claim D.15.** *The inequality*

$$(\hat{\lambda}\lambda^{n-r})^2 [\frac{1}{\alpha}(\hat{\lambda}\lambda^{n-1})^{2k} + \eta \sum_{j=0}^{k-1} (\hat{\lambda}\lambda^{n-1})^{2j} \sum_{i=0}^{n-1} \lambda^{2i}] > \eta \sum_{i=0}^{n-r} \lambda^{2i}$$

*holds for* $x_h^2\beta > 3, n > \frac{1}{2\alpha x_h^2\beta} - \frac{1}{x_h^2\beta}$.

*Proof Claim D.15.* Left hand side can be lower bounded according to eq. 40, while right hand side can be upper bounded according to eq. 41. Therefore it's enough to show that $\lambda^{2n}[\frac{1}{\alpha}\lambda^{2kn} + \eta\frac{1-\lambda^{2kn}}{1-\lambda^2}] > \eta\frac{1-\lambda^{2n}}{1-\lambda^2}$, which according to eq. 42 is equivalent to showing that $(2nx_h^2\beta-1)\frac{1}{\alpha}\lambda^{2(k+1)n}+2(2\lambda^{2n}-1) > 0$. Since $n > \frac{1}{2\alpha x_h^2\beta} - \frac{1}{x_h^2\beta}$ claim D.19 applies and therefore $\lambda^{2n} \geq e^{-\frac{2}{x_h^2\beta}}$. Consequently it's enough to show that $(2nx_h^2\beta-1)\frac{1}{\alpha}\lambda^{2(k+1)n}+2(2e^{-\frac{2}{x_h^2\beta}}-1) > 0$, which is true for $x_h^2\beta > 3$ by claim D.16.

$$(\hat{\lambda}\lambda^{n-r})^2 [\frac{1}{\alpha}(\hat{\lambda}\lambda^{n-1})^{2k} + \eta \sum_{j=0}^{k-1} (\hat{\lambda}\lambda^{n-1})^{2j} \sum_{i=0}^{n-1} \lambda^{2i}] >$$

$$(\hat{\lambda}\lambda^{n-1})^2 [\frac{1}{\alpha}(\hat{\lambda}\lambda^{n-1})^{2k} + \eta \sum_{j=0}^{k-1} (\hat{\lambda}\lambda^{n-1})^{2j} \sum_{i=0}^{n-1} \lambda^{2i}] >$$

$$\lambda^{2n}[\frac{1}{\alpha}\lambda^{2kn} + \eta \sum_{j=0}^{k-1} \lambda^{2jn} \sum_{i=0}^{n-1} \lambda^{2i}] = \lambda^{2n}[\frac{1}{\alpha}\lambda^{2kn} + \eta\frac{1-\lambda^{2kn}}{1-\lambda^2}]$$

(40)

First inequality holds because $\lambda < 1$ and $r > 1$, and second inequality holds because $\lambda < \hat{\lambda}$.

$$\eta \sum_{i=0}^{n-r} \lambda^{2i} < \eta \sum_{i=0}^{n-1} \lambda^{2i} = \eta \frac{1-\lambda^{2n}}{1-\lambda^2} \tag{41}$$

Inequality holds because $\lambda < \hat{\lambda}$ and $r > 1$.

$$\lambda^{2n}[\frac{1}{\alpha}\lambda^{2kn} + \eta\frac{1-\lambda^{2kn}}{1-\lambda^2}] > \eta\frac{1-\lambda^{2n}}{1-\lambda^2}$$

$$\lambda^{2n}(1-\lambda^2)\frac{1}{\alpha}\lambda^{2kn} + \eta\lambda^{2n}(1-\lambda^{2kn}) > \eta(1-\lambda^{2n})$$

$$(1-\lambda^2)\frac{1}{\alpha}\lambda^{2(k+1)n} + \eta(2\lambda^{2n}-\lambda^{2(k+1)n}-1) > 0$$

$$(\alpha + nx_h^2\beta)^2(1-\lambda^2)\frac{1}{\alpha}\lambda^{2(k+1)n} + 2(2\lambda^{2n}-\lambda^{2(k+1)n}-1) > 0 \tag{42}$$

$$(\alpha + nx_h^2\beta)^2(1-(1-\frac{1}{\alpha+nx_h^2\beta})^2)\frac{1}{\alpha}\lambda^{2(k+1)n} + 2(2\lambda^{2n}-\lambda^{2(k+1)n}-1) > 0$$

$$(2(\alpha+nx_h^2\beta)-1)\frac{1}{\alpha}\lambda^{2(k+1)n} + 2(2\lambda^{2n}-\lambda^{2(k+1)n}-1) > 0$$

$$2\lambda^{2(k+1)n} + (2nx^2\beta-1)\frac{1}{\alpha}\lambda^{2(k+1)n} + 2(2\lambda^{2n}-\lambda^{2(k+1)n}-1) > 0$$

$$(2nx_h^2\beta-1)\frac{1}{\alpha}\lambda^{2(k+1)n} + 2(2\lambda^{2n}-1) > 0$$

$\square$

**Claim D.16.** *For $x^2\beta > 3$ the inequality $(2e^{-\frac{2}{x^2\beta}} - 1) > 0$ holds.*

*Proof Claim D.16.* It's easy to see that the inequality holds only if $x^2\beta \geq \frac{-2}{\ln\frac{1}{2}}$. Since $\frac{-2}{\ln\frac{1}{2}} < 3$ claim is proved. $\square$

**Claim D.17.** *For $\dot{k}$ as defined in lemma A.4, and the conditions of claim D.19*

$$\frac{1}{\alpha}(e^{\frac{2}{x_h^2\beta}} + \alpha\frac{(e^{\frac{2}{x_h^2\beta}}-1)}{(\alpha+nx^2\beta)+\frac{1}{8}}) > \lambda^{-2}\eta\frac{1-\lambda^{-2(\lceil\dot{k}\rceil+1)n}}{1-\lambda^{-2}}.$$

*Proof Claim D.17.*

$$\eta\frac{1-\lambda^{-2(\lceil\dot{k}\rceil+1)n}}{\lambda^2-1} \leq \eta\frac{\lambda^{-2(\dot{k}+2)n}-1}{1-\lambda^2} = \eta\frac{\lambda^{-2(\frac{1}{2n}\log_\lambda(\frac{1}{1+\frac{1}{\alpha\eta}(1-\lambda^2)})-1+2)n}-1}{1-\lambda^2} =$$

$$\eta\frac{\lambda^{-\log_\lambda(\frac{1}{1+\frac{1}{\alpha\eta}(1-\lambda^2)})}\lambda^{-2n}-1}{1-\lambda^2} = \eta\frac{[1+\frac{1}{\alpha\eta}(1-\lambda^2)]\lambda^{-2n}-1}{1-\lambda^2} =$$

$$\eta\frac{(1-\lambda^2)\lambda^{-2n}\frac{1}{\alpha\eta}}{1-\lambda^2} + \eta\frac{\lambda^{-2n}-1}{1-\lambda^2} = \frac{1}{\alpha}\lambda^{-2n} + \frac{(\lambda^{-2n}-1)}{(\alpha+nx^2\beta)+\frac{1}{8}} \leq$$

$$e^{\frac{2}{x^2\beta}}\frac{1}{\alpha} + \frac{1}{\alpha}\alpha\frac{(e^{\frac{2}{x^2\beta}}-1)}{(\alpha+nx^2\beta)+\frac{1}{8}} = \frac{1}{\alpha}[e^{\frac{2}{x^2\beta}} + \alpha\frac{(e^{\frac{2}{x^2\beta}}-1)}{(\alpha+nx^2\beta)+\frac{1}{8}}]$$

where the fourth equality holds from eq. 43 and the second inequality holds from D.19.

$$\frac{\eta}{\lambda^2-1} = \eta\frac{1}{(1-\frac{\eta}{2}(\alpha+nx^2\beta))^2-1} = \eta\frac{1}{\eta(\alpha+nx^2\beta)+(\frac{\eta}{2}(\alpha+nx^2\beta))^2} =$$

$$\frac{1}{(\alpha+nx^2\beta)+\frac{\eta}{4}(\alpha+nx^2\beta)^2} = \frac{1}{(\alpha+nx^2\beta)+\frac{1}{8}} \tag{43}$$

$\square$

**Claim D.18.**

$$\forall k > 0 : 1 - (\frac{\lambda}{\hat{\lambda}})^k \geq \frac{\frac{3}{4}nx^2\beta}{(\alpha + nx^2\beta)^2 - (\alpha + \frac{1}{4}nx^2\beta)}.$$

*Proof Claim D.18.*

$$1 - (\frac{\lambda}{\hat{\lambda}})^k \geq 1 - \frac{\lambda}{\hat{\lambda}} = 1 - \frac{1 - \frac{1}{\alpha + nx^2\beta}}{1 - \frac{\alpha + \frac{1}{4}nx^2\beta}{(\alpha + nx^2\beta)^2}} = 1 - \frac{(\alpha + nx^2\beta)^2 - (\alpha + nx^2\beta)}{(\alpha + nx^2\beta)^2 - (\alpha + \frac{1}{4}nx^2\beta)} =$$

$$1 - \frac{\alpha^2 + 2nx^2\alpha\beta + (nx^2\beta)^2 - \alpha - nx^2\beta}{\alpha^2 + 2nx^2\alpha\beta + (nx^2\beta)^2 - \alpha - \frac{1}{4}nx^2\beta} = \frac{\frac{3}{4}nx^2\beta}{(\alpha + nx^2\beta)^2 - (\alpha + \frac{1}{4}nx^2\beta)}$$

Where first inequality holds because $\lambda < \hat{\lambda}$. □

**Claim D.19.** *For the conditions of claim D.21,*

$$(1 - \frac{1}{\alpha + nx^2\beta})^{2n} \geq e^{-\frac{2}{x^2\beta}}.$$

*Proof Claim D.19.* The proof is easily deduced from claims D.20 and D.21 □

**Claim D.20.**

$$\lim_{n \to \infty} (1 - \frac{1}{\alpha + nx^2\beta})^{2n} = e^{-\frac{2}{x^2\beta}}.$$

*Proof Lemma D.20.* From eq. 44, it is enough to find $\lim_{n \to \infty} \frac{\ln(1 - \frac{1}{\alpha + nx^2\beta})}{\frac{1}{2n}}$.

Since $\lim_{n \to \infty} \frac{\ln(1 - \frac{1}{\alpha + nx^2\beta})}{\frac{1}{2n}} = \frac{0}{0}$, and both the numerator and denominator are differentiable around $\infty$, the use of L'Hôpital's rule is possible as shown in eq. 45. This proves the claim.

$$(1 - \frac{1}{\alpha + nx^2\beta})^{2n} = e^{\ln[(1 - \frac{1}{\alpha + nx^2\beta})^{2n}]} = e^{2n\ln(1 - \frac{1}{\alpha + nx^2\beta})} = e^{\frac{\ln(1 - \frac{1}{\alpha + nx^2\beta})}{\frac{1}{2n}}} \tag{44}$$

$$\lim_{n \to \infty} \frac{\frac{d}{dn}\ln(1 - \frac{1}{\alpha + nx^2\beta})}{\frac{d}{dn}\frac{1}{2n}} = \lim \frac{\frac{x^2\beta}{(\alpha + nx^2\beta - 1)(\alpha + nx^2\beta)}}{-\frac{1}{2n^2}} = -\lim \frac{2n^2 x^2\beta}{(nx^2\beta)^2} = -\frac{2}{x^2\beta} \tag{45}$$

□

**Claim D.21.**

$$\forall n > \frac{1}{2\alpha x^2\beta} - \frac{1}{x^2\beta} : \frac{d}{dn}(1 - \frac{1}{\alpha + nx^2\beta})^{2n} < 0.$$

*Proof claim D.21.* First, a simplified term for the derivative is found at eq. 46.

$$\frac{d}{dn}(1 - \frac{1}{\alpha + nx^2\beta})^{2n} = \frac{d}{dn}e^{2n\ln(1 - \frac{1}{\alpha + nx^2\beta})} =$$

$$(1 - \frac{1}{\alpha + nx^2\beta})^{2n}[2\ln(1 - \frac{1}{\alpha + nx^2\beta}) + 2n\frac{1}{1 - \frac{1}{\alpha + nx^2\beta}} \cdot \frac{x^2\beta}{(\alpha + nx^2\beta)^2}] = \tag{46}$$

$$(1 - \frac{1}{\alpha + nx^2\beta})^{2n}[2\ln(1 - \frac{1}{\alpha + nx^2\beta}) + \frac{2nx^2\beta}{(\alpha + nx^2\beta - 1)(\alpha + nx^2\beta)}]$$

A lower bound for the $ln$ term can be found using Taylor's theorem as shown in eq .47, where $0 \leq \xi \leq \frac{1}{\alpha + nx^2\beta}$.

$$\ln(1 - \frac{1}{\alpha + nx^2\beta}) = -\frac{1}{\alpha + nx^2\beta} - \frac{1}{2}\frac{1}{(1 - \xi)^2}(\frac{1}{\alpha + nx^2\beta})^2 \leq$$

$$-\frac{1}{\alpha + nx^2\beta} - \frac{1}{2}(\frac{1}{\alpha + nx^2\beta})^2 \tag{47}$$

From equations 46 and 47 it is enough to find the terms for which $\frac{nx^2\beta}{(\alpha+nx^2\beta-1)(\alpha+nx^2\beta)} < \frac{1}{\alpha+nx^2\beta} + \frac{1}{2}\frac{1}{(\alpha+nx^2\beta)^2}$ holds. A simplified version of this inequality is found at (48), and it can be easily seen that for $\alpha > \frac{1}{2}(\frac{1}{nx^2\beta}+1) \iff n > \frac{1}{2\alpha x^2\beta} - \frac{1}{x^2\beta}$ this inequality holds.

$$
\begin{aligned}
\frac{nx^2\beta}{(\alpha+nx^2\beta-1)(\alpha+nx^2\beta)} &< \frac{1}{\alpha+nx^2\beta} + \frac{1}{2}\frac{1}{(\alpha+nx^2\beta)^2} \iff \\
0 &< 2\alpha^2 + 2nx^2\beta\alpha - 2\alpha - 2nx^2\beta + \alpha + nx^2\beta - 1 \iff \\
0 &< nx^2\beta(2\alpha-1) + \alpha(2\alpha-1) - 1
\end{aligned}
\tag{48}
$$

$\square$

**Claim D.22.** *For $n > \frac{\alpha}{x_h^2\beta}(e^{\frac{2}{x_h^2\beta}} - 2) + \frac{1}{2x_h^2\beta}$ and the conditions of claim D.19, $\dot{k}$, as defined in lemma A.4, is positive.*

*Proof Claim D.22.* The claim's inequality is simplified at eq. 49

$$
\begin{aligned}
\dot{k} &> 0 \\
\frac{1}{2n}\log_\lambda\left(\frac{1}{1+\frac{1}{\alpha\eta}(1-\lambda^2)}\right) - 1 &> 0 \\
\log_\lambda\left(\frac{1}{1+\frac{1}{\alpha\eta}(1-\lambda^2)}\right) &> 2n \\
\frac{\ln\left(\frac{1}{1+\frac{1}{\alpha\eta}(1-\lambda^2)}\right)}{\ln\lambda} &> 2n \\
\ln\left(\frac{1}{1+\frac{1}{\alpha\eta}(1-\lambda^2)}\right) &< 2n\ln\lambda \\
\ln\left(\frac{1}{1+\frac{1}{\alpha\eta}(1-\lambda^2)}\right) &< \ln\lambda^{2n} \\
\frac{1}{1+\frac{1}{\alpha\eta}(1-\lambda^2)} &< \lambda^{2n} \\
\lambda^{-2n} &< 1 + \frac{1}{\alpha\eta}(1-\lambda^2) \\
\lambda^{-2n} - 1 &< \frac{1}{\alpha\eta}(1-\lambda^2)
\end{aligned}
\tag{49}
$$

By claim D.19 $\lambda^{-2n} - 1 < e^{\frac{2}{x_h^2\beta}} - 1$, therefore it is enough to find terms for $e^{\frac{2}{x_h^2\beta}} - 1 < \frac{1}{\alpha\eta}(1-\lambda^2)$, which is done at eq. 50, which proves the claim.

$$
\begin{aligned}
e^{\frac{2}{x_h^2\beta}} - 1 &< \frac{1}{\alpha\eta}(1-\lambda^2) \\
\alpha\eta(e^{\frac{2}{x_h^2\beta}} - 1) &< (1-\lambda^2) \\
\alpha\eta(e^{\frac{2}{x_h^2\beta}} - 1) &< 1 - (1 - \frac{\eta}{2}(\alpha+nx_h^2\beta))^2 \\
\alpha(e^{\frac{2}{x_h^2\beta}} - 1) &< (\alpha+nx_h^2\beta) - \frac{\eta}{4}(\alpha+nx^2\beta)^2 \\
\alpha(e^{\frac{2}{x_h^2\beta}} - 1) &< (\alpha+nx_h^2\beta) - \frac{1}{2} \\
\alpha(e^{\frac{2}{x_h^2\beta}} - 2) + \frac{1}{2} &< nx_h^2\beta \\
\frac{\alpha}{x_h^2\beta}(e^{\frac{2}{x_h^2\beta}} - 2) + \frac{1}{2x_h^2\beta} &< n
\end{aligned}
\tag{50}
$$

$\qquad\qquad\qquad\qquad\qquad\qquad\qquad\qquad\qquad\qquad\qquad\qquad\qquad\square$

**Claim D.23.** *For $\dot{k}$ as defined in lemma A.4, and the conditions of lemma A.5*

$$p(\hat{\theta}_{(\lceil \dot{k} \rceil+1)n} > \mu_{(\lceil \dot{k} \rceil+1)n}|\hat{D}) \leq e^{-e^{-\frac{2}{x^2\beta}}\frac{\alpha}{2v_1}(\frac{3}{32x^2\beta})^2(\frac{c}{n})^2}.$$

*Proof claim D.23.*

$$p(\hat{\theta}_{(\lceil \dot{k} \rceil+1)n} > \mu_{(\lceil \dot{k} \rceil+1)n}|\hat{D}) \leq$$

$$\frac{1}{n}\sum_{r=1}^{n}\exp(-\frac{(\mu_{(\lceil \dot{k} \rceil+1)n} - \hat{\mu}^r_{(\lceil \dot{k} \rceil+1)n})^2}{2(\sigma^r_{(\lceil \dot{k} \rceil+1)n})^2}) \leq$$

$$\frac{1}{n}\sum_{r=1}^{n}\exp(-e^{-\frac{2}{x^2\beta}}\frac{\alpha}{2v_1}(\frac{3}{32x^2\beta})^2(\frac{c}{n})^2) =$$

$$\exp(-e^{-\frac{2}{x^2\beta}}\frac{\alpha}{2v_1}(\frac{3}{32x^2\beta})^2(\frac{c}{n})^2)$$

Where the first inequality holds due to lemma 4.4 and second inequality holds due to lemma A.5. $\square$

**Claim D.24.** *for $n > 1 + 10\frac{x_h^2}{x_l^2}\frac{\nu}{\beta}$, the inequality $\frac{1}{10}(\alpha + (z + x_n^2)\beta) > \nu(\hat{x}_n^2 - x_n^2)$ holds.*

*Proof Claim D.24.* Notice that $\frac{1}{10}(\alpha+(z+x_n^2)\beta) > \frac{1}{10}z\beta > \frac{1}{10}(n-1)x_l^2\beta$ and $\nu x_h^2 > \nu(\hat{x}_n^2 - x_n^2)$, Therefore a sufficient condition will be that $\frac{1}{10}(n-1)x_l^2\beta > \nu x_h^2$, which is equivalent to $n > 1 + \frac{x_h^2}{x_l^2}\frac{10\nu}{\beta}$. $\square$

**Claim D.25.** *For the $(\sigma^2)^*_\nu$ as defined in eq. 14*

$$(\sigma^2)^*_\nu > 0.$$

*Proof Claim D.25.*

$$(\sigma^2)^*_\nu = \nu\sigma^2 + (1-\nu)\hat{\sigma}^2 = \frac{\nu}{\alpha + (z + x_n^2)\beta} + \frac{1-\nu}{\alpha + (z + \hat{x}_n^2)\beta} =$$

$$\frac{\nu(\alpha + (z + \hat{x}_n^2)\beta) + (1-\nu)(\alpha + (z + x_n^2)\beta)}{(\alpha + (z + x_n^2)\beta)(\alpha + (z + \hat{x}_n^2)\beta)} = \frac{\alpha + (z + x_n^2)\beta + \nu(x_n^2 - \hat{x}_n^2)}{(\alpha + (z + x_n^2)\beta)(\alpha + (z + \hat{x}_n^2)\beta)} \tag{51}$$

Therefore, a sufficient condition is that $\alpha + (z + x_n^2)\beta + \nu(x_n^2 - \hat{x}_n^2) > 0$. Since the condition of Lemma 4.2 dictates $n > 1 + 10\frac{x_h^2}{x_l^2}\frac{\nu}{\beta}$ then claim D.24 holds, which satisfy this condition. $\square$

**Claim D.26.** *For the Bayesian linear regression problem on domain $\mathcal{D}$, and $\sigma, \hat{\sigma}$ defined in eq. 14*

$$\ln\frac{\sigma}{\hat{\sigma}} \leq \frac{x_h^2}{2(n-1)x_l^2}.$$

*Proof Claim D.26.* Consider $c_1 = \frac{x_h^2}{(n-1)x_l^2}$,

$$c_1 = \frac{x_h^2}{(n-1)x_l^2} > \frac{\hat{x}_n^2 - x_n^2}{z + x_n^2} > \frac{\hat{x}_n^2\beta - x_n^2\beta}{\alpha + (z + x_n^2)\beta} = \frac{\alpha + (z + \hat{x}_n^2)\beta}{\alpha + (z + x_n^2)\beta} - 1 \tag{52}$$

Where eq. 52 holds trivially for $\hat{x}_n \leq x_n$, therefore it is assumed that $\hat{x}_n > x_n$. From eq. 52, by Taylor theorem and $0 \leq \zeta \leq c_1$ following inequality holds

$$e^{c_1} = 1 + c_1 + \frac{e^\zeta}{2}(c_1)^2 > 1 + c_1 > \frac{\alpha + (z + \hat{x}_n^2)\beta}{\alpha + (z + x_n^2)\beta}$$

Consequently, because the natural logarithm is monotonically increasing the following equation also holds

$$\frac{1}{2}c_1 > \frac{1}{2}\ln\frac{\alpha + (z + \hat{x}_n)\beta}{\alpha + (z + x_n)\beta} = \ln\frac{\sigma}{\hat{\sigma}}$$

Therefore $\ln\frac{\sigma}{\hat{\sigma}} < \frac{1}{2}\frac{x_h^2}{(n-1)x_l^2}$ $\qquad\qquad\qquad\qquad\qquad\qquad\qquad\qquad\qquad\qquad\square$

**Claim D.27.** *For the Bayesian linear regression problem on domain $\mathcal{D}$, the conditions of Lemma 4.2 and $(\sigma^2)^*_\nu, \hat{\sigma}$ defined in eq. 14*

$$\frac{1}{2}(\nu - 1)\ln\frac{\hat{\sigma}^2}{(\sigma^2)^*_\nu} \leq \frac{1}{2}(\nu - 1)\frac{\nu x_h^2}{2((n-1)x_l^2 - \nu x_h^2)}.$$

*Proof Claim D.27.* consider $c_1 = \frac{\nu x_h^2}{((n-1)x_l^2 - \nu x_h^2)}$,

$$c_1 = \frac{\nu x_h^2}{(n-1)x_l^2 - \nu x_h^2} \geq^* \frac{\nu\beta x_h^2}{\alpha + (n-1)x_l^2\beta - \nu\beta x_h^2} \geq^*$$

$$\frac{\nu\beta\hat{x}_n^2}{\alpha + (z + x_n^2)\beta - \nu\beta x_n^2} \geq \frac{\nu\beta(\hat{x}_n^2 - x_n^2)}{\alpha + (z + x_n^2)\beta - \nu\beta(x_n^2 - \hat{x}_n^2)} =$$

$$\frac{\alpha + (z + x_n^2)\beta}{\alpha + (z + x_n^2)\beta + \nu\beta(x_n^2 - \hat{x}_n^2)} - 1 =$$

$$\frac{1}{\alpha + (z + \hat{x}_n^2)\beta} \cdot \frac{(\alpha + (z + x_n^2)\beta)(\alpha + (z + \hat{x}_n^2)\beta)}{\alpha + (z + x_n^2)\beta + \nu\beta(x_n^2 - \hat{x}_n^2)} - 1 =$$

$$\frac{\hat{\sigma}^2}{(\sigma^2)^*_\nu} - 1$$

Where inequalities * holds under assumption that $n > 1 + \nu\frac{x_h^2}{x_l^2}$, and last equality holds from eq. 51. Therefore, by using Taylor theorem and $0 \leq \zeta \leq c_1$ following inequality holds

$$e^{c_1} = 1 + c_1 + \frac{e^\zeta}{2}(c_1)^2 > 1 + c_1 \geq \frac{\hat{\sigma}^2}{(\sigma^2)^*_\nu}$$

From this inequality, and because the natural logarithm is monotonically increasing $\ln\frac{\hat{\sigma}^2}{(\sigma^2)^*_\nu} \leq c_1$, therefore

$$\frac{1}{2}(\nu - 1)\ln\frac{\hat{\sigma}^2}{(\sigma^2)^*_\nu} \leq \frac{1}{2}(\nu - 1)c_1 = \frac{1}{2}(\nu - 1)\frac{\nu x_h^2}{((n-1)x_l^2 - \nu x_h^2)}.$$

$\square$

**Claim D.28.** *For the Bayesian linear regression problem on domain $\mathcal{D}$, the definitions of eq. 14, and the conditions of Lemma 4.2, the value $\frac{\nu}{2}\frac{(\mu - \hat{\mu})^2}{(\sigma^2)^*_\nu}$ is bounded by*

$$2\nu\beta(\frac{x_h^4}{\frac{9}{10}n^{1-2\gamma_1}x_l^2}) + 2\nu\beta(\frac{(x_h^2\beta)(x_h^2\alpha + x_h^4\beta)}{\frac{9}{10}(x_l^2\beta)^2})\frac{(c + n^{\gamma_1})}{n^{2-\gamma_1}} + \frac{\nu}{2}(\frac{(x_h^2\alpha + x_h^4\beta)^2}{\frac{9}{10}x_l^6\beta})\frac{(c + n^{\gamma_1})^2}{n^3}.$$

*Proof Claim D.28.* First bound $|\mu - \hat{\mu}|$,

$$|\mu - \hat{\mu}| = \beta|\frac{q + x_n y_n}{\alpha + (z + x_n^2)\beta} - \frac{q + \hat{x}_n\hat{y}_n}{\alpha + (z + \hat{x}_n^2)\beta}| =$$

$$|\frac{(q + x_n y_n)(\alpha + (z + \hat{x}_n^2)\beta) - (q + \hat{x}_n\hat{y}_n)(\alpha + (z + x_n^2)\beta)}{(\alpha + (z + x_n^2)\beta)(\alpha + (z + \hat{x}_n^2)\beta)}| =$$

$$\beta|\frac{q\hat{x}_n^2\beta + x_n y_n\alpha + x_n y_n z\beta + x_n y_n\hat{x}_n^2\beta - qx_n^2\beta - \hat{x}_n\hat{y}_n\alpha - \hat{x}_n\hat{y}_n z\beta - \hat{x}_n\hat{y}_n x_n^2\beta}{(\alpha + (z + x_n^2)\beta)(\alpha + (z + \hat{x}_n^2)\beta)}| =$$

$$\beta|\frac{\hat{x}_n^2 z(\frac{q}{z} - \frac{\hat{y}_n}{\hat{x}_n})\beta - x_n^2 z(\frac{q}{z} - \frac{y_n}{x_n})\beta + \alpha(x_n y_n - \hat{x}_n\hat{y}_n) + x_n\hat{x}_n\beta(y_n\hat{x}_n - \hat{y}_n x_n)}{(\alpha + (z + x_n^2)\beta)(\alpha + (z + \hat{x}_n^2)\beta)}| <$$

$$\beta|\frac{\hat{x}_h^2 z(2n^{\gamma_1})\beta + \alpha x_h^2(c + n^{\gamma_1}) + x_h^4\beta(c + n^{\gamma_1})}{(\alpha + (z + x_n^2)\beta)(\alpha + (z + \hat{x}_n^2)\beta)}| =$$

$$\beta|\frac{2\hat{x}_h^2\beta zn^{\gamma_1} + (x_h^2\alpha + x_h^4\beta)(c + n^{\gamma_1})}{(\alpha + (z + x_n^2)\beta)(\alpha + (z + \hat{x}_n^2)\beta)}|$$

Therefore,

$$\frac{\nu}{2}\frac{(\mu-\hat{\mu})^2}{(\sigma^2)^*_\nu} \leq$$

$$\frac{\nu}{2}\beta^2\big(\frac{2\hat{x}_h^2\beta z n^{\gamma_1}+(x_h^2\alpha+x_h^4\beta)(c+n^{\gamma_1})}{(\alpha+(z+x_n^2)\beta)(\alpha+(z+\hat{x}_n^2)\beta)}\big)^2\cdot\big(\frac{\alpha+(z+x_n^2)\beta+\nu(x_n^2-\hat{x}_n^2)}{(\alpha+(z+x_n^2)\beta)(\alpha+(z+\hat{x}_n^2)\beta)}\big)^{-1}=$$

$$\frac{\nu}{2}\frac{\beta^2(2\hat{x}_h^2\beta z n^{\gamma_1}+(x_h^2\alpha+x_h^4\beta)(c+n^{\gamma_1}))^2}{(\alpha+(z+x_n^2)\beta)(\alpha+(z+\hat{x}_n^2)\beta)(\alpha+(z+x_n^2)\beta+\nu(x_n^2-\hat{x}_n^2))}\leq^*$$

$$\frac{\nu}{2}\frac{\beta^2(2x_h^2\beta z n^{\gamma_1}+(x_h^2\alpha+x_h^4\beta)(c+n^{\gamma_1}))^2}{\frac{9}{10}(\alpha+(z+x_n^2)\beta)(\alpha+(z+\hat{x}_n^2)\beta)(\alpha+(z+x_n^2)\beta)}=$$

$$\frac{\nu}{2}\beta^2\big(\frac{(2x_h^2\beta)^2z^2n^{2\gamma_1}+2(2x_h^2\beta)(x_h^2\alpha+x_h^4\beta)zn^{\gamma_1}(c+n^{\gamma_1})+(x_h^2\alpha+x_h^4\beta)^2(c+n^{\gamma_1})^2}{\frac{9}{10}(\alpha+(z+x_n^2)\beta)^2(\alpha+(z+\hat{x}_n^2)\beta)}\big)\leq$$

$$\frac{\nu}{2}\beta^2\big(\frac{(2x_h^2\beta)^2z^2n^{2\gamma_1}+(4x_h^2\beta)(x_h^2\alpha+x_h^4\beta)zn^{\gamma_1}(c+n^{\gamma_1})+(x_h^2\alpha+x_h^4\beta)^2(c+n^{\gamma_1})^2}{\frac{9}{10}((z+x_n^2)\beta)^2((z+\hat{x}_n^2)\beta)}\big)\leq^{**}$$

$$\frac{\nu}{2}\beta^2\big(\frac{(2x_h^2\beta)^2n^{2\gamma_1}}{\frac{9}{10}nx_l^2\beta^3}\big)+$$

$$\frac{\nu}{2}\beta^2\big(\frac{(4x_h^2\beta)(x_h^2\alpha+x_h^4\beta)n^{\gamma_1}(c+n^{\gamma_1})}{\frac{9}{10}(nx_l^2)^2\beta^3}\big)+\frac{\nu}{2}\beta^2\big(\frac{(x_h^2\alpha+x_h^4\beta)^2(c+n^{\gamma_1})^2}{\frac{9}{10}(nx_l^2\beta)^3}\big)=$$

$$2\nu\beta\big(\frac{x_h^4}{\frac{9}{10}n^{1-2\gamma_1}x_l^2}\big)+2\nu\beta\big(\frac{(x_h^2\beta)(x_h^2\alpha+x_h^4\beta)}{\frac{9}{10}(x_l^2\beta)^2}\big)\frac{(c+n^{\gamma_1})}{n^{2-\gamma_1}}+\frac{\nu}{2}\big(\frac{(x_h^2\alpha+x_h^4\beta)^2}{\frac{9}{10}x_l^6\beta}\big)\frac{(c+n^{\gamma_1})^2}{n^3}$$

Inequality * is true because Lemma 4.2 conditions dictates that $n > 1+\frac{x_h^2}{x_l^2}\frac{10\nu}{\beta}$, and according to claim D.24 this promises that $\frac{1}{10}(\alpha+(z+x_n^2)\beta)>\nu(\hat{x}_n^2-x_n^2)$. Inequality ** follows from $n >> 1 \Rightarrow (n-1)x_l \approx nx_l$. □

**Claim D.29.** *For the conditions and definitions of Lemma 4.3, one sample from the posterior is* $(\epsilon,\delta)$ *differentially private for the following terms on $n$ and $\nu$.*

$$\nu = 1 + \frac{2\ln(\frac{1}{\delta})}{\epsilon}$$

$$n \geq \max\{1+\frac{x_h^2}{x_l^2}\frac{8}{\epsilon}, 1+\nu\frac{x_h^2}{x_l^2}(1+8\frac{(\nu-1)}{\epsilon}),$$

$$(\frac{16\nu\beta x_h^4}{\frac{9}{10}\epsilon x_l^2})^{\frac{1}{1-2\gamma_1}},$$

$$(\frac{16\nu\beta}{\epsilon}(\frac{(x_h^2\beta)(x_h^2\alpha+x_h^4\beta)}{\frac{9}{10}(x_l^2\beta)^2})(c+n^{\gamma_1}))^{\frac{1}{2-\gamma_1}},$$

$$(\frac{4\nu}{\epsilon}(\frac{(x_h^2\alpha+x_h^4\beta)^2}{\frac{9}{10}x_l^6\beta})(c+n^{\gamma_1}))^{\frac{2}{3}}\}$$

*Proof Claim D.29.* By Lemma 4.3, one sample from the posterior is $(\epsilon_1+\frac{\ln(\frac{1}{\delta})}{\nu-1},\delta)$ differentially private. For each of the 6 terms of $\epsilon_1+\frac{\ln(\frac{1}{\delta})}{\nu-1}$, a lower bound on $n$ and $\nu$ is found at equations 53, 54, 55, 56, 57, 58 such that the sum of terms is upper bounded by $\epsilon$. These bounds match the claim's guarantee over $n$ and $\nu$ therefore proving the claim.

For term $\frac{\ln(\frac{1}{\delta})}{\nu-1}$

$$\begin{aligned}\frac{\ln(\frac{1}{\delta})}{\nu-1} &= \frac{\epsilon}{2} \iff \\ \frac{2\ln(\frac{1}{\delta})}{\epsilon}+1 &= \nu\end{aligned}$$

(53)

For term $\frac{x_h^2}{(n-1)x_l^2}$

$$\frac{x_h^2}{2(n-1)x_l^2} \le \frac{\epsilon}{16}$$
$$n \ge 1 + \frac{x_h^2}{x_l^2}\frac{8}{\epsilon}$$

(54)

For term $\frac{1}{2}(\nu - 1)\frac{\nu x_h^2}{(n-1)x_l^2 - \nu x_h^2}($

$$\frac{1}{2}(\nu - 1)\frac{\nu x_h^2}{(n-1)x_l^2 - \nu x_h^2} \le \frac{\epsilon}{16}$$
$$\frac{1}{2}(\nu - 1)\frac{16\nu x_h^2}{\epsilon} \le (n-1)x_l^2 - \nu x_h^2$$
$$n \ge 1 + \frac{1}{2}(\nu - 1)\frac{16\nu x_h^2}{\epsilon x_l^2} + \nu\frac{x_h^2}{x_l^2} = 1 + \nu\frac{x_h^2}{x_l^2}(1 + 8\frac{(\nu - 1)}{\epsilon})$$

(55)

For term $2\nu\beta(\frac{x_h^4}{\frac{9}{10}n^{1-2\gamma_1}x_l^2})$

$$2\nu\beta(\frac{x_h^4}{\frac{9}{10}n^{1-2\gamma_1}x_l^2}) \le \frac{\epsilon}{8}$$
$$\frac{16}{\epsilon}\nu\beta\frac{x_h^4}{\frac{9}{10}x_l^2} \le n^{1-2\gamma_1}$$
$$n \ge (\frac{16\nu\beta x_h^4}{\frac{9}{10}\epsilon x_l^2})^{\frac{1}{1-2\gamma_1}}$$

(56)

For term $2\nu\beta(\frac{(x_h^2\beta)(x_h^2\alpha+x_h^4\beta)}{\frac{9}{10}(x_l^2\beta)^2})\frac{(c+n^{\gamma_1})}{n^{2-\gamma_1}}$

$$2\nu\beta(\frac{(x_h^2\beta)(x_h^2\alpha + x_h^4\beta)}{\frac{9}{10}(x_l^2\beta)^2})\frac{(c+n^{\gamma_1})}{n^{2-\gamma_1}} \le \frac{\epsilon}{8}$$
$$n^{2-\gamma_1} \ge \frac{16\nu\beta}{\epsilon}(\frac{(x_h^2\beta)(x_h^2\alpha + x_h^4\beta)}{\frac{9}{10}(x_l^2\beta)^2})(c+n^{\gamma_1})$$
$$n \ge (\frac{16\nu\beta}{\epsilon}(\frac{(x_h^2\beta)(x_h^2\alpha + x_h^4\beta)}{\frac{9}{10}(x_l^2\beta)^2})(c+n^{\gamma_1}))^{\frac{1}{2-\gamma_1}}$$

(57)

For term $\frac{\nu}{2}(\frac{(x_h^2\alpha+x_h^4\beta)^2}{\frac{9}{10}x_l^6\beta})\frac{(c+n^{\gamma_1})^2}{n^3}$

$$\frac{\nu}{2}(\frac{(x_h^2\alpha + x_h^4\beta)^2}{\frac{9}{10}x_l^6\beta})\frac{(c+n^{\gamma_1})^2}{n^3} \le \frac{\epsilon}{8}$$
$$n^3 \ge \frac{4\nu}{\epsilon}(\frac{(x_h^2\alpha + x_h^4\beta)^2}{\frac{9}{10}x_l^6\beta})(c+n^{\gamma_1})^2$$
$$n \ge (\frac{4\nu}{\epsilon}(\frac{(x_h^2\alpha + x_h^4\beta)^2}{\frac{9}{10}x_l^6\beta})(c+n^{\gamma_1}))^{\frac{2}{3}}$$

(58)

$\square$

**Claim D.30.** *For* $c = n^{\gamma_2}, \gamma_1 < \gamma_2 < \frac{3}{2}$, *and the conditions and definitions of Lemma 4.3, one sample from the posterior is* $(\epsilon, \delta)$ *differentially private for following terms on* $n$ *and* $\nu$.

$$\nu = \frac{2\ln(\frac{1}{\delta})}{\epsilon} + 1$$

$$n \geq \max\{1 + \frac{x_h^2}{x_l^2}\frac{8}{\epsilon}, 1 + \nu\frac{x_h^2}{x_l^2}(1 + 8\frac{(\nu - 1)}{\epsilon}),$$

$$(\frac{16\nu\beta x_h^4}{\frac{9}{10}\epsilon x_l^2})^{\frac{1}{1-2\gamma_1}},$$

$$(\frac{16\nu\beta}{\epsilon}(\frac{(x_h^2\beta)(x_h^2\alpha + x_h^4\beta)}{\frac{9}{10}(x_l^2\beta)^2})(1 + \frac{1}{(1 + 10\frac{x_h^2}{x_l^2}\frac{\nu}{\beta})^{\gamma_2 - \gamma_1}}))^{\frac{1}{2-\gamma_1-\gamma_2}},$$

$$(\frac{4\nu}{\epsilon}(\frac{(x_h^2\alpha + x_h^4\beta)^2}{\frac{9}{10}x_l^6\beta})(1 + \frac{1}{(1 + 10\frac{x_h^2}{x_l^2}\frac{\nu}{\beta})^{\gamma_2 - \gamma_1}}))^{\frac{2}{3-2\gamma_2}}\}$$

*Proof Claim D.30.* Claim D.29 provides general lower bounds on $n$ for $(\epsilon, \delta)$ differential privacy. When $c = n^{\gamma_2}, \gamma_2 > \gamma_1$, these bounds can be simplified.

For condition $n \geq (\frac{16\nu\beta}{\epsilon}(\frac{(x_h^2\beta)(x_h^2\alpha + x_h^4\beta)}{\frac{9}{10}(x_l^2\beta)^2})(c + n^{\gamma_1}))^{\frac{1}{2-\gamma_1}}$,

$$(\frac{16\nu\beta}{\epsilon}(\frac{(x_h^2\beta)(x_h^2\alpha + x_h^4\beta)}{\frac{9}{10}(x_l^2\beta)^2})(c + n^{\gamma_1}))^{\frac{1}{2-\gamma_1}} =$$

$$(\frac{16\nu\beta}{\epsilon}(\frac{(x_h^2\beta)(x_h^2\alpha + x_h^4\beta)}{\frac{9}{10}(x_l^2\beta)^2})n^{\gamma_2}(1 + \frac{1}{n^{\gamma_2 - \gamma_1}}))^{\frac{1}{2-\gamma_1}} \leq$$

$$(\frac{16\nu\beta}{\epsilon}(\frac{(x_h^2\beta)(x_h^2\alpha + x_h^4\beta)}{\frac{9}{10}(x_l^2\beta)^2})n^{\gamma_2}(1 + \frac{1}{(1 + 10\frac{x_h^2}{x_l^2}\frac{\nu}{\beta})^{\gamma_2 - \gamma_1}}))^{\frac{1}{2-\gamma_1}}$$

, where the inequality holds since Lemma 4.3 dictates that $n \geq 1 + 10\frac{x_h^2}{x_l^2}\frac{\nu}{\beta}$. Consequently it's enough that

$$n > (\frac{16\nu\beta}{\epsilon}(\frac{(x_h^2\beta)(x_h^2\alpha + x_h^4\beta)}{\frac{9}{10}(x_l^2\beta)^2})(1 + \frac{1}{(1 + 10\frac{x_h^2}{x_l^2}\frac{\nu}{\beta})^{\gamma_2 - \gamma_1}}))^{\frac{1}{2-\gamma_1-\gamma_2}}.$$

Following same considerations for condition $n \geq (\frac{4\nu}{\epsilon}(\frac{(x_h^2\alpha + x_h^4\beta)^2}{\frac{9}{10}x_l^6\beta})(c + n^{\gamma_1}))^{\frac{2}{3}}$, it is enough that

$$n > (\frac{4\nu}{\epsilon}(\frac{(x_h^2\alpha + x_h^4\beta)^2}{\frac{9}{10}x_l^6\beta})(1 + \frac{1}{(1 + 10\frac{x_h^2}{x_l^2}\frac{\nu}{\beta})^{\gamma_2 - \gamma_1}}))^{\frac{2}{3-2\gamma_2}}$$

$\square$

# E   WASSERSTEIN DISTANCE PROOF

**Claim E.1.** *If* $p, q$ *are distributions with 2-Wasserstein distance* $W_2(p, q) = \epsilon^2$, *then we have* $p(B_r(x)) \leq q(B_{r+\epsilon}(x)) + \epsilon$.

*Proof.* If is the claim that $d_P^2 \leq d_w$ from Gibbs & Su (2002). Picking an optimal coupling and using Markov inequality we get $P(d(x, y) > \epsilon) \leq \frac{1}{\epsilon}\mathbb{E}[d(x, y)] = \epsilon$. As $\{(\tilde{x}, \tilde{y}) : \tilde{x} \in B_r(x)\} \subset \{(\tilde{x}, \tilde{y}) : \tilde{y} \in B_{r+\epsilon}(y)\} \cup \{(\tilde{x}, \tilde{y}) : d(\tilde{x}, \tilde{y}) > \epsilon\}$ we get $p(B_r(x)) \leq q(B_{r+\epsilon}(x)) + \epsilon$ (special case of Strassen theorem). $\square$

**Claim E.2.** *Let $p, q$ be continuous distributions on $\mathbb{R}^d$ with Wasserstein distance $W_2(p, q) < \epsilon^2$, and let $p_\delta, q_\delta$ be their convolutions with uniform distribution on $B_\delta(0)$. We assume both density functions are $L-$Lipshitz continuous. For $\lambda > \epsilon$ we have*

$$p_\lambda(x) \leq \frac{vol_d(\lambda)}{vol_d(\lambda - \epsilon)} q_\lambda(x) + \frac{\epsilon}{vol_d(\lambda - \epsilon)} + 2 \left( \frac{vol_d(\lambda)}{vol_d(\lambda - \epsilon)} - 1 \right) \lambda L. \tag{59}$$

*Proof.* We have $P(B_\lambda(x)) = P(B_{\lambda-\epsilon}(x)) + P(A(x; \lambda - \epsilon, \lambda))$ where $A(x; r_1, r_2)$ is the annulus around $x$ between radius $r_1$ and $r_2$. From continuity there exists $z \in P(B_\lambda(x))$ such that $p(z) = \frac{P(B_\lambda(x))}{vol_d(\lambda)}$, where $vol_d(r)$ is the volume of a ball of radius $r$ in $\mathbb{R}^d$. From Lipshitz continuity we have $P(A(x; \lambda - \epsilon, \lambda)) \leq (vol_d(\lambda) - vol_d(\lambda - \epsilon))(p(z) + 2\lambda L) = \left(1 - \frac{vol_d(\lambda-\epsilon)}{vol_d(\lambda)}\right) P(B_\lambda(x)) + \Delta$, where $\Delta = (vol_d(\lambda) - vol_d(\lambda - \epsilon))2\lambda L$. From this, we get

$$P(B_{\lambda-\epsilon}(x)) \geq \frac{vol_d(\lambda - \epsilon)}{vol_d(\lambda)} P(B_\lambda(x)) - \Delta. \tag{60}$$

Combining this with claim E.1, we get

$$P(B_\lambda(x)) \leq \frac{vol_d(\lambda)}{vol_d(\lambda - \epsilon)} (P(B_{\lambda-\epsilon}(x)) + \Delta) \leq \frac{vol_d(\lambda)}{vol_d(\lambda - \epsilon)} (Q(B_\lambda(x)) + \Delta + \epsilon). \tag{61}$$

We divide by $vol_d(\lambda)$ to get the densities $p_\lambda, q_\lambda$.

$$p_\lambda(x) \leq \frac{vol_d(\lambda)}{vol_d(\lambda - \epsilon)} q_\lambda(x) + \frac{\Delta + \epsilon}{vol_d(\lambda - \epsilon)} \tag{62}$$

$\square$

