# OpenReview forum: "Can Stochastic Gradient Langevin Dynamics Provide Differential Privacy for Deep Learning?"
_ICLR.cc/2022/Conference — ICLR 2022 Submitted_

### Official Review · Reviewer_1F53 · 2021-10-30

**Correctness:** 3
**Technical Novelty And Significance:** 3
**Empirical Novelty And Significance:** Not applicable
**Recommendation:** 3
**Confidence:** 4

**Main Review:**

Strengths:
1, It is interesting to know that SGLD can result in unbounded privacy loss during the middle of the sampling procedure.
2, Figure 1 clearly illustrates the main idea of the claiming point.

Weaknesses:
1, Theoretical results in this paper are based on Bayesian simple linear regression problems as shown in Eq(4) page 3.
However,  the paper mentions Bayesian neural networks in several misleading places. For example: in the abstract, "This interim region is essential, especially for Bayesian neural networks, as it is hard to guarantee convergence to the posterior" or in introduction "Neither of these cases is suitable for deep learning and many other problems, as one would limit the model’s accuracy and the other is unattainable in a reasonable time" in page 1. The authors should make it clear about their contributions. So readers can position this paper appropriately.

2, Subsections 4.1 - 4.2 are proof sketches to show that approximate sampling of the posterior with Bayesian linear regression by SGLD is not differential private in some steps (Theorem 1).  It is better to put them into a new section and explicitly state their relation with Theorem 1.

3， Section 4.3 tries to remove unknown c in eq 5. However, $p(\theta | W)$ is not the original posterior $p(\theta | D)$ based on dataset $D$ anymore.  What's the relationship between $p(\theta | W)$ and $p(\theta | D)$?

4,  The manuscript is not ready and needs to be further proofread. For example, comma and period are missing in many places:
"In this section, we will consider the differential privacy guarantees provided by taking one sample from the posterior for the Bayesian linear regression problem on domain D". Similarly, Theorem 1, Lemma 4.1  4.4 and 4.5
It also should be "steps" in "but there will be some step in which SGLD will result in unbounded loss of privacy."

**Summary Of The Paper:**

The paper studies the differential privacy of stochastic gradient Langevin dynamics(SGLD) as MCMC method. The paper shows that approximate sampling using SGLD may result in an unbounded privacy loss in the middle regime, via Bayesian linear regression.

**Summary Of The Review:**

The paper explores the privacy-preserving performance of SGLD and shows its privacy loss can be unbounded in the middle regime of sampling. The finding is interesting and useful.
However, the authors should make their contributions clearly and some sentences are misleading. The paper should also be re-organized and proofread before it can be accepted.

---

> ### Author Response · Authors · 2021-11-14
> **Response to Reviewer 1F53**
>
> We thank the reviewer for his comments. Please see our responses below.
>
> **1**
> > “Theoretical results in this paper are based on Bayesian simple linear regression problems [..]. However, the paper mentions Bayesian neural networks in several misleading places. For example: [...]. The authors should make it clear about their contributions. So readers can position this paper appropriately.”
>
> Thank you for pointing out that a clarification is needed. Please see our explanation about the generality and contribution of our results in the general comment.
>
> **2**
> > “Subsections 4.1 - 4.2 are proof sketches [..]. It is better to put them into a new section and explicitly state their relation with Theorem 1.”
>
> Thank you for pointing out this issue. We will add an explanation about what each section contains for better readability.
>
> **3**
> > “Section 4.3 tries to remove unknown c in eq 5. However, $P(\theta | W)$ is not the original posterior $P(\theta | D)$ based on dataset $D$ anymore. What's the relationship between  $P(\theta | W)$ and $P(\theta | D)$?
>
> $W$ is formed by preprocessing $D$. The preprocessing includes enforcing the data into a bounded region and removing outliers (lines 1-14). Both of these steps are considered standard practice in machine learning. Since few outliers can considerably affect $P(\theta|D)$, then $P(\theta | D)$ and $P(\theta | W)$ might differ if such outliers exist in the data.
>
> **4**
> > “The manuscript is not ready and needs to be further proofread. For example, [...]. Similarly, Theorem 1...”
>
> Thank you for letting us know about these mistakes in the manuscript. We will proofread it again and fix them.

---

### Official Review · Reviewer_48Es · 2021-10-31

**Correctness:** 3
**Technical Novelty And Significance:** 3
**Empirical Novelty And Significance:** Not applicable
**Recommendation:** 6
**Confidence:** 3

**Main Review:**

Overall, this paper presents a rigorous analysis of differential privacy of Bayesian learning using SGLD. It uses Bayesian linear regression as a simple example to demonstrate that while differential privacy holds at the beginning of the SGLD updates and similarly at the convergence, but it may not hold during the intermediate steps of SGLD updates. Both the theoretical analysis and the empirical graph in Figure 1 backs their claim.

The paper is mainly a theoretical paper and seems to appropriately analyse the differential privacy of SGLD. The claims seems accurate although I could not verify the details of all the proof as it is fairly long.

Having said that, the paper can be significantly improved in its writing. At many places, it assumes a lot of background from the reader and uses terms without providing required explanations. For example, On Page 2, when discussing Ma et al. (2019), it mentions about  \epsilon-mixing time bound without providing any clear context or explanation.

Also, as per my understanding, there are a couple of statements which seem incorrect: On page 2, when starting to discuss differential privacy, it says “…, a differentially private
algorithm promises the data owners that their *utility* will not change, with high probability, by adding their data to the algorithm’s database.” I do not think differential privacy makes any claim on utility.

Theorem 1, which is a key result of this paper uses three notations: \epsilon, \epsilon’, \epsilon’’. The role of \epsilon and \epsilon’ does not seem clear. It appears there is an error. Are \epsilon and \epsilon’ same?

On page 4, the parameters n; c; xl; xh; gamma_1 etc are not explained properly.

The Figure 1 is referred as Figure 4 – should be corrected.

On page 7, the sentence “It then estimates the average slope and throws away the outliers that deviate too much from the average slope.” It is not clear what authors mean by “slope” here?

Spelling errors:
-On page 4: “known” should be “know”
-On page 4: “a well known results” should be “well known results”!
-On page 6: “peeked” should be “peaked”
-At many places in the text, “i’th”, “j’th” etc use the math symbols without Latex mode.
-A lot of places, full stop is missing (both in text and in Lemma statements).


**Summary Of The Paper:**

This paper studies the privacy guarantee of Bayesian learning using Stochastic Gradient Langevin Dynamics (SGLD). Since the SGLD updates are stochastic, it is often thought the solution can be suitable for privacy-preserving of the data that is used to train the algorithm. Using a counter-example, this paper shows that it is not necessarily correct to assume so.

**Summary Of The Review:**

This paper analyses the differential privacy of the SGLD algorithm. It uses Bayesian linear regression as an example to demonstrate that while differential privacy holds at the beginning of the SGLD updates and similarly at the convergence, but it may not hold during the intermediate steps of SGLD updates. The results seem convincing.

---

> ### Author Response · Authors · 2021-11-14
> **Response to Reviewer 48Es**
>
> We thank the reviewer for his comments. Please see our responses below.
>
> **1**
> > “the paper can be significantly improved in its writing. At many places, it assumes a lot of background from the reader and uses terms without providing required explanations.”
>
> Thank you for drawing our attention to the missing explanations for some of the terms. We will fix this issue.
>
> **2**
> > “On page 2, when starting to discuss differential privacy, it says [...]. I do not think differential privacy makes any claim on utility.”
>
> Please note that we considered the Economic View of DP, as described in [1] - section 2.3.1. However, to prevent confusion, we will rephrase.
>
> **3**
> > “Theorem 1, [..]. The role of \epsilon and \epsilon’ does not seem clear. It appears there is an error. Are \epsilon and \epsilon’ same?”
>
> Indeed, the phrasing might be a bit complicated. The theorem says that we can always find a Bayesian inference problem such that the posterior can be as private as desired, but sampling using SGLD might be as non-private as desired. $\epsilon$ and $\epsilon’$ are different. $\epsilon$ describes the privacy at the posterior. $\epsilon’$ describes the lack of privacy in the interim region. We show that $\epsilon’$ can be much bigger than $\epsilon$.
>
> Considering your comment, we can change the phrasing of the theorem to a clearer version by removing the use of $\epsilon$$’$$’$:
> $\forall\ \delta < 0.5, \epsilon, \epsilon'$ there exist a domain and Bayesian inference problem where a single sample from the posterior distribution is $(\epsilon, \delta)$ differentially private, but, when performing approximate sampling by running SGLD, there is a number of steps $T$ such that sampling from SGLD after $T$ iterations is not $(\epsilon', \delta)$ differentially private.
>
> **4**
> > “On page 4, the parameters n; c; xl; xh; gamma_1 etc are not explained properly.”
>
> Thank you for pointing out this issue. We will add explanations.
>
> **5**
> > “The Figure 1 is referred as Figure 4 – should be corrected.”
>
> Thank you for pointing out this issue. We will fix it.
>
> **6**
> > “On page 7, the sentence “It then [...] slope.” It is not clear what authors mean by “slope” here?”
>
> By "slope", we meant for $\frac{y_i}{x_i}$ and the average slope is as computed in line 12 (Maybe better called "weighted average"). From your review, we understand that using "slope" might be misleading. We will fix it to $\frac{y_i}{x_i}$, and refer to the specific line when discussing about taking the average.
>
> **7**
> > “Spelling errors: - [...] -At many places in the text, “i’th”, “j’th” etc use the math symbols without Latex mode. -A lot of places, full stop is missing (both in text and in Lemma statements).”
>
> Thank you for letting us know about these spelling errors, missing Latex mode, and missing full stops. We will fix these issues.
>
> [1] Cynthia Dwork and Aaron Roth. The algorithmic foundations of differential privacy. Found. Trends Theor. Comput. Sci., 9(3–4):211–407, August 2014. ISSN 1551-305X. doi: 10.1561/0400000042. URL https://doi.org/10.1561/0400000042.

---

> ### Comment · Reviewer_48Es · 2021-11-30
> **Post rebuttal comments**
>
> I have read author's rebuttal. While there were several clarity issues, the authors have promised to improve these issues in the new version, so I keep my score of marginally above the acceptance threshold.

---

### Official Review · Reviewer_gB6E · 2021-10-31

**Correctness:** 4
**Technical Novelty And Significance:** 2
**Empirical Novelty And Significance:** 2
**Recommendation:** 5
**Confidence:** 4

**Main Review:**

1. The counterexample constructed is fairly restrictive. It is for a particular model, for a particular data set, for a particular stochastic scheme (i.e., cyclic-SGLD) and for a particular learning rate. Does the same result hold if we use the common sample in each step of the SGLD? So I doubt that this example provides general insights.
2. According to the proofs in the appendix, k in Lemma 4.5 is fairly small. In other words, the privacy breach can occur when the SGLD has only scanned the full data set a very small number of times (less than 10 epochs, as shown in Figure 1). Thus, even for this particular counterexample, I don't think the result is practically meaningful.
3. Section 5 is really an incomplete analysis.

**Summary Of The Paper:**

This paper provides one concrete example, showing that revealing one posterior sample generated by SGLD has the risk of a privacy breach when the SLGD sampling iterations number is moderate, while the exact posterior sampling has little risk of a privacy breach.

**Summary Of The Review:**

Overall, I consider the contribution of the paper is quite restrictive.

By definition, it is sufficient to find a pair of neighboring data sets to counterprove the loss of privacy. But the results also depend on the specific setup of the SGLD algorithm, which I believe is not very proper.

In the common privacy-preservation algorithm, one typically injects Laplace or Gaussian noise. To show it works, we always need to have some lower bound of noise variance. Similarly, if SGLD preserves privacy, there are potentially some requirements on the algorithm implementation. To counter-prove that, I suppose one needs to show that no matter how one tunes the SGLD algorithm, the privacy breath is inevitable.

---

> ### Author Response · Authors · 2021-11-14
> **Response to Reviewer gB6E**
>
> We thank the reviewer for his comments. Please see our responses below.
>
> **1**
> > “The counterexample constructed is fairly restrictive. [...] for a particular stochastic scheme (i.e., cyclic-SGLD) [...]. Does the same result hold if we use the common sample in each step of the SGLD? So I doubt that this example provides general insights.”
> > “In the common privacy-preservation algorithm, one typically injects Laplace or Gaussian noise. To show it works, [...] I suppose one needs to show that no matter how one tunes the SGLD algorithm, the privacy breath is inevitable.”
>
> Please note that we do not wish to prove that SGLD is never private but to show that it might not be private. Thus, no guarantees can be given. For the importance of this claim, please see our explanation about generality in the general comment.  Regarding the usage of cyclic-SGLD, it is common practice to use cyclic-SGD in machine learning, and it was proved to converge at a faster rate than non-cyclic SGD (see [1]).
>
> **2**
> > “According to the proofs in the appendix, k in Lemma 4.5 is fairly small. [...] the privacy breach can occur when the SGLD has only scanned the full data set a very small number of times (less than 10 epochs, as shown in Figure 1). Thus, even for this particular counterexample, I don't think the result is practically meaningful.”
>
> Please note that from Figure 1, the highest privacy breach occurs approximetly at epoch 40. Also, this is merely one example. As $\dot{k}$ has a closed form solution we can find it’s value for different scenarios. For example, a dataset of size 1131965, with $\alpha = 0.1$, $\beta = 1$ and $x_h = 5$, will give $\dot{k} \approx 485$.
>
> **3**
> > “Section 5 is really an incomplete analysis”
>
> Please see our clarification on section 5 in the general comment.
>
> [1] - Chulhee Yun, Suvrit Sra, and Ali Jadbabaie. Open problem: Can single-shuffle SGD be better than reshuffling SGD and gd? In Conference on Learning Theory, COLT, 2021.

---

### Official Review · Reviewer_AyXE · 2021-11-01

**Correctness:** 4
**Technical Novelty And Significance:** 3
**Empirical Novelty And Significance:** Not applicable
**Recommendation:** 5
**Confidence:** 4

**Main Review:**

This work is quite interesting and important in the sense that SGLD is used in many works in literature and it is before proved that SGLD with specific parameter choices provides (\epsilon, \delta)-DP. This paper finds a counter example to the previous finding with correct analysis. However the structure of the paper can be improved. In the title and introduction, it is claimed that SGLD might not provide (\epsilon, \delta)-DP for deep learning. But the analysis are made for Bayesian linear regression. It is not clear to me that whether it is generalizable to (Bayesian) deep neural networks or not.

One weakness of this paper is the literature review. There are papers that uses SGLD for differentially private deep learning, it will be very useful to cite these works to understand whether these methods provide (\epsilon, \delta)-DP eventually or not.

It is confusing what is proposed in Section 5. It is mentioned that the bound scale poorly with dimension, but can still be useful for Bayesian sampling in low-dimensional problems. Is the method still proposed for (\epsilon, \delta)-DP for deep networks as an alternative?

**Summary Of The Paper:**

This paper shows that even when the posterior is as private as targeted in the beginning, sampling from posterior with SGLS might not be as private as targeted. The authors prove the theorem on Bayesian linear regression problem. They prove that for n big enough sampling from the posterior is (\epsilon, \delta) differentially private (DP), but there is a step in which releasing a sample will not be (\epsilon^\prime, \delta)-DP for \epsilon^\prime=\omega(n \epsilon).

**Summary Of The Review:**

It is important to show SGLD might not always give (\epsilon, \delta) differential privacy guarantee. But the text should be improved to clarify the points that I mentioned above. Maybe the title and the introduction should be revised, or some analysis could be added to show this is also applicable to deep learning.

---

> ### Author Response · Authors · 2021-11-14
> **Response to Reviewer AyXE**
>
> We thank the reviewer for his comments. Please see our responses below.
>
> **1**
> > “However the structure of the paper can be improved. [...] It is not clear to me that whether it is generalizable to (Bayesian) deep neural networks or not.”
>
> > "Maybe the title and the introduction should be revised, or some analysis could be added to show this is also applicable to deep learning."
>
> Thank you for pointing out this issue. Please refer to our explanation on generality in the general comment.
>
> **2**
> > “One weakness of this paper is the literature review. There are papers that uses SGLD for differentially private deep learning, it will be very useful to cite these works to understand whether these methods provide (\epsilon, \delta)-DP eventually or not.”
>
> Indeed we mentioned more papers which discuss the DP of SGLD (see [1], [2], [3], [4], [5], [6]). We provided a short explanation of the privacy guarantees offered in [1]. Mainly, the paper presents decent privacy guarantees for SGLD only for a limited number of steps (or step size) and at approximate convergence. When considering the interim regime, the paper does not provide any guarantees. The rest of the papers follow similar lines, providing guarantees only for a limited number of steps or for the posterior. We can, however, provide further explanations for each of those papers.
>
> **3**
> > “It is confusing what is proposed in Section 5.[...] Is the method still proposed for (\epsilon, \delta)-DP for deep networks as an alternative?”
>
> Thank you for raising this question. Please see our clarification on section 5 in the general comment.
>
>
> [1] Yu-Xiang Wang, Stephen Fienberg, and Alex Smola. Privacy for free: Posterior sampling and stochastic gradient monte carlo. In Proceedings of the 32nd International Conference on Machine Learning, volume 37 of Proceedings of Machine Learning Research, pp. 2493–2502, Lille, France, 07–09 Jul 2015. PMLR. URL https://proceedings.mlr.press/v37/ wangg15.html.
>
> [2] James R. Foulds, Joseph Geumlek, Max Welling, and Kamalika Chaudhuri. On the theory and practice of privacy-preserving bayesian data analysis. In Uncertainty in Artificial Intelligence, UAI, 2016.
>
> [3] Zuhe Zhang, Benjamin I. P. Rubinstein, and Christos Dimitrakakis. On the differential privacy of bayesian inference. In AAAI Conference on Artificial Intelligence, 2016.
>
> [4] Christos Dimitrakakis, Blaine Nelson, Zuhe Zhang, Aikaterini Mitrokotsa, and Benjamin I. P. Rubinstein. Differential privacy for bayesian inference through posterior sampling. Journal of Machine Learning Research, 18(11):1–39, 2017. URL http://jmlr.org/papers/v18/ 15-257.html.
>
> [5] Joseph Geumlek, Shuang Song, and Kamalika Chaudhuri. Renyi differential privacy mechanisms for posterior sampling. In Advances in Neural Information Processing NeurIPS, 2017.
>
> [6] Arun Ganesh and Kunal Talwar. Faster differentially private samplers via renyi divergence analysis ´ of discretized langevin mcmc. ArXiv, abs/2010.14658, 2020.

---

### Author Response · Authors · 2021-11-14
**General comment**

We want to thank the reviewers for their time, effort, and thoughtful comments. We were encouraged by their description of the work as *quite interesting and important* (R-AyXE), the results as *convincing* (R-48Es), and the finding as *interesting and useful* (R-1F53).  The reviewers found it *important* to show that SGLD might not always give $(\epsilon, \delta)$ - differential privacy guarantee (R-AyXE).

A common concern raised by part of the reviewers (R-AyXE, R-gB6E, R-1F53) regarded the generality and contribution of our results for Deep Neural Networks (DNNs). Following is an explanation of the relevance of our results for DNNs.

We claim in the paper that when using SGLD on DNNs, the area of interest is the interim regime. This is because running until guaranteed convergence isn’t feasible, and running a small number of steps results in low performance (also, for a small number of steps, privacy is based on Gaussian Mechanism and not the Bayesian approach). Consequently, the question of whether SGLD can provide privacy guarantees in the interim regime is of great importance for DNNs.

The need for guarantees in this interim regime becomes even more critical, as we cannot evaluate privacy empirically for DNNs. Even if SGLD can be private for a specific problem or when using a specific learning rate, we do not know in advance when will it be private. Our counterexample shows that SGLD can have a substantial privacy breach at the interim regime, even for simple convex problems and even when the posterior is as private as desired. Therefore, guarantees cannot be given even under strong conditions, stronger than what we can assume for DNNs. We will rewrite the paper to make this point clearer.

Due to comments on section 5 by reviewers R-AyXE and R-gB6E, we provide clarification over section 5. The method described in section 5 isn't useful for deep networks since the method fails to scale. We tried to gain valuable results for Wasserstein distance and got an interesting result, but it does not scale. However, we thought this result was interesting and could be of use for low-dimensional problems, so we decided to add it to the paper. As it is only applicable for low dimensions, we can move this section to supplementary.

---

### Decision · Program_Chairs · 2022-01-20

**Decision:**

Reject

**Comment:**

This paper shows that SLGD can be non-private (in the sense of differential privacy) even when a single step satisfies DP and also when sampling from the true posterior distribution is DP. I believe that it is useful to understand the behavior of SLGD in the intermediate regime. At the same time the primary question is whether SLGD is DP when the parameters are chosen so as to achieve some meaningful approximation guarantees after some fixed number of steps T and the algorithm achieves them while satisfying DP (but at the same does not satisfy DP for some number of step T' >T). Otherwise the setting is somewhat artificial and I find the result to be less interesting and surprising. So while I think the overall direction of this work is interesting I believe it needs to be strengthened to be sufficiently compelling.